

# Functional renormalization group approach for signal detection

Vincent Lahoche[1⋆], Dine Ousmane Samary[1,2†] and Mohamed Tamaazousti[1‡]

**1** Université Paris Saclay, CEA, LIST, Gif-sur-Yvette, F-91191, France
**2** Faculté des Sciences et Techniques (ICMPA-UNESCO Chair),
Université d'Abomey-Calavi, 072 BP 50, Bénin

⋆ vincent.lahoche@cea.fr , † dine.ousmanesamary@cipma.uac.bj ,
‡ mohamed.tamaazousti@cea.fr

## Abstract

This review paper utilizes renormalization group techniques for signal detection in nearly continuous positive spectra. We emphasize the universal aspects of the analogue field-theory approach. The primary objective is to present an extended self-consistent construction of the analogue effective field-theory framework for data, which can be interpreted as a maximum entropy model. In particular, we leverage universality arguments to justify the $\mathbb{Z}_2$ symmetry of the classical action, highlighting the existence of both a large-scale (local) regime and a small-scale (nonlocal) regime. Secondly, in relation to noise models, we observe the universal relationship between phase transitions and symmetry breaking near the detection threshold. Finally, we address the challenge of defining the covariance matrix for tensor-like data. Based on the cutting graph prescription, we note the superiority of definitions that rely on complete graphs of large size for data analysis.

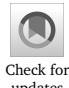
# 1 Introduction

**Renormalization group and AI.** Techniques of data analysis are mainly considered as useful tools for physicists in order to investigate experimental results comprising a massive number of data, which is of great importance nowadays in applied domain. One of the purposes of this paper is to emphasise that data analysis can be also viewed as a physical problem involving an unconventional Euclidean field theory. Moreover, such a framework is specially adapted to an analysis by the renormalization group (RG), the specificity of the theory bringing new and challenging issues. In physics, the RG is a general concept relevant to tackling problems involving a very large number of interacting degrees of freedom [1–5]. All the technical incarnations of this concept aim to perform the same thing: to approximate the exact (but generally partially known) description by a simple but effective theory where the relevant variables from statistical or quantum states are extracted. This theory ignores some details, which are irrelevant at a given level of description regarding some experimental precision. In other words, RG aims to identify physical states that cannot be distinguished experimentally, the basin of attraction of the effective description defining an equivalence class of microscopic states. As a general concept, RG has a numerous range of applications among them statistical mechanics of critical phenomena, glassy and out-of-equilibrium systems, turbulence, particle physics, quantum gravity, etc [6]. Consequently, it appears that RG is more than a technical trick, it is a conceptual key for understanding the modern physics.

Gell-Mann and Low [7] first proposed the concept of RG when using the reflection about scale transformations in quantum electrodynamics in particle physics. Later, Kadanoff and Wilson [8–11] applied the conception of RG into statistical mechanics of critical phenomena. They proposed to replace the global description of the system involving integral's overall wavelength with a sliced description, integrating momenta gradually into slices containing a few numbers of modes. Integrating out degrees of freedom with short wavelengths provide an effective description for the remaining long wavelengths degrees of freedom, involving effective Hamiltonian. Therefore, RG transformations proceed by eliminating degrees of freedom

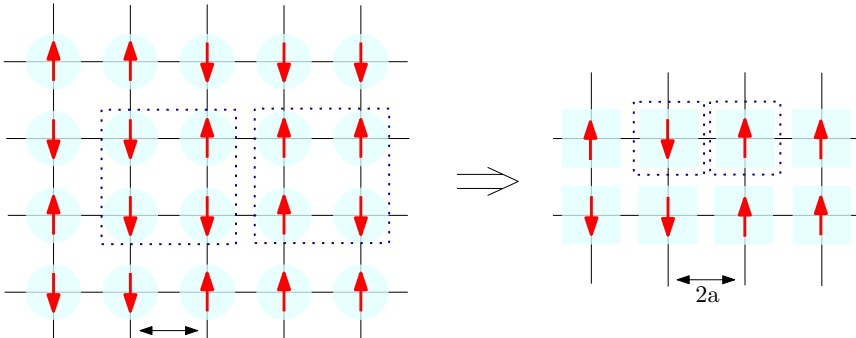

Figure 1: Illustration of the Kadanoff's block-spin RG transformation $\mathcal{T}$: Spins in the initial lattice with spacing $a$ are averaged into blocks of four spins, and interactions between spins are replaced by interactions between blocks with spacing $2a$.

through a coarse-grained description. The Kadanoff's "block-spin" delivers another incarnation of the same "coarse-graining" approach: for Ising model, it proposes to replace the sum over all spin configurations with constrained sums having fixed average values in the interior of cells. Figure 1 recalls the general strategy for the standard Ising model on a square lattice. Averaging into blocks of four spins, the initial Hamiltonian $H(J, a, \phi)$ for some spin configuration $\phi$ and coupling $J$ is transformed as an effective Hamiltonian $H(J', 2a, \phi') = \mathcal{T}[H(J, a, \phi)]$, describing interactions into blocks. The iteration of this transformation maps Hamiltonians onto Hamiltonians (i.e. models onto models), all encoding the same long-distance physics but describing interactions between very different objects.

All these RG specific realizations/applications underline its general concept that is able to extract relevant features for systems involving a very large number of interacting degrees of freedom. For this reason, the RG is considered as a clever way to dilute the information and with strong connections with information theory [12–16], showing its evolution within large number of applications.

Wilson's partial integration procedure is not necessary to construct RG. Indeed as identified by Wegner from a reflection about reparametrization invariance, RG transformations can be viewed as a suitable change of variables between so-called "fields"; and "couplings"; [17–19]. In other words, and without loss of generalities, RG is a reparametrization of the partition function. Information theory provides a nice explanation of why RG trajectories are relevant: the RG flow corresponds to a form of entropic dynamics of field configurations, mathematically equivalent to a local application of the statistical inference with maximum entropy prescription [20, 21].

Indeed, the inference problem of recovering the microscopic distribution from the partial knowledge of the large-scale effective theory is equivalent to finding the equivalence class of microscopic distributions in its basin of attraction. Quantitatively, the ability to provide a clear distinction between two large-scale asymptotic states starting in two different equivalent classes depends on the property that the relevant perturbations survive or not at a large scale. This relevance moreover can be understood intrinsically from information theory, accordingly to a suitable notion of "distance" between states in the information geometric point of view, where state space looks like a differential manifold with metric given by the Fisher information 2-form [15, 22]. In that setting, a computable measure of (relative) distinguishable is provided by the relative entropy, and the notion of equivalent class can be thought as follow: states having "distance" smaller than some working precision being said equivalents [13, 23, 24].

The last modern important application of the RG probably concerns artificial intelligence (AI). Indeed, just in the past decade the number of publications linking RG, data analysis and machine learning has significantly increased[1] [25–35], regardless whether it is used for a simple analogy (for instance to interpret the behaviour of neural networks) or as a basis for a new approach. It is not surprising that RG and AI techniques have non-vanishing intersections: all techniques of modern data analysis aim to extract relevant (i.e. exploitable) regularities of correlated datasets of very large dimensions such as RG do. Note that RG provides not only a dimensional reduction but also a non-trivial clustering through the existence of different universality classes. Even only for this perspective, it is justified to search for links between the methods of the RG in physics and those of AI.

In this regard, an interesting relation between these two techniques has been established with many machine learning tools, especially with principal component analysis (PCA). PCA is one of the most popular methods to suppress redundancy and denoising in a raw dataset. There are many incarnations of the general PCA strategy, which has been extensively used since more than one hundred years in various scientific domains [36], in particular: condensed matter physics [37], high- energy physics [38–40], quantitative finance [41,42], biology and neuroscience [43,44], chemistry [45,46], geology [47], computer vision [48], random matrix theory [49], machine and deep learning [50,51].[2] Basically, PCA works as a linear projection along the vector space spanned by eigenvectors corresponding to the larger eigenvalues of the covariance matrix. Standard PCA however works efficiently for datasets whose covariance matrix's spectrum exhibits a few numbers isolated spikes out of a bulk made of delocalized eigenvectors. In that way, a very small number of modes captures the most relevant features of the covariance.

**State of the art.** This publication is continuation of several studies, not only ours, aiming to exploit the complementary between PCA and RG to tackle the problem of signal detection in nearly continuous spectra. A first tentative to connect them was investigated in [29, 52–54]. The authors interpret the arbitrary separation between noise and information as a physical cut-off, $\Lambda$, and investigate the scaling behaviour of couplings when this cut-off changes. To construct the RG flow, they propose to describe correlations in complex datasets throughout an interaction of statistical field theory at equilibrium with $\mathbb{Z}_2$ symmetry, describing an unconventional kind of matter filling a fictitious space of dimension 1. The *interacting particle spectrum* is moreover assumed to be given by the eigenvalues of the covariance matrix of data. For a dataset taking the form of a suitably mean-shifted and normalized $N \times P$ matrix $X = \{X_{ai}\}$, with $a \in \{1, \cdots, P\}$ and $i \in \{1, \cdots, N\}$, the covariance matrix $C$ is defined as the average of $X^T X$, describing correlations between type-$i$ variables. The spectrum of the covariance matrix provides a non-trivial notion of scale, from which we can construct a Wilsonian coarse-graining, integrating out smallest eigenvalues at first following a suitable slicing. Following the standard definition, we call deep ultraviolet (UV) the region of small eigenvalues and deep infrared (IR) the region of large eigenvalues. We could have expected that for a purely noisy datum the Gaussian fixed point would be stable and that any deviation from the Gaussian distribution would be irrelevant from coarse-graining. Indeed, seeing a noisy signal as "the least organized as possible" (i.e. having maximum entropy), it is therefore tempting to associate information as an underlying organization for which interactions would be likely to account.

---

[1]The given list is far from exhaustive.

[2]The range of applications is at least as numerous as the one of the RG.

Nevertheless, it has been shown that this naive expectation is wrong. For instance, the analytic MP law, in the large eigenvalue region of the spectrum (deep IR), quartic and sixtic *local* perturbations are relevant.[3] The situation is even worse in the small eigenvalue region (deep UV), where the number of relevant couplings becomes arbitrarily large, and canonical dimensions take arbitrary large values. These conclusions are as universal as MP's law and are essentially insensitive to the sparsity for eigenvalues for simulated random behaviours. Furthermore, this observation does not concern only MP. In [55], non-analytic noises for tensorial data have been investigated and exhibit similar behaviour regarding the instability of the Gaussian behaviour.

Another surprising observation is that a strong enough signal merged in some universal noise makes quartic and sixtic couplings irrelevant and the Gaussian distribution stable. Thus, it is possible to characterize the presence of a signal in a spectrum by the asymptotic properties of the physical states they underlie. These observations were however confined to dimensional aspects and should require further investigations of the RG flow of these theories.

In [52–55], the authors exploited the effective average action (EAA) method [6, 56–60] in this unconventional context. The originality of this framework is to focus on the effective action for integrated-out degrees of freedom rather than on the classical action for remaining degrees of freedom. Hence, the bare (i.e. the microscopic) action is left unchanged but infrared contributions are suppressed from the effective action, including quantum effects. Mathematically the interpolation between UV and IR physics is provided by the effective average action $\Gamma_k$, the effective action for integrated-out degrees of freedom up to the scale $k$. In contrast to the previous picture where the UV cut-off $\Lambda$ looks like a separation between information and noise, the cut-off $k$ looks like an IR rather than a UV cut-off. This change of point of view corresponds also to a change of paradigm: in some sense, the authors focus on determining what the "noise" is rather than what the "information" is. It should be noted, that this difference is not just convenient for technical reasons. The noise models are likely more general than the signal patterns.

Regarding the use of nonperturbative techniques, there are essentially two arguments to justify it. The first one is that we expect that the perturbation theory fails due to the relevance of some interactions in the deep IR for purely noisy spectra. The second one is that the effective Hamiltonian, as those obtained following Wilson-Kadanoff strategy, is a very abstract object. In contrast, working with the EAA, $\Gamma_k$ allows us to make easy contact with physically relevant quantities like effective potential. EAA has been considered in [53–55] for RG investigations based on spectra obtained as a controlled deformation around analytic MP law. The surprising lessons of these investigations are summarized by the following "empirical" statement:

**Empirical statement 1** *Concerning the effective local matter field whose particles density spectrum is given by the empirical eigenvalue distribution of the data's covariance matrix (assumed positive and nearly continuous), it has been observed that:*

- *For purely noisy data, only local quartic and sixtic couplings can be relevant to marginal in the large eigenvalue region (IR) domain. Moreover, there is a non-vanishing compact region around the Gaussian fixed point where all trajectories end toward the $\mathbb{Z}_2$ symmetric phase.*

- *A strong enough signal makes the quartic and sixtic local couplings irrelevants. Moreover, it induces a lack of symmetry restoration in the deep IR, for some trajectories which end continuously toward a broken phase. Hence the strength of the signal plays the role of the inverse of the temperature $\beta = 1/T$ in the physics of phase transitions.*

---

[3]By local interactions, we mean here "interactions at contact point". An extended discussion is provided in section 3.

It is important to realize that these results do not depend on specific details of any special problem. Nonetheless, they stress a general feature of nearly continuous spectra in the vicinity of the Universal MP law. One can think that it is a specific property of the MP law, which is only one example of a universal model of practical interest. The results reported in [53–55] point out that these conclusions are in fact not restricted only to the MP law. Investigating the case of a noise materialized by a random tensor considered in the mathematical formalism of the tensorial PCA [61–64], the authors have confirmed the conclusions of statement 1.

**Purpose of this paper.** This paper follows (continues) the previous bibliographic line. Written for a physicist audience, its first goal is to provide a complete and as self-contained as possible presentation of the underlying field theory. In particular, we propose a derivation of the field theory framework which exploits the universality of models of noise from an explicit construction using large binary datasets, looking like a kind of Ising model familiar to physicists. The second main goal of this paper is to provide solid evidence for a generalization of the Empirical statement 1 through a systematic investigation of different well-known noise models. The challenges and motivations underlying this study are fully discussed in Section 2.

**Remark for the physicist reader:** Even if we voluntary adopted a physicist-oriented redaction, we opted for a pedagogical approach in order to make it more accessible to a wider public, and notably specialists closer to information theory or artificial intelligence, who could be interested in our study.

## 2 Motivations and outline

### 2.1 Short review on the spiked matrix models

The spiked matrix model [65–67] illustrates the paradigmatic problem of understanding the eigenvalue distribution of a matrix built as a few numbers of prominent eigenvectors planted into a random matrix. So far to be as simple as one can suspect, this model provide a very useful statistical model for PCA. In a famous paper [68], Péché, Ben-Arous and Baik showed that in the Wishart ensemble the spiked model exhibits a sharp phase transition when the strength of the signal materialized by the spike reach some critical value. Detection and recovering are allowed only from this point and below this critical value the larger eigenvalue remains embedded in the bulk of the delocalized eigenvectors. This section start with a short review of the one-spike matrix models and the underlying phase transition, focusing on the simplified case of the *Gaussian Wigner model* (GWM) disturbed with a single deterministic vector. More concretely we materialize the data that we are aiming to investigate as a $N \times N$ real random matrix $Q$ whose entries split as a sum of two contributions:

$$Q_{ij} = \beta u_i u_j + \frac{1}{\sqrt{N}} M_{ij}, \tag{1}$$

where $u = (u_1, \cdots, u_N) \in \mathbb{R}^N$, $|u| = 1$ is a suitably normalized vector, and $M$ is a $N \times N$ real orthogonal matrix whose entries are assumed to be randomly distributed accordingly to the *Gaussian orthogonal ensemble* (GOE) i.e. off-diagonal entries are distributed accordingly to $\mathcal{N}(0, 1)$ whereas the diagonal entries are distributed as $\mathcal{N}(0, 2)$.[4] $\beta \in \mathbb{R}^+$ materializes the size of the signal.

---

[4] We recall that the notation $\mathcal{N}(x, \sigma^2)$ means normal law with means $x$ and standard deviation $\sigma^2$.

We denote as $\{\lambda_\mu\}$ the set of eigenvalues of $Q$. The "unspiked" model $\beta = 0$ corresponds to a white noise, and for $N \to \infty$, the empirical eigenvalue distribution $\mu_E(\lambda)$ ($\lambda \in \mathbb{R}$) defined as:

$$\mu_E(\lambda) = \frac{1}{N} \sum_\mu \delta(\lambda - \lambda_\mu), \tag{2}$$

converges weakly in statistic toward the Wigner semi-circle law $\mu_W(\lambda)$:

$$\mu_W(\lambda) = \frac{1}{2\pi} \sqrt{4 - \lambda^2} \mathbf{1}_{[-2,2]}, \tag{3}$$

where $\mathbf{1}_{[-2,2]}$ is the windows distribution, equals to 1 in the interval $\lambda \in [-2, 2]$ and to 0 otherwise. In general, for $\beta \neq 0$, we have the following theorem [66, 68]:

**Theorem 1** *Let $Q = \beta u u^T + M/\sqrt{N}$ a spiked Wigner matrix with $u^2 = 1$ and $M \in GOE$. We have:*

- *For $\beta \leq 1$, the largest eigenvalue of $Q$ converge almost surely toward 2 as $N \to \infty$ with Tracy-Widom distribution of order $N^{-2/3}$.*

- *For $\beta > 1$, the larger eigenvalue converge almost surely toward $\beta + 1/\beta > 2$ as $N \to \infty$, accordingly to a Gaussian error function of order $N^{-1/2}$.*

In the point of view of signal detection, this result means that:

1. As soon as $\beta > 1$, the signal can be easily detected and recovered using standard iterative methods and PCA is optimal.

2. For $\beta < 1$ however, signal detection and recovering are almost impossible in practice, except under some assumptions about the prior but in that case PCA is never optimal.

3. For the critical value $\beta = 1$ consistency tests exist to distinguish the spike, see [69].

Despite the fact that we focused on the Gaussian ensemble, the previous statement hold for many random symmetric matrices as the size of the matrix approach infinity, provided that entries are i.i.d random variables with zero mean and finite variance. This leads to the real symmetric Wigner ensemble **Wig**(O($N$)). More precisely: [66, 67]:

**Definition 1** *The real symmetric Wigner ensemble **Wig**(O($N$)) is a non-vanishing set of statistical models for real and symmetric random matrix, and $M \in \textbf{Wig}(O(N))$ if and only if:*

- *Entries $M_{ij}$ are i.i.d randomly distributed with zero mean ($\mathbb{E}(M_{ij}) = 0$).*

- *Standard deviation for $M_{12}$ equals 1 ($\mathbb{E}(M_{12}^2) = 1$).*

- *Momenta of the distribution are all bounded ($\mathbb{E}(\mathrm{Tr}M^k) < \infty$),*

*where $\mathbb{E}$ denotes the standard statistical averaging.*

In this respect, Wigner law is a consequence of the universality of large size random matrices as the central limit theorem is for scalar probabilities. Let us enunciate the complete statement [66, 70–72]:

**Theorem 2** *Let $Y = M/\sqrt{N}$ with $M \in \textbf{Wig}(\mathcal{O}(N))$ a random symmetric $N \times N$ matrix. Then, as $N \to \infty$, the empirical distribution $\mu_E$ for $Y$'s eigenvalues spectrum converge weakly in statistic toward the Wigner semicircle law $\mu_W$.*

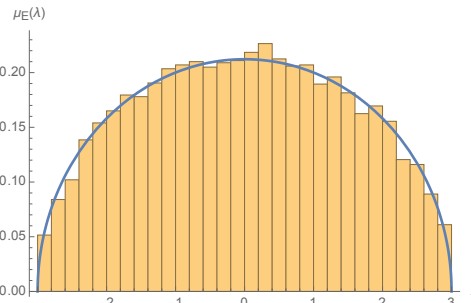
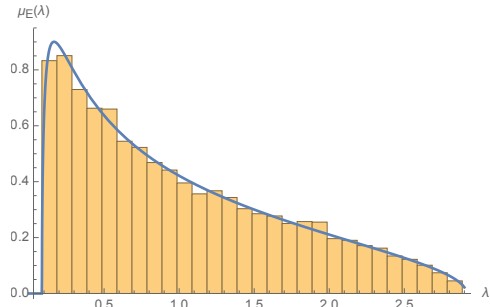

Figure 2: Illustration of the convergence toward universal laws. On both sides we show eigenvalues histograms for Wigner (on the left) and white Wishart (on the right) matrices of size $10^4$. The blue lines materialize the limit Wigner semi-circle ($\mu_W$) and MP ($\mu_{MP}$) laws.

The Wigner semicircle law is an universal feature of a matrix ensemble which is (almost) independent of the measure. An another well know universal law for random matrix will be discuss in the next section: the Marchenko–Pastur (MP) law. It describes the convergence of the density spectrum in the white *Wishart orthogonal ensemble* (i.i.d **WOE**) defined as follows:

**Definition 2** *The white **WOE** is the set of statistical models for positive definite $N \times N$ matrices of the form:*

$$Y = \frac{1}{P}XX^T, \qquad (4)$$

*and $Y \in$ **WOE** if and only if $X$ is a $N \times P$ matrix having i.i.d distributed reals entries[5] (higher momenta being assumed finite as well).*

The MP theorem state that:

**Theorem 3** *In the limit $P, N \rightarrow \infty$, keeping the ration $N/P = \alpha \geq 1$ fixed, the empirical distribution $\mu_E$ converge in statistic toward the almost surely MP distribution $\mu_{MP}$, with support between $[\lambda_-, \lambda_+]$:*

$$\mu_{MP}(\lambda) = \frac{1}{2\pi\sigma^2}\frac{\sqrt{(\lambda - \lambda_-)(\lambda_+ - \lambda)}}{\lambda\alpha}\mathbf{1}_{[\lambda_-,\lambda_+]}, \qquad (5)$$

*where $\lambda_\pm = (1 \pm \sqrt{\alpha})^2$.*

Figure 2 illustrates the convergence toward the Wigner and MP laws. Theorem 1 generalizes for other universality class, and for $N \rightarrow \infty$ the position $\lambda_{\text{out}}$ of the *outlier* in the density spectrum is given by:

$$\lambda_{\text{out}} = \mathfrak{g}^{-1}(1/\beta), \qquad (6)$$

where $\mathfrak{g}^{-1}$ denotes the inverse of the *Stieltjes transform* $\mathfrak{g}$[6] [67]. For low-rank perturbation moreover, the law generalizes again as:

$$\lambda_{\text{out}} = \mathfrak{g}^{-1}(1/\beta_k), \qquad (7)$$

where $\beta_k$ denotes the strength of the $k$-th deterministic perturbation – the detection level being for $\beta_k > \beta_k^* := \mathfrak{g}(\lambda_+)$, where $\lambda_+$ is the surely largest eigenvalue in the spectra. The existence

---

[5]Non-Wishart matrices can be defined more generally as [67]: $\mathbb{E}(X_{ij}X_{kl}) = C_{ik}\delta_{jl}$, for some covariance matrix $C$ with zero means and finite variance $\sigma^2$.

[6]The Stieltjes (or Cauchy) transform of a $N \times N$ matrix $A$ is defined as $1/N$ times the trace of the matrix resolvent: $G_A(z) = (z\text{Id} - A)^{-1}$.

of universality theorems explains why mathematical methods of random matrix theory are powerful in practice for PCA. Indeed, universality means that statistical features of datasets do not depend on the specificity of distributions but on the finiteness of momenta. With this respect, it should come as no surprise to observe that universal distributions fit almost perfectly empirical noisy spectra in many practical situations involving datasets with large dimensions. Real data are by nature noisy and components of eigenvectors of universal distributions are *delocalized*, with maximum entropy density,[7] hence they contain no information and provide the best mathematical incarnation of what is noise.

In counterpart and as it is the case for the one-spike matrix model discussed above, information are materialized with *localized eigenvectors*.[8] Universal distributions provide thus a good mathematical incarnation for purely noisy data.

Historically, this universal behavior has been stressed in nuclear physics with the works of Wigner, Dyson, Gaudin, Mehta and others [71]. These pioneer studies have shown the universality of the *Wigner surmise* for a large number of an interacting particle following partially unknown laws. Since, universality has been observed in almost all the domains of science, from nuclear physics to biology, chemistry or economy.

To conclude this survey, we would like in particular to comment on *sparsity*. For $N \to \infty$ we stated that the largest eigenvalue converge almost surely toward a value $\lambda_+$, equals to 2 in **Wig** and $(1 + \sqrt{\alpha})^2$ for MP. For universal distributions which behave asymptotically as $\mu(\lambda) \sim (\lambda_+ - \lambda)^\delta$ for $N$ large enough, and for reasons that we will further develop in the next section, we introduce the following definition:

**Definition 3** *For an universal distribution $\mu(\lambda)$ behaving as a power law $\mu(\lambda) \sim (\lambda_+ - \lambda)^\delta$ in the vicinity of $\lambda_+$, we call $d_0 := 2\delta + 2$ the asymptotic dimension of the distribution.*

The thickness $\delta\lambda := |\lambda_{\max} - \lambda_+|$ between the largest eigenvalue and $\lambda_+$ can be estimated by the observation that $\mu(\lambda_{\max})\delta\lambda$ must be equals to $1/N$, the minimal separation from which we can distinguish two eigenvalues [67]. This leads to:

$$\delta\lambda \sim N^{-\frac{2}{d_0}} . \tag{8}$$

For the Wigner and the MP distributions, it is easy to check that critical dimensions are equals $\delta = 1/2$, $d_0 = 3$ and $\delta\lambda \sim N^{-2/3}$. Note that $d_0 = 3$ is actually a general feature for convex potentials except at the critical points, see [67,74].

## 2.2 The nearly continuous spectra issue

Standard PCA tools work well for spectra involving one or a few discrete spikes. In such a situation, a very small number of eigenvalues capture a large fraction of the total variance materialized by a gap in eigenvalues, for some $K = \Lambda$ in the fraction:

$$\zeta(K) := \frac{\sum_{\mu=0}^{K} \lambda_\mu}{\operatorname{Tr} C} . \tag{9}$$

---

[7]For the Gaussian ensembles, eigenvectors are completely delocalized means that all their entries can not be greater than the typical size $\sim N^{-1/2}$. The condition is moreover extremely sharp in probabilities, and for each element, $u_i$ of the eigenbasis $\mathcal{B} = \{u_1, \cdots, u_n\}$, the infinite norm follows the probability laws:

$$P\left(\|u_k\|_\infty \geq \frac{N^\alpha}{\sqrt{N}}\right) \leq N^{-D},$$

for some $D, \alpha > 0$. See [73].

[8]i.e. almost deterministic, having a Gaussian distribution with a small width (of order $\sqrt{N}$ in the Wigner ensemble).

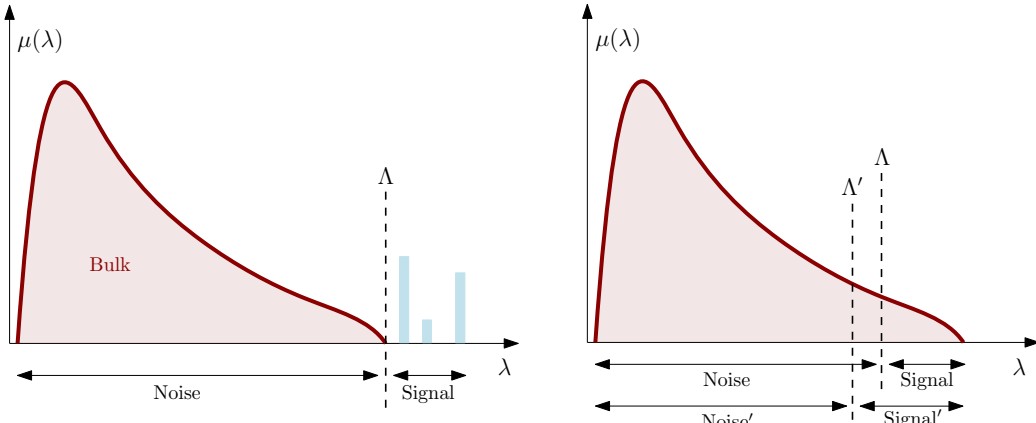

Figure 3: *On the left*: Typical empirical spectrum exhibiting some localised spikes out of a bulk (in red) made of delocalized eigenvectors. The cut-off $K = \Lambda$ provides a clean separation between delocalized eigenvectors (noise) and localized ones (information). *On the right*: Arbitrariness in the choice of the cut-off $\Lambda$ in nearly continous spectra.

In that equation $C$ denotes the *covariance matrix* and $\{\lambda_\mu\}$ the set of its eigenvalues. For concreteness, we focus on datasets displaying as a $N \times P$ matrix $X = \{X_{ai}\}$, where indices $a$ and $i$ runs respectively along the sets $\{1, \cdots, P\}$ and $\{1, \cdots, N\}$. Assuming the matrix $X$ suitably mean-shifted, we define the covariance matrix $C$ as in the Wishart ensemble as the $N \times N$ matrix

$$C = \frac{X^T X}{P}, \tag{10}$$

where $T$ means standard transposition. Note that for datasets having high variances, to avoid that variable with large variance dominate the PCA, it is suitable to work with the reduced matrix:

$$\tilde{C}_{ij} = \frac{C_{ij}}{\sqrt{C_{ii} C_{jj}}}, \tag{11}$$

called *correlation matrix* [36]. Figure 3 (on the left) illustrates qualitatively the situation where some discrete spikes capture a large fraction of the covariance matrix and dimensional reduction provided by PCA works well.

In this paper we are wondered mainly about the opposite situation - meaning where the number of spikes is large enough and the spectrum almost continuous. In practice, this happens as soon as a large number of relevant features display in restricted windows of eigenvalue. In that setting, the gap for $\zeta(K)$ vanishes and standard PCA fails to provide a clean separation between noisy degrees of freedom and relevant ones. It is important at this stage to stress that this situation is far from be marginal: covariance matrix with eigenvalues spectra almost continuous [29,75–78] are actually much more common. Note that noisy and relevant degrees of freedom fail to decouple for nearly continuous spectra is reminiscent of the non-decoupling of physical scales occurring for instance in statistical mechanics of critical phenomena for magnetic systems. We will discuss further this analogy at later stage.

Regarding the problem of signal detection, standard algorithms, based for instance on the Python package "randomly", exploit the universal features of random matrices (MP and Tracy-Widom distribution for the qualitative illustration in Figure 4), to distinguish noisy degrees of freedom from others. Such a method works in practice but has some disadvantages, among them:

1. It requires in principle a well quantitative understanding of different kinds of noise.

2. It has to be able to deal with the sparsity of the data, which is especially relevant for nearly continuous spectra, the strip of length $N^{-\frac{1}{1+\delta}}$ having a non-vanishing entanglement with relevant eigendirections.

3. Related in part to the above concerns, relevant eigenvectors are strongly mixed with delocalized eigenvectors of the bulk.

Figure on the left illustrates the third point qualitatively. On the right we provided a concrete example, adding large rank $L$ deterministic matrix to a purely Gaussian $N \times N$ Wishart matrix $M$:

$$Q = M + \sum_{k}^{L} \beta_k u_k u_k^T, \tag{12}$$

the strengths $\beta_k$ of the deterministic perturbations $|u_k| = 1$ being adjusted to display relevant features almost continuously from the surely maximal eigenvalue $\lambda_+$ of $M$ in the large $N$ limit, encouraging the non-decoupling between different scales of the spectrum. This implies a strong mixing between eigenvectors of $M$ and the deterministic vectors $u_k$, having for a consequence to dilute information between a very large number of components [79, 80].

Embedded within the entanglement, even if around the value $\lambda_+$ predicted by the most appropriate universal noise model (the MP law in the Figure) is cut, we cannot argue that, in general, what is on the left is noise and what is on the right is information. Regarding the signal detection and recovering, the difficulty can be traced from the intrinsic computational hardness of finding optimal *k-means clustering* (the simple planar *k*-means problem being, for instance, NP-Hard, see [81]). This is in substance the origin of arbitrariness illustrated in Figure 3.

As pointed out in the introduction, in physics, the goal of RG is to analyses large scale regularities by ignoring microscopic details exactly such as PCA and clustering aim to do. The combination of these approaches can be done at the formal level. For instance, the Kadanoff block-spin approach above-mentioned, and more broadly the coarse-graining underlying the Wilson approach, is nothing but a kind of clustering cleverly organized following a hierarchy inherited from the existence of an intrinsic notion of scale. For this reason, it can be expected that RG could be a powerful method to extract relevant features of an almost continuous dataset, in complement to standard methods, and this is besides exactly what RG does in ordinary field theory.

Fields theories involves generally continuous spectra for particles. RG shows relevant matters and interactions regarding the shape of such spectra. As a concrete example: the standard euclidean scalar field theory in $\mathbb{R}^d$, which describe equilibrium fluctuations of the real scalar field $\phi : \mathbb{R}^d \rightarrow \mathbb{R}$ through the Gibbs state:

$$p[\Phi] \propto e^{-S[\Phi]}, \tag{13}$$

for some field configuration $\Phi = \{\phi(x)\}$, with classical Ginzburg-Landau [Zinn-Justin] action:

$$S[\Phi] := \frac{1}{2} \int_{\mathbb{R}^d} d^d x \, \phi(x)(-\Delta + m^2)\phi(x) + \frac{g}{4!} \int_{\mathbb{R}^d} d^d x \, \phi^4(x), \tag{14}$$

for some coupling constant $g$ for the quartic term, $m^2$ is the mass of the free particles and $\Delta$ denotes the standard Laplacian $\Delta = \sum_{i=1}^{d} \partial^2 / \partial x_i^2$. In the vicinity of the Gaussian line[9] where $g$ is almost zero, it is well known that the relevance of the coupling constant for the coarse-graining depends on the dimension of the background space $d$ and more precisely of the dimension's sign of the coupling constant [82]. For the model described by (14), the dimension of $g$ is $[g] = d - 4$, the notation $[A]$ being defined such that $[\Delta] = -2$. For $d > 4$,

---

[9]The Gaussian line is parametrized by the value of the mass $m^2$, which is always relevant.

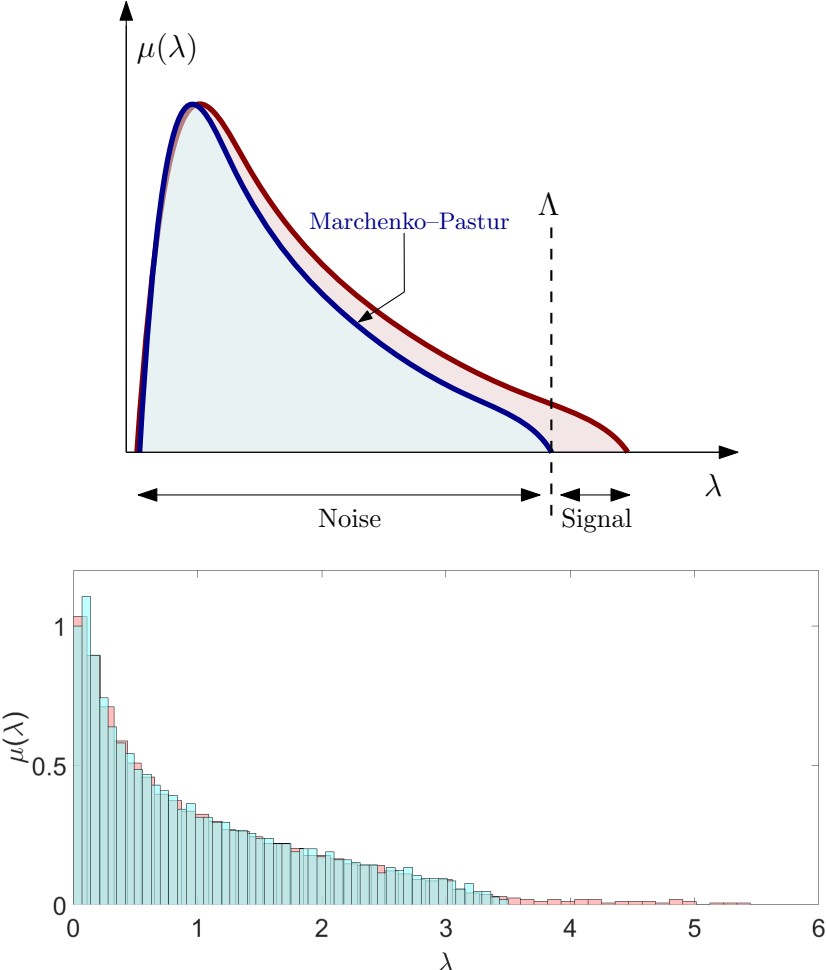

Figure 4: Covariance matrix for nearly continuous datasets. *On the top* Qualitative illustration of the deviations from the universal MP law (in blue), obtained by completely randomize the data matrix. *On the bottom:* Illustration for a spectrum obtained by adding large rank deterministic matrix to a purely Gaussian Wishart noise.

the quartic interaction is irrelevant, meaning that for sufficiently large scale, the theory is essentially Gaussian. In contrast, for $d < 4$, the interaction increases, and the RG flow is repealed from the Gaussian line $g = 0$. This simple example highlights the connection between space dimension and the relevance of operators. However, in the computation of the integrals for rotationally invariant models, the dimension of space appears in practice essentially through the momentum distribution $\rho(|\vec{p}\,|) = (\vec{p}^{\,2})^{\frac{d}{2}-1}$, or more precisely through the spectral distribution of the kinetic operator $\hat{H}_0 := -\Delta + m^2$ whose eigenvalues are $E^2 = \vec{p}^{\,2} + m^2$ and the density spectrum $\tilde{\rho}(E^2)$ is:

$$\tilde{\rho}(E^2) = \frac{\rho(\sqrt{E^2 - m^2})}{2\sqrt{E^2 - m^2}}\,. \tag{15}$$

In terms of the original distribution $p[\Phi]$ this corresponds to a coarse-graining, and the coarse grained field:

$$\psi(E) := \int d^d p\, \delta(E - \sqrt{\vec{p}^{\,2} + m^2})\phi(\vec{p}\,)\,, \tag{16}$$

which induce an effective distribution $\tilde{p}[\psi]$:

$$\tilde{p}[\psi] \propto \int \prod_{\vec{p}} [d\phi(\vec{p})] \delta\left(\psi(E) - \int d^d p\, \delta(E - \sqrt{\vec{p}^{\,2} + m^2})\phi(\vec{p})\right) e^{-S[\Phi]}, \qquad (17)$$

where $\prod_{\vec{p}}[d\phi(\vec{p})]$ denotes the path integral measure.

Figure 5 (on the left) shows the typical shape of the distribution $\tilde{\rho}$ for $d = 5$ and $d = 3$. On the right side of the figure we show the same region of the spectrum for degrees of freedom labelled with the eigenvalues of the free propagator $\hat{H}_0^{-1}$, corresponding to the density $\tilde{\mu}(1/E^2) := \tilde{\rho}(1/E^2)/(E^2)^2$. The figure have to be compared with Figures 3 and 4. Hence, one can almost say that, regarding the power counting, the intrinsic ability of couplings to survive along the RG flow depends on the shape of the distribution (which depends on $d$) and indirectly on the background space dimension. In other words, the shape of the distribution $\tilde{\rho}$ – i.e. the (free) energy density spectrum of particles filling the space $\mathbb{R}^d$ – decides essentially if interactions are relevant or not in the vicinity of the Gaussian region. Hence, varying the dimension, the shape of the distribution change, as well as the large scale description of the considered field.[10]

From that standpoint, ordinary field theory provides an explicit example of a situation where RG may extract relevant features for degrees of freedom associated with a continuous spectrum. One could even go further by imagining a situation where the effective dimension of the background space would depend on the energy scale, and the shape of the spectrum would thus change along with the flow. This situation is quite similar to the one of the continuous spectra that we mentioned above in Figure 3 and 4, which are not power laws globally and stress the point of view adopted in this paper.

The previous observation suggests a correspondence with the signal detection issue for nearly continuous spectra and relevance of RG trajectories for a field theory. Indeed, one could consider an effective field, describing some matter filling an abstract space and whose interacting particle density spectrum would be given by the density $\mu(\lambda)$; this density playing the same role as $\tilde{\mu}(1/E^2)$ for the nearly Gaussian example (14)).[11]

This field theory would provide an effective description of the underlying degrees of freedom, whose dataset describe correlations. Degrees of freedom can be very different for two different sets of data: they may describe - but not limited to - either interactions between genes in a cell, neurons in a brain fragment or asset prices for financial markets. Once again, it is a reminiscent of what happens in physics. Ising model, magnetic systems and lattice gas provide elementary examples by describing the interaction between very different kinds of degrees of freedom, the Ising spin being very far from the true description of atomic forces in a magnet or molecular forces in a braid of DNA [83]. These models have similar properties for long-range scales and are also effectively described by field theory (14) for large scales [3]. One may expect the same kind of universality for the field theory that we aim to consider and, despite the strong difference between degrees of freedom of different datasets, that they can obey to the same effective description through a field theory. The universality of the field theoretical description is therefore essential to validate the approach. Indeed, from this approach the question is not about the research for information that can be find in the spectrum but rather focusing on identifying the nature of interactions and the kind of effective physics that can be construct on such a spectrum. If these two questions seem very different, the scope of the second is however universal.

---

[10]This argument assume that the form of the interactions is fixed, as well as the underlying locality principle used to construct them.

[11]Note that we do not associate $\mu(\lambda)$ with the free spectrum but rather with the *interacting spectrum* because we expect that the covariance matrix describes non-Gaussian correlations.

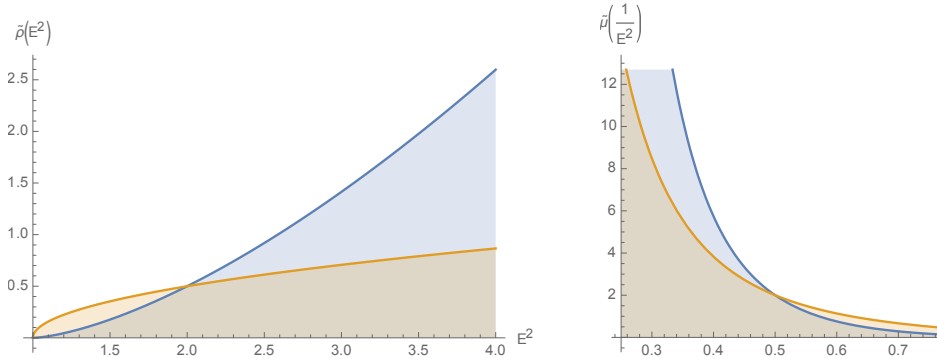

Figure 5: Eigenvalue density spectrum for a scalar field theory. *On right:* Energy density spectrum $\tilde{\rho}(E^2)$ for free particles with mass $m^2 = 1$ in dimension $d = 5$ (in blue) and $d = 3$ (in orange). *On left:* Eigenvalue density spectrum $\tilde{\mu}(1/E^2)$ of the free propagator $\hat{H}_0^{-1}$ for $d = 5$ (in blue) and $d = 3$ (in orange).

Such a field theory has been introduced in [29, 52], where the authors focused on local interactions in the momentum space. With regard to the continuity of these aforementioned articles, this paper aims to provide another derivation highlighting the role of the non-local contributions in sections 3. Moreover, we provide extended numerical investigations and application in sections 4 and 5.

## 2.3 Working methodology

This paper is organized as follows:

- The section 3 gives arguments in favour of an effective description by a field theory exploiting the universality of noise models and the simplest of statistical inference methods: the maximum entropy estimate.

- The section 4 in turn presents the basis of the non-perturbative RG formalism. In particular, a suitable definition of "dimensionless flow" will be introduced through a generalization of the notion of canonical dimension, taking into account an intrinsic scale dependence.

- Finally, the section 5 gives the result of detailed investigations around different common noise models, underlining therefore the universal value of the empirical proposition 1. We will also show that in the field of tensorial PCA, the RG approach allows justifying the efficiency of some tensorial invariants over others, giving thus an alternative point of view to the recent work [84].

## 3 Analogue field theory for spectral analysis

Analogue models in physics allow to simulate a large number of phenomena, closely relating to real problems for which experiments could be difficult to carry out. A famous example is provided by general relativity, for which analogous models of condensed matter physics have made it possible to simulate the behavior of extreme objects such as black holes [85]. Conversely, these analogue models have been a fruitful source of inspiration for still open issues, such as that of quantum gravity for instance. [86, 87]. We propose here an analogous model of field theory for data science, knowing that the choice of such a model is not unique. Such as it is with analogous gravity, we expect that it will depend on the phenomena we want

to simulate and the precision of the experiments. In this article, and unless explicitly stated otherwise, the reader should keep in mind that we are more interested in the "threshold" of detection of a signal, where it is most likely to be found, it is, at the level of the "tail" of the spectrum.

We will begin by discussing a simple but instructive example of "binary" data materialized mathematically by spins arranged on an arbitrary network. This kind of formalism was used recently by Schneidman, Berry, Segev and Bialek as a statistical description of neurons activity in the brain [75–77, 88], where authors considered a coarse-graining approach. Even though we focus on a different formalism and philosophy, their results back up our approach. For such a simple model, an effective field theory can be easily derived through a maximum entropy estimator [15, 20, 21] fixing the shape of the probability distributions as a generalized Gibbs state from statistical inference, based on the knowledge of partial information (2-point correlations) on the system.

In the following of this section, we generalize the construction and propose an efficient model able to reproduce data correlations in the experiment described in the next section 5. As an important insight, let us mention that the resulting effective field theory exhibits different behavior in the tail of the spectrum, where it seems legitimates to look at interactions like local monomials, as in the heart of the bulk, where a specific non-locality appears as a consequence of the non-locality of eigenvectors. Our goal is to understand, from this formalism of field theory, how the shape of the distribution influences the tendency of certain operators to strengthen or on the contrary to disappear depending on the experimental scale and how this can be related to the detectability of a signal. We insist on the close relationship this theoretical construction has with experience, and that a complete understanding of the material of the present section also requires reading the experimental section 5.

## 3.1 Maximum entropy estimator for spins

Now assume an abstract network $\mathcal{G}$ looking as a connected graph with $N$ nodes. To each node $i$ of the network that we call site, is attached a binary variables $s_i = \pm 1$ (see Figure 6). We denote as $\sigma := \{s_1, s_2, \cdots, s_N\}$ a typical state of the system (i.e. a typical configuration of spins). The number of states is $\Omega = 2^N$ which increases rapidly with the number of sites $N$.[12] An ideal random network should explore freely all these configurations, but we conjecture it is not the case by assuming the existence of a global organization materialized by a probability distribution $p(\sigma)$ over the $\Omega$ spin configurations $\sigma$. Therefore, global Gibbs entropy is bounded by the entropy of the ideal random network:

$$S_G := -\sum_{\sigma} p(\sigma) \ln(p(\sigma)) \leq N \ln(2), \tag{18}$$

the subscript "G" being for "Gibbs".[13] The network must be large enough to use methods of statistical mechanics and, in particular, we assume the number of a required experiment to fix all the parameters in probability distribution $p(\sigma)$ is too large to be considered. The best compromise we can do in such a way is to estimate the probability distribution about a few experiments, by laying down for instance several correlations. The underlying form of statistical inference is exactly what is supported by standard statistical mechanics, as Jaynes stressed in its works [20, 21], viewing the statistical mechanics through the filter of information theory as an inference problem based on the *maximum entropy estimate* (MEE).[14]

---

[12]Eddington number estimate the number of atoms in the Universe as $\sim 10^{80}$. Then, the number of states for a network having $N \sim 270$ sites is larger than the number of atoms in the Universe.

[13]It would be more appropriate to speak of Shannon entropy in this context, where information theory plays an important role, but we have chosen to refer to Gibbs, more familiar to physicists.

[14]Note that the maximum entropy estimate minimizes the relative entropy (or Kullback-Leibler divergence) with the uniform distribution in the bound (18).

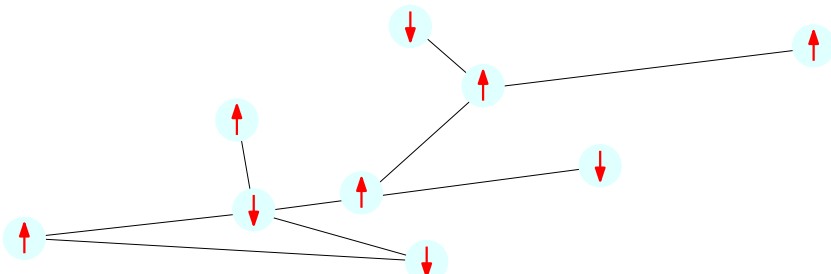

Figure 6: A typical abstract network decorated with spins $s_i$ to each nodes.

Concerning the missing information due to our partial knowledge (materialized by the probability), the maximum entropy distribution is the less structured as possible and the less constrained one, with some experiences able to falsify its efficiency.

Consequently and applied to our model, we can draw the following developments. It will be assumed that we have only partial knowledge about the statistics of the distribution, as it is the case in many practical situations. For our purpose we assume to know the first and second momenta, namely the average spins $\mu = \{\mu_i\}$ and the standard deviation matrix (covariance matrix) $C = \{C_{ij}\}$:

$$\sum_{\sigma \in \Omega} p(\sigma) s_i = \mu_i, \qquad \sum_{\sigma \in \Omega} p(\sigma)(s_i - \mu_i)(s_j - \mu_j) = C_{ij}. \tag{19}$$

The covariance matrix $C$ materializes mathematically the correlations between spins $i$ and $j$. However, it is a simple exercise to show that the distribution $\bar{p}(\sigma)$ which maximizes $S_G$ under constraints (19) takes the form:

$$\bar{p}(\sigma) = \frac{e^{-H(\sigma)}}{Z}, \tag{20}$$

for some kind of Hamiltonian given by:

$$H(\sigma) := -\frac{1}{2} \sum_{i,j}^{N} s_i K_{ij} s_j + \sum_{i=1}^{N} h_i s_i, \tag{21}$$

and the partition function $Z$ reads as:

$$Z := \sum_{\sigma \in \Omega} e^{-H(\sigma)}. \tag{22}$$

The (ill-posed) inverse problem to find the pair $\{J_{ij}, h_i\}$ which solve equations (19) is a very hard task [23, 24]. In practice, they can be estimated from standard Monte-Carlo simulations. One difficulty comes from the observation that experiments have finite precision; for this reason, the solution cannot be unique. On the contrary, several solutions might be able to simulate the 2-point correlations between spins at a given level of precision, and cannot be distinguished.

As pointed out in the last paragraph, such a maximum entropy model has been considered recently in a series of papers [75–77] to describe the electric activity of neurons in the brain. Experimentally, the correlation function between neurons is estimated by constructing time sample $\delta x_i(t)$ for neuron (i) (suitably mean-shifted), for a discrete-time $t$. The authors considered a specific coarse-graining to extract relevant features of the distributions, looking as a block-spin partial integration. More precisely, they construct coarse-grained distributions $\bar{p}(\tilde{\sigma})$, by replacing the original variables $s_i$ with:

$$s_i \to \tilde{s}_i \propto \sum_{j=1}^{N} \left( \sum_{\mu=1}^{\Lambda} u_i^{(\mu)} u_j^{(\mu)} \right) s_j, \tag{23}$$

where $u_i^{(\mu)}$ is the i-th component of the (normalized) eigenvector associated to the eigenvalue $\lambda_\mu$. The effective description $\bar{p}(\tilde{\sigma})$ is then obtained by averaging over spin configurations $\sigma$ keeping $\tilde{\sigma} = \{\tilde{s}_i\}$ fixed.[15] Interestingly, the authors showed that probability distributions of coarse-grained variables undergo a non-Gaussian form, suggesting that collective behavior of neurons is well described by a non-Gaussian fixed point for correlation spectrum exhibiting power-law dependencies for large eigenvalues. Note that similar observations have been done for $2D$ Ising model on a rectangular lattice [89,90], making numerical simulations using Metropolis algorithm as time-evolution. Their results show that the spectral density changes shape in the vicinity of the critical temperature, behaving like a power law, whereas it agrees with the predictions of the theory of random matrices for high temperature (the MP law).

The efficiency of such a coarse-grained description is not a surprise, RG aiming to describe emergent physics from effective models, ignoring the underlying microscopic interactions. A moment of reflection suggests that, even though this model has been derived in a specific context, it must be largely independent of the nature of data involved, at least on a large-scale. This is due to the existence of universal laws, which blinds spectres to the specificity's of data.

The correlation spectrum for the $2D$ Ising model at high temperature provides an elementary example. The characteristics of this spectrum have nothing to do themselves with the Ising model. From this observation and for a particular type of data, it should be imagine that we have constructed a probability distribution model able to decide whether a signal is present in the corresponding spectrum, assumed to be in the vicinity of a universal noise like MP. This could be achieved for example by investigating what kind of correlations survive along the RG flow.[16] Then, the universality of the spectra implies that this model will also be able to detect the presence of a signal for data of any other nature, as long as the corresponding spectrum remains in the vicinity of the same universal noise. However, it transpires then a difficulty from this interpretation.

For the binary model for instance, if we consider that the predictions of the model have a universal scope it follows that the binary variables $s_i$ are directly attached to the starting problem. To be useful, such a model have to be embedded in a framework that represents a good limit (from the RG point of view) for a large variety of models – see Figure 7. In physics, a large number of discrete models seem to be well described by a field theory within a suitable limit, and we will therefore seek to build such an effective model.

This observation is the motivation underlying the series of papers [52–55], whose statement 1 summarizes the main conclusions. The links between field theory and the Ising model can be formally constructed. We recall here a classical derivation which will prove to be instructive in the following. Using the standard Hubbard-Stratonovich transformation:

$$e^{-H(\sigma)} \propto \int_{-\infty}^{+\infty} \left[ \prod_{i=1}^{N} d\phi_i \right] e^{-\frac{1}{2}\sum_{i,j}^{N} \phi_i K_{ij}^{-1} \phi_j} \prod_{j=1}^{N} e^{(-h_j+\phi_j)s_j}, \tag{24}$$

where we assumed that $K^{-1}$ exist, i.e. $\det K \neq 0$ and where we introduced the field $\Phi = \{\phi_1, \cdots, \phi_N\}$. The sum over the spin configurations can be performed $\sum_\sigma e^{\sum_j(-h_j+\phi_j)s_j} = \prod_j 2\cosh(\phi_j - h_j)$. This leads to an effective model, describing random configurations for $\Phi$. For a centered distribution we have to set $h_i = 0$, and the corresponding probability density $P(\Phi)$ reads as

$$P(\Phi) = \frac{1}{Z} \exp\left( -\frac{1}{2} \sum_{i,j}^{N} \phi_i (K_{ij}^{-1} - \delta_{ij})\phi_j - \frac{1}{12} \sum_{i=1}^{N} \phi_i^4 + \mathcal{O}(\phi_i^6) \right), \tag{25}$$

---

[15]Using a step function to impose the constraint ensure that $\tilde{\sigma}$ variables remain spins.

[16]That will be our point of view in the rest of this paper, see the next subsection.

where:

$$Z = \int_{-\infty}^{+\infty} \left[ \prod_{i=1}^{N} d\phi_i \right] \exp\left( -\frac{1}{2} \sum_{i,j}^{N} \phi_i K_{ij}^{-1} \phi_j + \sum_{i=1}^{N} \ln(2\cosh(\phi_i)) \right). \tag{26}$$

In terms of these new variables $\Phi$, the correlation functions between spins read as:

$$C_{ij} = \langle \tanh(\phi_i) \tanh(\phi_j) \rangle. \tag{27}$$

We insist again that this formal construction, which one finds in several standard textbooks [91–93] to justify the passage to a field theory for the Ising model, rests on the fact that the inverse $K^{-1}$ exist. For the Ising model, this manipulation is only justified on a large scale (in the vicinity of the transition), when $K(\vec{p}) \propto \sum_{\ell=1}^{d} \cos(p_\ell)$ (the Fourier transform of $K$) can be expanded to a power of $p$: $K(\vec{p}) \sim (2d + \vec{p}^{\,2}) + \mathcal{O}(p_\ell^4)$, which is invertible. The derivation does not need to be rigorously made. Indeed, in regard to RG universality, essential features for the effective description we are looking for are mainly the form and structure of the interactions.

## 3.2 Analogue effective field theory candidates

In this section, we construct an effective field theory framework able to mimic the correlations in datasets around universal noises. Such field theory and the resulting RG are expected to be as simple as possible and able to save considerable computing time compared to usual numerical methods.[17] The theory of fields that we propose here is probably far from being the only one possible nor even from being optimal from the point of view of what such a formalism could allow learning, however our aim here is to show what such a formalism can bring, without worrying about whether it is optimal or not. We will discuss essentially two kinds of regimes where interactions can be suitably expanded in terms of local and non-local monomials in the eigenspace of the covariance matrix. The local approximation has been largely discussed in the reference papers [52, 54], and a large part of this section is therefore devoted to a presentation of the non-local sector of the theory.

### 3.2.1 Gaussian model

We consider a dataset $\mathcal{E}$ whose correlation matrix is assumed to have a nearly continuous spectrum. We assume that, at a sufficiently large scale of description i.e. after a large number of steps in the RG, the data $\mathcal{E}$ can be approximated by a field $\Phi = \{\phi_1, \phi_2, \cdots, \phi_N\}$ with $N$ components and having 2-point function components $G_{ij} := \langle \phi_i \phi_j \rangle$ equals to the components of the correlation matrix $C$:

$$\langle \phi_i \phi_j \rangle \equiv C_{ij}, \tag{28}$$

where the notation $\langle F(\Phi) \rangle$ means averaging with respect to some probability distribution $p(\Phi)$:

$$\langle F(\Phi) \rangle := \int [d\Phi] \, p(\Phi) F(\Phi), \tag{29}$$

with $[d\Phi] := \prod_{i=1}^{N} d\phi_i$. The correlation matrix $C$ at this scale is assumed to be cut off from its most ultraviolet degrees of freedom (from these smallest eigenvalues). We recall the definition of UV and IR scales given above in the introduction:

**Definition 4** *The spectrum of the correlation matrix $C$, assumed to be nearly continuous, provides a canonical notion of scale. UV scales correspond to small eigenvalues and IR scales to large eigenvalues. Moreover the largest eigenvalue $\lambda_+$ bounding the spectra from above, define a canonical correlation length $\xi = \sqrt{\lambda_+}$.*

---

[17]This computational advantage of the RG was recently pointed out in the study of Brownian motion, [94, 95] and spin glass dynamics [34].

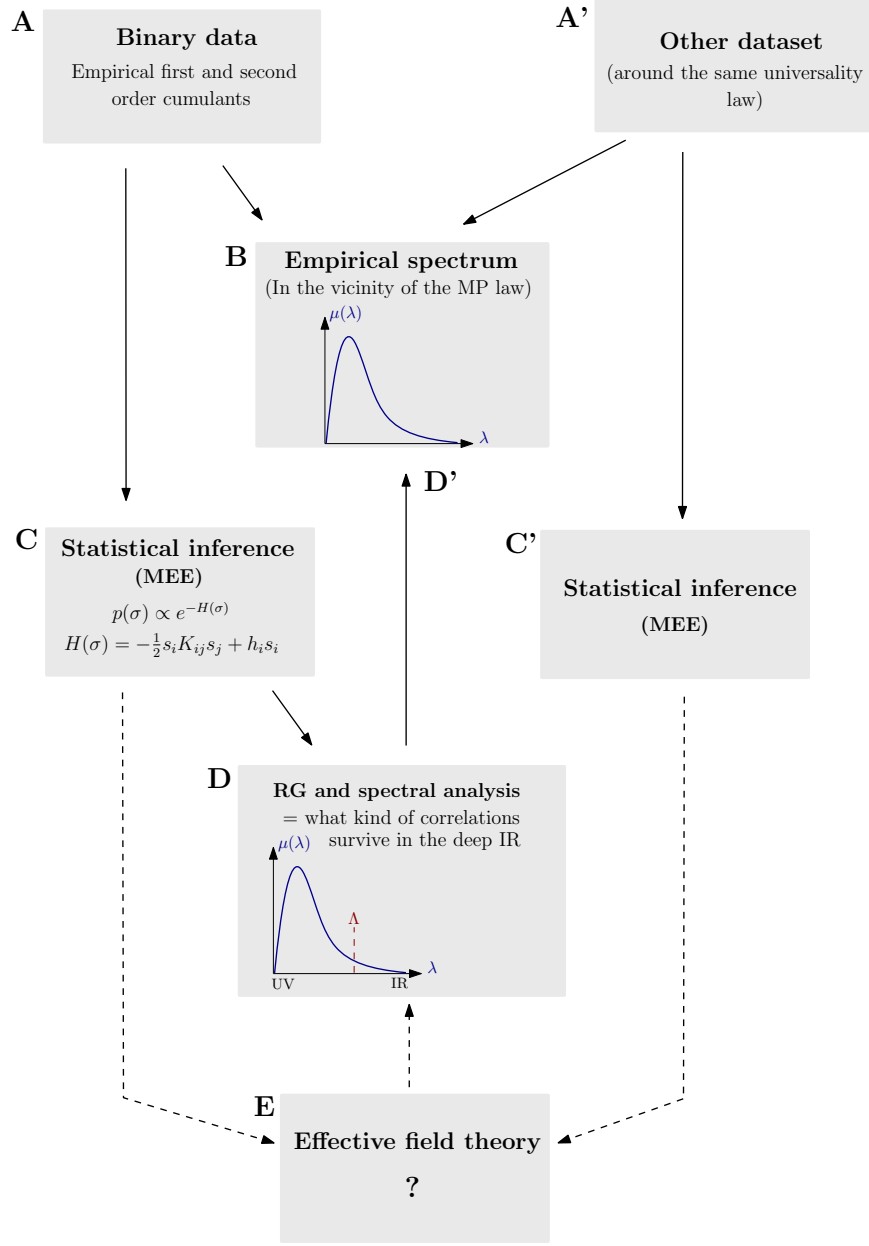

Figure 7: Steps **A**-**C** (**A'**-**C'**) describe how to construct a maximum entropy estimate for binary like data whose statistics features are close to MP law. From step **C**, we can construct an RG map that describes how couplings change with scale (defined by the spectrum itself, "ultraviolet" scales corresponding to small eigenvalues and "infrared" scales to large eigenvalues of the correlation matrix). If the relevance or irrelevance of couplings changes for MP law is disturbed with a small signal, one can establish a criterion to decide if a binary dataset blind a signal (step **D**). Because the spectrum that we consider cannot be distinguished from another spectrum close to the same universal noise, the relevance criterion must be held for datasets of different nature (**D'**). This suggests the existence of effective universal models of the "field theory" type, capturing most of the characteristics of particular maximum entropy models and avoiding any special considerations on the precise nature of the data (**E**).

Based on constraint (28), the MEE takes the form:

$$p(\Phi) = \frac{1}{Z} \exp \left( -\frac{1}{2} \sum_{i,j=1}^{N} \phi_i D_{ij} \phi_j \right), \tag{30}$$

with $Z = (2\pi)^{N/2}/\sqrt{\det D}$. MEE (30) agrees with the constraint (28) if $C_{ij} = D_{ij}^{-1}$. Note that, because we assume to work at scale large enough, one expects the matrix $C$ invertible, and the zero eigenvalues removed from the coarse-graining, as discussed at the end of the previous section 3.1. The Gaussian model (30) fix the form of higher correlations functions from Wick theorem: odd functions vanish and even functions decompose as the sum of the product of 2-point correlations function:

$$\langle \phi_i \phi_j \phi_k \phi_l \cdots \rangle = C_{ij} \times C_{kl} \times \cdots + \text{perm}, \tag{31}$$

where perm runs over all allowed pairing of indices. The matrix $C$, being assumed to be symmetric and positive. It is diagonalisable and eigenvectors $u^{(\mu)}$ can be suitably normalized:

$$\sum_{j=1}^{N} C_{ij} u_j^{(\mu)} = \lambda_\mu u_i^{(\mu)}, \qquad \sum_{i=1}^{N} u_i^{(\mu)} u_i^{(\mu')} = \delta_{\mu\mu'}. \tag{32}$$

For purely random matrices,[18] the components of the eigenvectors are uniformly distributed on the sphere $\mathcal{S}_{N-1}$. It is suitable to work in the spectral representation, and to introduce the field $\Psi$ whose components $\psi_\mu$ are defined as:

$$\psi_\mu := \sum_{i=1}^{N} \phi_i u_i^{(\mu)}. \tag{33}$$

For a purely noisy data, it is equivalent to apply a random rotation on the vector $\Phi$: $\Psi = \mathbf{O}\Phi$, for $\mathbf{O}$ being Haar distributed over the group O($N$). In that respect, the MME (30) reads as $p(\Psi) \propto e^{-H(\Psi)}$, with:

$$H(\psi) = \frac{1}{2} \sum_{\mu=1}^{N} \psi_\mu (\lambda_\mu)^{-1} \psi_\mu. \tag{34}$$

For positive defined matrices, on which we focused on this paper, $\lambda_\mu > 0$. Let $\lambda_+ = \xi^2$ the larger eigenvalue. It is convenient to write the propagator $\lambda_\mu$ as:

$$\lambda_\mu = \frac{1}{p^2 + m^2}, \tag{35}$$

with $m^2 := \xi^{-2}$. The "momentum" $p$ is positive definite $p \geq 0$, and its larger value $p = \Lambda$ define the smallest eigenvalue of the spectra:

$$\lambda_- =: \frac{1}{\Lambda^2 + m^2}. \tag{36}$$

This larger value $p = \Lambda$ plays the role of a UV cut-off in the field theory language. It defines the microscopic scale, which moves away along with RG flow. Formally, the field $\psi_\mu$ looks like the field that we called $\psi(E)$ in section 2.2, equation (16).

For this reason, it is tempting to introduce a more fundamental representation, playing the role of the states $\psi(\vec{p})$. This however poses the problem of the dimension of the space

---

[18]i.e. Purely noisy data, suitably materialized by random matrices of Wigner or Wishart kinds for instance.

$\mathbb{R}^d$ in which define such a quantity of movement $\vec{p}$ (i.e. The number of components that we have to give for the quantity $\vec{p}$). A way to define it is to observe that the asymptotically usual model of noise behaves like a power law. As noticed in definition 3 for instance, MP has asymptotic dimension $d_0 = 3$ and momenta are distributed accordingly to Fourier modes of a 3-dimensional space. However, such a way raised an another issue, because the distribution shape changes as we move toward the UV scale. Strictly speaking, it makes sense only in the deep IR. One can think, to circumvent the difficulty to understand deviations from the power-law as corrections to the derivative expansion, that can be resumed to an effective distribution $\rho(p^2)$.

However, this interpretation is not yet satisfactory. Indeed real distributions have to diverge from MP's asymptotic behavior, and the asymptotic dimension is most likely not to be an integer. One could then seek to make sense of such a non-integer dimension on the side of fractal geometry[19] [96], but everything would become unnecessarily complicated. A way to avoid this difficulty is to set $d = 1$ i.e. to understand $\vec{p}$ as a relative number $p \in \mathbb{R}$ and the corresponding field $\psi(p)$ as a non-conventional matter field filling this abstract space of dimension 1. Physically, this is equivalent to doubling the number of states, and for each eigenvalue $\lambda_\mu$ we have two solutions $(+|p_\mu|, -|p_\mu|)$. This doubling of states will facilitate the construction of interactions in the next section. For now, let us just note that the new Hamiltonian:

$$\tilde{H}(\psi) = \frac{1}{2} \sum_{p_\mu} \psi(p_\mu)(p^2 + m^2)\psi(-p_\mu), \tag{37}$$

reproduce exactly the same correlations as the initial Gaussian model (30) from Wick theorem. Within this approximation fixing the effective dimension of space to 1, we obviously lose the relation between momentum distribution and space dimension. However, the asymptotic dimension $d_0$ is not physically relevant in general. For instance in the previous section 3.1, we noticed that 2-point correlation spectrum for 2D Ising model at high temperature follows the MP law and, then, the dimension of the background field $d = 2$ does not coincides with the asymptotic dimension of the distribution $d_0 = 3$.

### 3.2.2 Nearly Gaussian distribution for noisy data

Gaussian distribution generally fails to provide a good estimate, and Wick theorem predictions break-down for usual data sets, where correlations higher than 2-points cannot be reduced as a product of 2-point ones. This failing invites to consider non-Gaussian corrections to the Hamiltonian. Indeed, as announced in the introduction (see Empirical statement 1), we will see that the Gaussian fixed point is unstable even for non-structured (i.e. purely noisy) data. However, there is no one single way to disturb a Gaussian measure.

**Effective model in the large-N limit.** Because we focus on the MME, the probability measure have to remain an exponential law, and the Hamiltonian (37) receives monomial contributions involving more than two fields. The discussion of Section 3.1 may suggest to us the path to be followed. Indeed, we showed that, for a model able to describe correlations for a purely MP law, interactions like $\phi_i^n$ emerges naturally. Thus, a first attempt could be, in replacement of (30):

$$p(\Phi) = \frac{1}{Z} \exp\left(-\frac{1}{2}\sum_{i,j=1}^N \phi_i \tilde{D}_{ij}\phi_j - \frac{g_4}{4!}\sum_i \phi_i^4 - \frac{g_6}{6!}\sum_i \phi_i^6 + \cdots\right), \tag{38}$$

where the kinetic kernel $\tilde{D}$ differs from the exact one $D := C^{-1}$ by quantum corrections:

$$C_{ij} = \tilde{D}_{ij}^{-1} + \mathcal{O}(g_4, g_6, \cdots). \tag{39}$$

---

[19]The Kock curve provides an elementary example, having dimension $d \approx 1.26$.

Each term generated by couplings $g_4$, $g_6$ and so on, are deviations from the Gaussian predictions. For couplings small enough, deviations are expected to be small and $\tilde{D}$ is close to $C^{-1}$, and the eigenbasis of the two matrices are essentially the same. In terms of the field $\Psi$ defined by equation (33) (where we assume $\{u_i^{(\mu)}\}$ is the eigenbasis of $C$), the Hamiltonian reads:

$$H[\Psi] = \frac{1}{2} \sum_{\mu=1}^{N} \psi_\mu (p_\mu^2 + m^2) \psi_\mu + \sum_{\{\mu_i\}} V_{\mu_1 \mu_2 \mu_3 \mu_4}^{(4)} \psi_{\mu_1} \psi_{\mu_2} \psi_{\mu_3} \psi_{\mu_4} + \mathcal{O}(\psi^6), \tag{40}$$

where we introduced the symbols $V_{\mu_1 \cdots \mu_{2n}}^{(2n)}$ defined as:

$$V_{\mu_1 \cdots \mu_{2n}}^{(2n)} := \sum_{i=1}^{N} u_i^{(\mu_1)} u_i^{(\mu_2)} \cdots u_i^{(\mu_{2n})}. \tag{41}$$

For purely noisy data suitably materialized by random matrices of Wigner and Wishart kinds, for instance, eigenvectors are delocalized with entries not greater than $\sim N^{-1/2}$ [73] and the corresponding rotation eigenmatrix is asymptotically Haar distributed on the group $O(N)$ for large $N$.[20] Hence for large $N$, the sum in (41) is almost zero in general.

However, moment of reflection shows that two special configurations for indices $\{\mu_1, \cdots, \mu_{2n}\}$ are relevants. First of all, when all indices are equals i.e. $\mu_1 = \mu_2 = \cdots = \mu_{2n}$. In that way, because $|u_i^\mu| \sim N^{-1/2}$,

$$V_{\mu \cdots \mu}^{(2n)} \sim \sum_i (N^{-1/2})^{2n} = N^{1-n}. \tag{43}$$

The second relevant configuration is for indices equals pairwise. Because eigenvectors are uniformly distributed on the sphere $\mathcal{S}_{N-1}$ they must be invariant by rotation (in law), and the averaging of quantities like $u_i^{(\mu)} u_j^{(\mu)}$ (for fixed $(i,j)$) reads:

$$\langle u_i^{(\mu)} u_j^{(\mu)} \rangle \approx \frac{1}{N} \sum_{\mu=1}^{N} u_i^{(\mu)} u_j^{(\mu)} = \frac{1}{N} \delta_{ij}. \tag{44}$$

Hence, setting $\mu_1 = \mu_2$, $\mu_3 = \mu_4$ and so on, we must have again:

$$V_{\mu_1 \mu_1 \cdots \mu_{2n-1} \mu_{2n-1}}^{(2n)} \sim N^{-n} \sum_i 1 = N^{1-n}. \tag{45}$$

In that way, $V_{\mu_1 \mu_2 \mu_3 \mu_4}^{(4)}$ reads as:

$$V_{\mu_1 \mu_2 \mu_3 \mu_4}^{(4)} \approx \frac{1}{N} \left( \delta_{\mu_1 \mu_2} \delta_{\mu_2 \mu_3} \delta_{\mu_3 \mu_4} + \delta_{\mu_1 \mu_2} \delta_{\mu_3 \mu_4} + \delta_{\mu_1 \mu_3} \delta_{\mu_2 \mu_4} + \delta_{\mu_1 \mu_4} \delta_{\mu_2 \mu_3} \right), \tag{46}$$

and:

$$\sum_{\{\mu_i\}} V_{\mu_1 \mu_2 \mu_3 \mu_4}^{(4)} \psi_{\mu_1} \psi_{\mu_2} \psi_{\mu_3} \psi_{\mu_4} \sim \frac{3}{N} \left( \sum_{\mu=1}^{N} \psi_\mu^2 \right)^2 + \frac{1}{N} \sum_{\mu=1}^{N} \psi_\mu^4. \tag{47}$$

The validity of approximation (46) can be numerically checked. Table in particular 1 summarizes simulations for different configurations of indices $\mu_i$. We show in particular that for the quartic vertex neglected configurations are of order $1/N^2$ whereas the leading ones are of order $1/N$.

---

[20]Without additional information the distribution of $s = u_i^{(\mu)}$ as $i$ varies can be estimated again with a maximum entropy distribution, the *Porter-Thomas* distribution:

$$p(s) = \frac{e^{-\frac{s^2}{2}}}{\sqrt{2\pi}}. \tag{42}$$

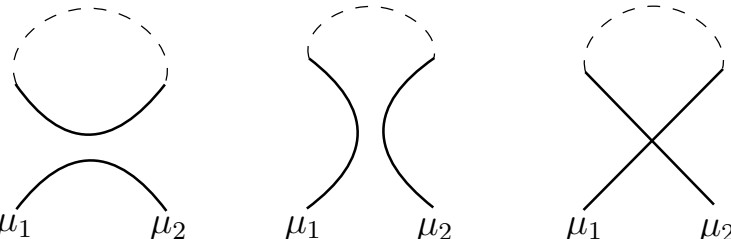

Figure 8: Feynman diagrams contributing to the one-loop self energy. The dotted edges materialize the Wick contractions with propagator $\lambda_\mu$. Solid edges materialize contractions of field indices.

Table 1: Numerical simulation of four-point correlations functions ($S_4$, $S_2$ and $S_1$) of eigenvectors. On the table, $S_4$, $S_2$ and $S_1$ correspond respectively to the terms where four, two and one different eigenvectors are involved. In this table, we indicate the mean and standard deviation values of these functions obtained by randomly choosing four eigenvectors ($10^6$ samples) from the 1500 ones of a typical empirical covariance (of size $1500 \times 1500$) of a random matrix.

| $S_4$ | $S_2$ | $S_1$ |
|---|---|---|
| $-1.509 \times 10^{-8} \pm 1.721 \times 10^{-5}$ | $6.667 \times 10^{-4} \pm 4.910 \times 10^{-5}$ | $0.002 \pm 8.315 \times 10^{-5}$ |

**Quantum corrections.** For such a quartic model quantum corrections can be easily investigated for large $N$. Relevant one-loop diagrams for 2 points functions are pictured on Figure 8 for the two kinds of vertices. These verices are pictured with solid edges materializing Kronecker delta defining how the field indices are contracted. Dotted edges on the other hand materialize Wick contractions, with propagator $C_{\mu\nu} = \lambda_\mu \delta_{\mu\nu}$. The first diagram on the left behaves as $\sim \delta_{\mu_1 \mu_2} N^{-1} \sum_\mu \lambda_\mu \equiv \delta_{\mu_1 \mu_2} N^{-1} \mathrm{Tr}\, C$. The second and third diagrams however behave as $\sim \delta_{\mu_1 \mu_2} N^{-1} \lambda_{\mu_1} \sim \delta_{\mu_1 \mu_2} N^{-2} \mathrm{Tr}\, C$, where we used self averaging property of random matrices. Then, in the large $N$ limit, the two last contributions on the right are of order $1/N$ with respect to the first one. More generally, contributions that create a large enough number of faces[21] will contribute significantly to the perturbation series and, for this reason, we can discard the vertex $\sum_{\mu=1}^N \psi_\mu^4$ in (47), whose contributions are always irrelevant for the contributions of the non-local term. Hence, we conclude that in the large $N$ limit, purely noisy data could be described by the nearly Gaussian distribution:

$$p(\Psi) = \frac{1}{Z} \exp\left( -\frac{1}{2} \sum_{\mu=1}^N \psi_\mu \lambda_\mu^{-1} \psi_\mu - \frac{g_4}{8N} \left( \sum_{\mu=1}^N \psi_\mu^2 \right)^2 + \mathcal{O}(\psi^6) \right) . \tag{48}$$

The remaining interactions - sextic, octic and so on - can be explicitly constructed in the same way, and we get $\sim \frac{1}{N^{n-1}} \left( \sum_\mu \psi_\mu^2 \right)^n$ for a monomial contribution involving $2n$ fields.

Interestingly, the interactions of the resulting field theory exhibit an effective $O(N)$ symmetry. As the explicit construction above shows, this is a consequence of the delocalized nature of the eigenvectors, implying that field configurations are defined as $\psi_\mu := \sum_i u_i^{(\mu)} \phi_i$ are rotationally invariant.

---

[21]Faces are conventionally defined as closed cycles.

It is instructive to investigate how perturbation theory relies on the spectra of $\tilde{D}^{-1}$ and $C$. The standard perturbation theory as described in any quantum mechanics textbook[22] give us the shifts of these quantities due to quantum corrections. We define the symmetric matrix $\epsilon \Xi_{ij} := C_{ij} - \tilde{D}_{ij}^{-1} = \mathcal{O}(g)$ the quantum corrections to the free propagator $\tilde{D}_{ij}^{-1}$, the parameter $\epsilon$ being assumed to be "small" and set it to 1 at the end of the computation. If we denote as $\lambda_\mu$ and $u_i^{(\mu)}$ on one hand, and has $\tilde{\lambda}_\mu$ and $\tilde{u}_i^{(\mu)}$ on the other hand respectively the eigenvalues and eigenvectors of matrices $C$ and $\tilde{D}^{-1}$, then:

$$\lambda_\mu = \tilde{\lambda}_\mu + \epsilon \Xi_{\mu\mu} + \epsilon^2 \sum_{\nu \neq \mu} \frac{|\Xi_{\mu\nu}|^2}{\tilde{\lambda}_\mu - \tilde{\lambda}_\nu} + \mathcal{O}(\epsilon^3), \tag{49}$$

and:

$$u_i^{(\mu)} = \tilde{u}_i^{(\mu)} + \epsilon \sum_{\nu \neq \mu} \frac{\Xi_{\mu\nu}}{\tilde{\lambda}_\mu - \tilde{\lambda}_\nu} \tilde{u}_i^{(\nu)} + \mathcal{O}(\epsilon^2), \tag{50}$$

where $\Xi_{\mu\nu} := \sum_{i,j} \Xi_{ij} \tilde{u}_i^{(\mu)} \tilde{u}_j^{(\nu)}$. In particular:

$$\sum_i u_i^{(\mu)} \tilde{u}_i^{(\nu)} = \delta_{\mu\nu} + \mathcal{O}(\epsilon^2). \tag{51}$$

Let us investigate how relevant contributions of the perturbation theory modifies the eigenvalues and eigenvectors. Using (48), the order $g$ correction to the 2-point function reads:

$$\langle \psi_\mu \psi_\nu \rangle = \tilde{\lambda}_\mu \delta_{\mu\nu} + \Xi_{\mu\nu}, \tag{52}$$

where, following the discussion before equation (48), the relevant contribution at order $g$ in the large $N$ limit for $\Xi_{\mu\nu}$ corresponds to the diagram on left in Figure 8. It is a simple exercise to show that:

$$\Xi_{\mu\nu} = -\tilde{\lambda}_\mu^2 \frac{g}{4} \left( \frac{1}{N} \sum_\mu \lambda_\mu \right) \delta_{\mu\nu} \rightarrow -\tilde{\lambda}_\mu^2 \frac{g}{4} \left( \int \mu(\lambda) \lambda d\lambda \right) \delta_{\mu\nu}. \tag{53}$$

Hence, off-diagonal contributions in (50) and (49) vanish, and:

$$\lambda_\mu = \tilde{\lambda}_\mu - \tilde{\lambda}_\mu^2 \frac{g}{4} \left( \int \mu(\lambda) \lambda d\lambda \right) + \mathcal{O}(g^2). \tag{54}$$

This is the standard Dyson equation, corresponding here to a global shift of mass. Indeed, defining $\lambda_\mu^{-1} := p_\mu^2 + m^2$ and $\tilde{\lambda}_\mu^{-1} := \tilde{p}_\mu^2 + \tilde{m}^2$, the previous equation reads:

$$\frac{1}{p_\mu^2 + m^2} = \frac{1}{\tilde{p}_\mu^2 + \tilde{m}^2} - \frac{1}{\tilde{p}_\mu^2 + \tilde{m}^2} \frac{g}{4} \left( \int \mu(\lambda) \lambda d\lambda \right) \frac{1}{\tilde{p}_\mu^2 + \tilde{m}^2} + \mathcal{O}(g^2), \tag{55}$$

or:

$$\frac{1}{p_\mu^2 + m^2} = \frac{1}{\tilde{p}_\mu^2 + \tilde{m}^2 + \frac{g}{4} \left( \int \mu(\lambda) \lambda d\lambda \right)} + \mathcal{O}(g^2), \tag{56}$$

meaning that $p_\mu^2 = \tilde{p}_\mu^2$ and:

$$m^2 = \tilde{m}^2 + \frac{g}{4} \int \mu(\lambda) \lambda d\lambda + \mathcal{O}(g^2). \tag{57}$$

---

[22]This is an elementary calculus that we can found for instance in [97].

Exploiting the delocalized structure of eigenvectors for a noisy dataset, we saw in this section how to construct a non-local theory space valid near the Gaussian regime. Although one expects that the domain of validity of the corresponding theoretical space is not as restrictive as the assumptions of the derivation, the later does not allows its scope. In particular, it seems quite difficult to consider distributions having large non-Gaussian effects. In the following section 3.2.3, we will propose an alternative to get away from the Gaussian point. To spoil its conclusions, this construction focus only on the tail of the spectrum (the deep IR), where it can be suitably approached by a power-law distribution. In that limit, one expect that the effective model behaves as an ordinary field theory,[23] and ultra-local interactions involving a Dirac delta conservation for the effective momentum $p_\mu$ are quite natural. The assumption to be close to the Gaussian point is removed, but in counterpart the UV scales are totally hidden for us. Hence, we identified two different regimes:

- A non local regime, valid in the vicinity of the Gaussian point for noisy datasets.

- A local regime, valid at the tail of the spectrum, but without additional assumption on the spectrum or on the size of the coupling.

These two limits are incompatibles and point out one of the greatest difficulties for data field theory as soon as one seeks to move away from the Gaussian distribution. Indeed, construct a theory space valid only for a restricted regime of the couple field theory/dataset seems like the best compromise, and how to move away from the Gaussian point will depend on this regime. Yet it is not too surprising as in the modern conception of field theory we are used to think that the validity of a field theory model is only effective in a restricted range of energy. Indeed, this is not an aspect specific to field theory, but more broadly in elementary physics. For instance, it is well known that the form of the friction forces depends on the speed: linearly if the latter is low and quadratically when it is high, without any possibility to switch continuously from one to the other. In that context, the unusual is not that the field theory has - maybe - a limited scope, but that the domains of validity seem to be very narrow.

**Remark 1** *The results we obtained about the 4-point correlations of the eigenvectors show that they practically do not overlap, which is consistent with their delocalized character for a random matrix. One could hope to use this property to detect the signal directly. For example, study how the $S_i$ of the table 1 change when a signal is added. We would then observe that the last sum $S_4$ is really sensitive to the presence of a signal. This sensitivity can be seen as a limitation of the non-local model, but one could not use it as a detection criterion since the values need to be "calibrated" by pure noise but also because their sensitivity threshold is well below the error bars. It is one of the strong points of formalism that we develop to present universal criteria defining a noise.*

### 3.2.3 Local field theory approximation
**Concluding remarks.** Instead of considering the full spectrum we are focusing on its tail i.e. the region of small momenta $p_\mu \approx 0$ which corresponds, from the ordinary field theory perspective, to large distance physics. In many instances (see section 5), the spectrum behaves asymptotically as a power law $\rho(p) \sim (p^2)^\delta$.

For purely noisy data we often have[24] $\delta = 1/2$ except at critical points (see definition 3 and [67]). Recalling that momentum distribution is $\rho_{\mathbb{R}^d}(\vec{p}) = (\vec{p}^2)^{\frac{d-2}{2}}$ in a space of dimension $d$, the exponent $\delta$ can be formally converted as a dimension: $d_0 = 2\delta + 2$. This dimension is equals to 3 for $\delta = 1/2$, but has no reason to be an integer in general. If it is an integer the

---

[23]For which momenta distributions are power laws as well.
[24]This is in particular the case for MP and Wigner distributions.

problem matches however exactly with an ordinary field theory. On a concrete example, now consider a purely noisy distribution well described by the MP law with $d_0 = 3$.

As argued in section 3.1 that the standard Ising model provide a good MEE we expect, for large scales, the model well described by an effective field theory like (25). For such a model, fields interact locally (i.e. at the same spatial index $i$), and a typical interaction reads as $V_n = \sum_i \phi_i^{2n}$. In Fourier modes, for systems where it is well defined, this interaction becomes:

$$V_n^{(d)} \sim \sum_{\{\vec{p}_1, \cdots, \vec{p}_{2n}\}} \delta\left(\sum_{i=1}^{2n} \vec{p}_i\right) \prod_{i=1}^{2n} \psi(\vec{p}_i). \tag{58}$$

In cases where $d_0$ is an integer, it is suitable to understand $\vec{p}$ as a $d$-dimensional vector ($\vec{p} \in \mathbb{R}^d$) but as pointed out before, this is not generally the case. Moreover, even if the asymptotic dimension is an integer which can be suitably interpreted as the dimension of some background space, only the tail of the spectrum is concerned, and the identification breaks as we investigate on UV scales.

Note that in such case some arrangements are still possible by expanding the true distribution $\rho(p^2)$ in power of $p^2$. This is for instance allowed concerning the MP (see equation (5)) law which reads as [52]

$$\rho_{MP}(p^2) = \frac{\sqrt{\lambda_+ \lambda_-}}{2\pi\alpha\sigma^2} \frac{(p^2)^{1/2}}{(p^2 + 1/\lambda_+)^2} \left(\frac{\lambda_+ - \lambda_-}{\lambda_+ \lambda_-} - p^2\right)^{1/2}, \tag{59}$$

and for small enough $p^2$,

$$\rho_{MP}(p^2) \approx \frac{\sqrt{\lambda_+ \lambda_-}}{2\pi\alpha\sigma^2} \lambda_+^2 (p^2)^{1/2} \left(1 + a_1 p^2 + a_2 (p^2)^2 + \cdots\right). \tag{60}$$

This difficulty was raised on in the previous section and has been solved by fixing the effective dimension to 1. In that way, we avoid the difficulty to define the effective (and generally non-integer) dimension of the space to which $\vec{p}$ belongs. In counterpart the elementary volume in momentum space remains $\rho(p^2)p\,dp$ rather than $dp$ as it should be in dimension 1. Typical interaction (58) therefore becomes:

$$V_n^{(1)}[\psi] \sim \bar{V}_n^{(1)}[\psi] := \sum_{\{p_1, \cdots, p_{2n}\}} \delta\left(\sum_{i=1}^{2n} p_i\right) \prod_{i=1}^{2n} \psi(p_i), \tag{61}$$

and we introduce the following definition:

**Definition 5** *The local theory space is the functional space spanned by interactions $V_n^{(1)}$ defined by (61). A local functional $H[\psi]$ then admits the following expansion:*

$$H[\psi] := N \sum_{n=1}^{K} \frac{u_{2n}}{N^n} \bar{V}_n^{(1)}[\psi], \tag{62}$$

*for some coupling constant $\{u_{2n}\}$.*

The origin of the factor $1/N^{n-1}$ will be motivated in section 4. We were able to argue that at least some models exhibiting these interaction can reproduce a completely noisy spectrum, like MP. By the same universality arguments discussed in Figure 7, we expect the validity of the model to go beyond the specific framework of its construction. Thus, and if it is true that in some particular cases this model constitutes a good approximation of field theory, around a universal noise, then this same model should be able to serve as a basis for any problem in

the neighbourhood of the same universality class. For this reason, we expect the local model to be a good approximation when trying to highlight the presence of a signal at the tail of the spectrum. The whole section 4 is dedicated to the study of the non-perturbative group, corresponding to this theory space for a wide variety of spectrum around ordinary noise models and showing the relevance of the local approximation.

## 4   Effective average action for local theory

In this section, we discuss the investigations by the RG. We work in the functional renormalization-group (FRG) formalism, which gathers a whole set of techniques and effective equations translating the dependence of interactions on the observation scale in the Wilsonian point of view. In this framework, the RG is conceived as a progressive partial integration of the degrees of freedom, the integrated effects (UV) being hidden in the values of the coupling constants. In the ordinary case, this RG is well defined when the degrees of freedom can be unambiguously labelled to define a unique physically relevant integration order. Nevertheless, in standard field theories, such as the one described by the action (14), this labelling is given by the spectrum of the kinetic operator $H_0 := -\Delta^2 + m^2$, the large eigenvalues corresponding to the UV scales being integrated first.

For the field theory we are now focusing on, the relevant spectrum will be the *interacting 2-point function*, i.e. the matrix, $C$. We will choose to integrate over the large moments $p_\mu$ to build the effective theory valid in the IR. Although unconventional and posing some technical difficulties, basing the partial integration on the empirical distribution of $C$ rather than upon the kinetic matrix $\tilde{D}^{-1}$ lies on the fact that the latter is essentially unknown. Indeed, $\tilde{D}^{-1}$ is related to $C$ by the following equation:

$$C = \tilde{D}^{-1} + \tilde{D}^{-1}\Sigma\tilde{D}^{-1} + \tilde{D}^{-1}\Sigma\tilde{D}^{-1}\Sigma\tilde{D}^{-1} + \cdots, \qquad (63)$$

where the self-energy $\Sigma = \mathcal{O}(\mathfrak{G})$ contain the 1PI information on quantum corrections ($\mathfrak{G}$ denoting the set of coupling constants).

The reverse inference problem to determine $\tilde{D}^{-1}$ from the knowledge of $C$ is expected hard. Difficulty comes from the fact that a solution is generally not unique, and that the so-called irrelevant operators playing a negligible role on a large scale. Moreover, the accuracy at which the large scale solution is measured is not infinite, the basin of attraction of UV theories admitting this limit in regard to the experimental errors is large enough [3] - see Figure 9. We say that RG has a *large river effect* [98]. In that way (63) looks as a big constraint along the flow, linking UV scales where 2-point function is $\tilde{D}^{-1}$ and IR scales where 2-point function is $C$. It is expected that such a constraint cannot be solved exactly, and we need approximations to deal with it.

All our investigations in this paper focus on different versions of the local potential approximation. In this approximation and in the symmetric phase, only the mass has a non-trivial flow for 2-point functions, and all the eigenvalues of the $\tilde{D}$ matrix are translated by the same constant:

$$C_{ij}^{-1} \approx \tilde{D}_{ij} + \kappa(\mathfrak{G})\delta_{ij}, \qquad (64)$$

the constant $\kappa$ summing all the quantum effects. The numerical results of the next section will show the consistency of this assumption.

In this paper, we will limit ourselves to studying the existence or non-existence of phase transitions as a function of signal strength, focusing essentially on the shape of the effective potential, and assumption (64) seems no so restrictive. From the Wilsonian point of view, the RG can be understood as a mapping between Hamiltonians at different scales of description,

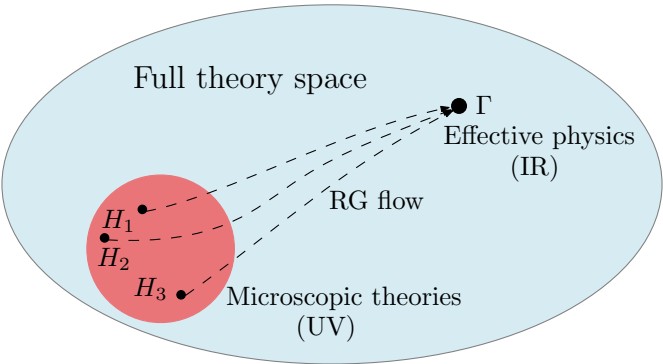

Figure 9: Basin of attraction of the IR theory.

the notion of scale being fixed by the spectrum of the two-point function. There are many practical incarnations of this idea, one of the best known being the Polchinski equation [11] which describes precisely the evolution of the Hamiltonian when degrees of freedom are partially integrated within an infinitesimal window.

However, in practise and in particular when one is interested in non-perturbative phenomena, the framework of the effective average action (EAA) method[25] through the formalism of Wetterich-Morris [56, 57] is preferred. This method offers, in the non-perturbative sector, a better convergence properties than the Polchinski equation [99].[26] The central object of the EAA method $\Gamma_k$ is the effective action for integrated out (i.e. UV) degrees of freedom, thus interpolating between the microscopic Hamiltonian $H$ and the global effective action (i.e. IR) of the $\Gamma$ model, including all quantum corrections and usually defined as the Legendre transform of the free energy $\mathcal{W} := \ln Z$.

A motivation justifying the non-perturbative formalism use was the surprising observation that the couplings all become strongly relevant in the deep UV (see Figure 10). Thus, even if we focus on studying the IR behavior first (where the signal is located!), we cannot exclude the possibility that the flow was carried very far from the Gaussian point. Finally, it should be noted that the fact that a small number of couplings (essentially sextic and quartic – see empirical statement (1)) survive in the IR also justifies the vertex expansion that we will preferentially use in this study, upstream of more elaborate methods capable of considering the deep UV.

We note to conclude the approximation schemes we will consider in the next sections (vertex expansion and local potential approximation) are well known in the literature (see [98] for instance). However, we have chosen, in view of the unconventional characteristics of the theory and because we do not assume the reader to be familiar with this formalism, to give many details on the construction of the flow equations.

---

[25]We use the terminology "action" to designate IR quantities. We call for instance "effective action" the generating functional of 1PI diagrams $\Gamma$, reserving the name "Hamiltonian" for UV quantities. The two definitions coincide when no fluctuations are integrated out.

[26]Note that the philosophy underlying the Wetterich-Morris approach differs from the Wilson-Polchinski strategy. In the Wilson-Polchinski point of view, degrees of freedom are progressively integrated-out from UV to IR scales, and the microscopic description (the classical hamiltonian $H$) changes at each steps. In the Wetterich-Morris point of view on the other hand, the microscopic hamiltonian is left unchanged, but IR contributions to the classical action are progressively removed.

## 4.1 The Wetterich-Morris equation

The effective action $\Gamma[M]$, which describe IR physics is defined as the Legendre transform of the free energy $\mathcal{W}[\chi] := \ln Z(\chi)$,

$$\Gamma[M] + \mathcal{W}[\chi] = \sum_\mu \chi(-p_\mu) M(p_\mu), \tag{65}$$

where the classical field $M = \{M(p_\mu)\}$ is:

$$M(p_\mu) = \frac{\partial W[\chi]}{\partial \chi(-p_\mu)}. \tag{66}$$

The starting point of the EAA method is to modify the microscopic Hamiltonian $H[\psi]$ by adding a mass term $\Delta H_k[\psi]$,

$$\Delta H_k[\psi] = \frac{1}{2} \sum_\mu \psi(p_\mu) r_k(p_\mu^2) \psi(-p_\mu). \tag{67}$$

It depends both on the momentum $p_\mu$ and a continuous and positive index $k \in [0, \Lambda]$. This index have the dimension of a momentum and will play the role of a *IR cut-off scale*. The upper bound $\Lambda$ corresponds to the UV cut-off, materializing the microscopic scale but not coincide necessarily with the difference $1/\lambda_- - m^2$ discussed in Section 3.2.1, equation 36. We thus build a continuous family of models, admitting in principle the same physics at large distances, and interpolating smoothly between UV and IR scales:

$$W_k[\chi] := \ln \int [d\psi] e^{-H[\psi] - \Delta H_k[\psi] + \sum_\mu \chi(-p_\mu)\psi(p_\mu)}. \tag{68}$$

The momenta scale $r_k(p_\mu^2)$ is designed to provides an operational description of the coarse-graining procedure and its design much satisfy the following requirements:

1. $r_{k=0}(p^2) = 0 \ \forall p^2$, meaning that for $k = 0$, $W_k \equiv W$, all the fluctuations are integrated out.

2. $r_{k=\Lambda}(p^2) \gg 1$, meaning that all fluctuations are frozen with a very large mass in the deep UV.

3. $r_k(p^2) \approx 0$ for $p^2/k^2 < 1$, meaning that high energy modes with respect to the scale $k^2$ are essentially unaffected by the regulator. In contrast, low energy modes must have a large mass that decouples them from long-distance physics.

The first two conditions ensure that we find the two effective descriptions at the boundaries: on the one hand in the deep UV where physics is described by $H$, and on the other hand in the deep IR where physics is described by the effective action $\Gamma$. The interpolation between them is achieved by the *effective averaged action* $\Gamma_k$ defined as:

$$\Gamma_k[M] + W_k[\chi] = \sum_\mu \chi(-p_\mu) M(p_\mu) - \frac{1}{2} \sum_\mu M(p_\mu) r_k(p_\mu^2) M(-p_\mu), \tag{69}$$

such that $\Gamma_{k=0} \equiv \Gamma$ and, from the conditions on $r_k$, $\Gamma_{k=\Lambda} \sim H$. $\Gamma_k$, as $k$ varies describes a trajectory through the theory space (see Figure 9), and the different couplings change, there variations being described by the Wetterich-Morris equation:

$$\dot{\Gamma}_k = \frac{1}{2} \sum_\mu \dot{r}_k(p_\mu^2) \left( \Gamma_k^{(2)} + r_k \right)^{-1} (p_\mu, -p_\mu), \tag{70}$$

the dot being defined as:

$$\dot{X} \equiv \frac{dX}{dt} := k\frac{dX}{dk}. \tag{71}$$

Up to the assumption that we work into the local theory space, equation (70) is exact.

Unfortunately, solving it exactly is reputed to be a difficult or impossible task, even for simple problems. Obtaining nonperturbative information on the flow behavior therefore necessarily requires approximations. Although there are general methods, most of them have to be adapted to each problem, and we will in the next section examine how these methods can be applied to the unconventional field theory we consider.

Let us conclude this section with a remark. The equation (70) involves a sum over momenta $p_\mu$. Because we will focus on the $N \gg 1$ limit in our calculations and simulations, one expects to be able to substitute the discrete sum by an integral invoking the density $\sum_\mu \to \int \rho(p^2)p\,dp$. This substitution however ignores the "zero" mode associated to the eigenvalue $\lambda_+ = 1/m^2$. The RG in this context is best understood as describing the evolution of the effective couplings coupling the zero modes. Thus, the smallest value of $k$ does not have to be strictly zero but stops at the eigenvalue just above the zero modes, the spacing between the eigenvalues being of the order of $1/N$, we would rather have $k \in [\sim 1/\sqrt{N}, \Lambda]$. Hence, for large $N$:

$$\dot{\Gamma}_k = N \int_0^\infty dp\, \rho(p^2)p\dot{r}_k(p^2)\left(\Gamma_k^{(2)} + r_k\right)^{-1}(p,-p), \tag{72}$$

where we put the upper limit to $+\infty$, assuming the windows of momenta allowed by $\dot{r}_k$ is quite restrictive. This situation is reminiscent of finite geometry models, which is not surprising since the integral on the modes is well bounded $\frac{1}{N}\sum_\mu 1 \sim \int \rho(p^2)p\,dp = \mathcal{O}(1)$. If we think for example of a one-dimensional lattice theory with periodic boundary conditions, the moment is quantized as $p_n = \frac{2\pi n}{N}$, $N$ denotes the number of sites. The difference between two values is then equal to $2\pi/N$, and the number of moments equal to $\sum_n 1 = N$, the "volume" of the network.

## 4.2 Scaling dimensions

The notion of scale plays an essential role insofar as the RG aims precisely at determining the scale dependence of physical laws. In this context, we have fixed the scale by the moment $p_\mu$. Another important piece of information for the RG is the way couplings change in the neighbourhood of a fixed point when quantum corrections are negligible. In the vicinity of the Gaussian fixed point, this dependence defines the *scaling dimension*. We are aiming to generalize the standard definition in this context.

As a first looks, we focus on the *symmetric phase*, where $M = 0$ is assumed to be a stable solution of the quantum equations of the move. In that phase, expansion around vanishing classical field is allowed, and from the expected $\mathbb{Z}_2$ symmetry of the microscopic model (see (25)), it must contain only even couplings. In that way, one expects that odd vertex function $\Gamma_k^{(2n+1)}$ vanish identically. Taking the second derivative for $M$ of Wetterich-Morris equation (70), we have:

$$\dot{\Gamma}_k^{(2)}(p_{\mu_1}, -p_{\mu_1}) = \frac{1}{2}\sum_{p_\mu} \dot{r}_k(p_\mu^2)G_k^2(p_\mu^2)\Gamma_k^{(4)}(p_\mu, -p_\mu, p_{\mu_1}, -p_{\mu_1}), \tag{73}$$

where:

$$G_k(p_\mu^2) = \left(\Gamma_k^{(2)} + r_k\right)^{-1}(p_\mu, -p_\mu), \tag{74}$$

and where we assume to work into the local theory space.

First, it is worth noting that the mass in the deep IR, corresponding to the inverse of the largest eigenvalue and defined by the condition

$$m^2 := \Gamma_{k=0}^{(2)}(0,0), \tag{75}$$

must behave like any eigenvalue under a global dilation of the spectrum. In other words, $m^2$ must have the same scaling behavior as $\Lambda^2$. Hence, the 2-point function at zero momenta,

$$u_2(k) := \Gamma_k^{(2)}(0,0), \tag{76}$$

must have the same scaling as $k^2$, and we define the *dimensionless* mass $\bar{u}_2$ as:

$$u_2(k) =: k^2 \bar{u}_2(k). \tag{77}$$

Following the standard definition in field theory, this means that the scale dimension of the mass $u_2$ is 2.

We shall now return to the equation (73). Assuming that $\dot{r}_k$ allows only a narrow window of moments around $k^2$, and that $k^2 \ll 1$ following the fact that we are only interested in the tail of the spectrum, we expect to be able to neglect the dependence in $p_\mu$ in $\Gamma_k^{(4)}$, replacing $p_\mu$ by 0. Finally, in the local potential approximation, $\Gamma_k^{(4)}$ must have the form of a local vertex,

$$\Gamma_k^{(4)}(p_1, p_2, p_3, p_4) = \frac{u_4(k)}{N} \delta_{p_1+p_2+p_3+p_4}, \tag{78}$$

accordingly to the definition 5. We assume to work in the large $N$ limit so that we can suitably replace the sum by an integral involving density $\rho(p^2)$. For a power-law distribution $\rho(p^2) \sim (p^2)^\delta$, the remaining loop integral behaves as $k^{2\delta+2}$. Hence taking into account the scaling dimension for $u_2$ and setting $p_{\mu_1} = 0$, we conclude that explicit $k$-dependence on both sides of the flow equations cancels if $g_2 \sim k^{2-2\delta}$ – i.e. if $g_2$ has scaling dimension $d_{g_2} = 2-2\delta$. For an ordinary field theory in dimension $d$, $\delta = d/2 - 1$, and we recover the ordinary power counting $d_{g_2} = 4-d$.

In our case, the situation is not quite so rosy. We suppose that the distribution of $p_\mu$ is given by $\rho(p^2)$, which is not a power law. As we pointed out above, it is exactly as if the effective dimension of the space depended on the scale. Under these conditions, it is impossible to imagine getting rid of the explicit scale dependence completely. The best compromise can be imagined is to move this dependence to the level of the linear term in the coupling, through a canonical scale-dependent dimension.

In this paper, we will focus on step regulators, $r_k(p^2) \propto \theta(k^2 - p^2)$, where $\theta(x)$ denotes the ordinary Heaviside step function, equals to zero for $x < 0$ and equals to 1 for $x > 0$. In that way, the scaling factor $L(k)$ for the loop integral involving in (73) behaves as:

$$L(k) = \int_0^k \rho(p^2) p\, dp. \tag{79}$$

For a power law distribution we recover $L(k) \propto k^{2\delta+2}$ knowing that the time $t$ of the flow defined by (71) is $dt \propto d\ln L$. We generalize this definition, by replacing the time $dt = d\ln(k)$ by a new time $\tau$ defined by:

$$d\tau := d\ln\left(\int_0^k \rho(p^2) p\, dp\right). \tag{80}$$

For a power law distribution we get $d\tau = (2\delta + 2)dt$. To be more concrete, we consider the Litim regulator:

$$r_k(p^2) := z(k)(k^2 - p^2)\theta(k^2 - p^2), \tag{81}$$

where we included the wave function renormalization effects $z(k)$ and which we will covered later on in the next section. If we work with this regulator in the symmetric phase, the contributions of effective propagators $G(p_\mu^2)$ drop out the integral as a factor $(1+\bar{u}_2)^{-1}$. Moreover, because the internal loop has no dependency with respect the external momenta, the anomalous dimension must have a vanishing flow:

$$\dot{z} = 0.$$ (82)

Hence, making the substitution $t \to \tau$, it is easy to check that canonical dimension for $u_2(k)$ is multiplied by $t'$, where the notation $X'$ denotes the derivative with respect to $\tau$, and we denote the new scaling dimension as

$$\boxed{\dim_\tau(u_2) := 2t'.}$$ (83)

In that way, the contribution proportional to $u_4$ receives the scale-dependent factor $\rho(k^2)k^{-2}(t')^2$. This equation will be derived with full details in the next section and the reader may assume it at the present stage. Getting rid of the explicit scale dependence in the loop term thus amounts to defining the *locally dimensionless coupling* as:

$$\bar{u}_4 := u_4 \frac{\rho(k^2)}{k^2}\left(\frac{dt}{d\tau}\right)^2.$$ (84)

Using $\bar{u}_4$ instead of $u_4$ in the flow equation for $u_4$ hence introduce a linear contribution $-\dim_\tau(u_4)\bar{u}_4$, with:

$$\boxed{\dim_\tau(u_4) := -2\left(\frac{t''}{t'} + t'\left(\frac{1}{2}\frac{d\ln\rho}{dt} - 1\right)\right).}$$ (85)

The flow equation for $u_4$ in turn fix the $\tau$-dimension of $u_6$, and we define the local dimensionless sextic coupling as:

$$u_6 k^2 \left(\frac{\rho(k^2)}{k^2}\left(\frac{dt}{d\tau}\right)^2\right)^2 =: \bar{u}_6.$$ (86)

Hence, replacing $u_6$ with $\bar{u}_6$ in its own flow equation introduce the linear term $-\dim_\tau(u_6)\bar{u}_6$, with:

$$\boxed{-\dim_\tau(u_6) := 2\frac{dt}{d\tau} + 4\left(\frac{t''}{t'} + t'\left(\frac{1}{2}\frac{d\ln\rho}{dt} - 1\right)\right).}$$ (87)

The same argument can be generalized, and for $u_{2p}$ a simple recurrence leads to:

$$\boxed{-\dim_\tau(u_{2p}) := 2(p-2)\frac{dt}{d\tau} - (p-1)\dim_\tau(u_4).}$$ (88)

To anticipate the forthcoming extended discussions of Section 5, we illustrate the behavior of canonical dimensions here for the MP distribution. Figure 10 provides a numerical plot of $\dim_\tau(u_{2n})$ for MP distribution, the observed behavior being qualitatively the same for other choices of parameters, and in fact, very similar for other models of noise. In the deep IR i.e. in the domain corresponding to large eigenvalues, we can see that only a few couplings - the quartic and sextic ones - are relevant in agreement with the empirical statement 1. This is not too surprising. Indeed the momentum distribution behaving as $(p^2)^\delta$ with $\delta = 1/2$, and the canonical dimensions for a power law being $\dim_t(u_{2p}) = 2(1-(p-1)\delta)$, we find that relevant couplings are for $p = 1, 2, 3$, the last case corresponding to a marginal coupling with

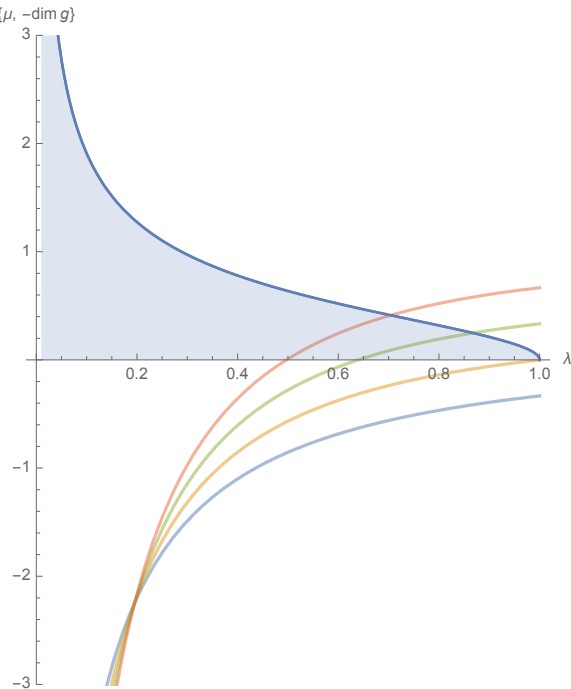

Figure 10: The canonical dimension for MP distribution with $\alpha = 1$ and $\sigma = 0.5$. The purple curve corresponds to the MP distribution $\rho_{MP}(p^2)$.

vanishing scaling dimension. What we obtain in that limit, a field theory having a few numbers of relevant directions is the most current one in field theory.[27]

In contrast, in the deep UV, i.e. in the domain of very small eigenvalues, the canonical dimensions become positives for an arbitrarily large number of interactions. This corresponds to a *dimension crisis*, meaning in the RG language, that an arbitrarily large number of couplings become relevant toward the IR scales, with arbitrary larges values. In such a regime the flow is no longer predictive since an arbitrarily large number of couplings must be initially fixed, and the truncations become arbitrarily large. We have two regimes, a "good" IR regime and a "bad" UV regime, a pessimistic estimate of the transition point between these two regimes, $k =: \Lambda_0$ being given by the scale where the canonical dimension of $u_8$ cancels, i.e.

$$\left[ \frac{dt}{d\tau} - \frac{3}{4} \dim_\tau(u_4) \right]_{t=\ln(\Lambda_0)} = 0 \,. \tag{89}$$

Our field theory was not intended to be more than an effective model valid at large distances anyway, but it is interesting to note that the theory sets its limits in a way. Our field theory was not intended to be more than an effective model valid at large distances anyway, but it is interesting to note that the theory sets its limits in a way. Numerically, we find for MP that this limit corresponds to the eigenvalue domain $\lambda \sim \lambda_+/3$. To summarize, the theory shows the existence of two regions. A region that we will call the *learnable region* (LR), for $k < \Lambda_0$. In this region, only $u_4$ and $u_6$ are relevant and the theory seems to be effectively predictive. On the contrary, for $k \gg \Lambda_0$, what we will call the *deep noisy region* (DNR), the number of relevant couplings becomes arbitrarily large, and the values taken by the dimensions also diverge.

---

[27]Once again, we assume that the form of interactions is fixed in this argument. In general, the power counting depends on the form of the interactions.

## 4.3 Local potential approximation

In this section, we introduce the *local potential approximation* (LPA) to construct solutions of the exact Wetterich-Morris equation (70). To begin we focus on the zero vacuum expansion, assuming to be in the symmetric phase. This assumption allows deducing flow equation in a simple form, with zero anomalous dimension to all orders. This formalism will allow familiarizing the reader with the specificity of the field theory that we consider, and in particular to highlight technical points discussed in the previous section about scaling dimension. However, because numerical investigations of section 5 show that a symmetry breaking is expected for a strong enough signal, we extend the formalism outside of the symmetric phase and consider expansion around the non-zero vacuum. All our derivations assume the validity of the *derivative expansion* (DE) [98]. We discuss the validity of this assumption in section 5 and, for this, we provide an explicit derivation for anomalous dimension, which does not vanish in the non-symmetric phase.

### 4.3.1 Symmetric phase expansion

In the symmetric phase, $\Gamma_k$ can be expanded in power of $M$. It is suitable to introduce the following decomposition: $\Gamma_k[M] = \Gamma_{k,\text{kin}}[M] + U_k[M]$, where $\Gamma_{k,\text{kin}}[M]$, the *kinetic part*, keeps only the quadratic terms in $M$ and $U_k[M]$, the *potential*, is a sum of monomials with powers of $M$ higher than 2. In the LPA, $U_k[M]$ is assumed to be a purely local function accordingly to the definition 5. Moreover, we assume that $U_k$ is an even function, i.e. $U_k[M] = U_k[-M]$. For the kinetic parts $\Gamma_{k,\text{kin}}[M]$, whose inverse propagates the local modes, we assume the validity of the (DE), and make the ansatz:

$$\Gamma_{k,\text{kin}}[M] = \frac{1}{2} \sum_p M(-p)(z(k,p^2)p^2 + u_2(k))M(p), \tag{90}$$

where $z(k,p^2)$ expands in power of $p^2$ as $z(k,p^2) = z(k) + \mathcal{O}(p^2)$.

In this section, we focus on the first order of the DE and keep only the term of order $(p^2)^0$ in the expansion of $z(k,p^2)$. Moreover, in the symmetric phase, as we will do explicitly in the next subsection, the flow equation for $z(k)$ vanishes exactly. It is therefore suitable to fix the normalization of fields, such that $z(k) = 1 \ \forall \, k$.

The derivation of the flow equations follows the strategy explained in the previous section. Taking the second derivative of the Equation (70) with respect to $M(p_\mu)$ leads to the flow equation (73). Then, taking successive derivatives, we generate flow equations for higher vertex function, but the flow for $\Gamma_k^{(2n)}$ involving $\Gamma_k^{(2n+2)}$, the hierarchy does not stop anywhere. To stop it, we must truncate the flow, i.e. project it into a finite-dimensional subspace, by posing:

$$\Gamma_k^{(2M)} = 0, \tag{91}$$

up to a given $M$. In that section we will consider explicitly the truncation around $M = 3$, taking into account only local sextic effective interactions:

$$\Gamma_k[M] = \frac{1}{2} \sum_p M(-p)(p^2 + u_2(k))M(p)$$

$$+ \frac{u_4(k)}{4!N} \sum_{\{p_i\}} \delta\left(\sum_i p_i\right) \prod_{i=1}^{4} M(p_i)$$

$$+ \frac{u_6(k)}{6!N^2} \sum_{\{p_i\}} \delta\left(\sum_i p_i\right) \prod_{i=1}^{6} M(p_i). \tag{92}$$

From this one, we straightforwardly deduce that:

$$\Gamma^{(2)}_{k,\mu_1\mu_2} = \delta_{p_{\mu_1},-p_{\mu_2}} \left( p^2_{\mu_1} + u_2(k) \right) , \tag{93}$$

and:

$$\Gamma^{(4)}_k(p_{\mu_1}, p_{\mu_2}, p_{\mu_3}, p_{\mu_4}) = \frac{g}{4!N} \sum_\pi \delta_{0,p_{\pi(\mu_1)}+p_{\pi(\mu_2)}+p_{\pi(\mu_3)}+p_{\pi(\mu_4)}} , \tag{94}$$

where $\pi$ denotes elements of the permutation group of four elements. Note that the origin of the factors $1/N$ and $1/N^2$ can be easily traced now. Indeed, as we will see just below, the $1/N$ in front of $u_4$ ensures that (73) can be rewritten as an integral in the large $N$ limit, involving the effective distribution $\rho(p^2)$. In the same way, the $1/N^2$ in front of $u_6$ ensures that all the contributions to the flow of $u_4$ receive the same power $1/N$. The argument can be easily generalized, and we understand the origin of the factor $1/N^{n-1}$ in definition 5 for $u_{2n}$. Finally, the division by $1/(2n)!$ ensures that the symmetry factors of the Feynman diagrams match exactly with the dimension of its discrete symmetry group.

The effective propagator can be easily computed:

$$G_k(p^2_\mu) = \frac{1}{p^2_\mu + u_2 + r_k(p^2_\mu)} . \tag{95}$$

Moreover, we choose to work with the Litim regulator (81) which turns out to be a convenient choice for doing analytical calculations. The choice of the controller is known to be a serious problem, which can lead to non-trivial and pathological dependence of the results [98]. The fact is that, although the Wetterich equation formally implies certain independence for the choice of the controller, the truncation itself can introduce a strong dependence, even when $k \to 0$. The Litim regulator will therefore be the easy solution. Moreover, focusing on the shape of the effective potential and not on the computation of critical exponents, we expect the dependence on the regulator to be fewer [100].

The flow equation for $u_2$ can be deduced from (73) setting external momenta to zero. We obtain:

$$\dot{u}_2 = -\frac{1}{2N} \frac{2k^2}{(k^2+u_2)^2} \sum_{p_\mu} \theta(k^2 - p^2_\mu) \Gamma^{(4)}_k(p_\mu, -p_\mu, p_{\mu_1}, -p_{\mu_1}) \Big|_{p_{\mu_1}=0} . \tag{96}$$

The factor $1/N$ in front of the sum allows to convert it as an integral:

$$\dot{\bar{u}}_2 = -2\bar{u}_2 - \frac{2u_4}{(1+\bar{u}_2)^2} \frac{1}{k^4} \int_0^k \rho(p^2) p \, dp , \tag{97}$$

where $\bar{u}_2 := k^{-2} u_2$ and where we used the expression (94) for $\Gamma^{(4)}_k$. Using the time flow $\tau$ defined by (80), we get:

$$\frac{d\bar{u}_2}{d\tau} = -2\frac{dt}{d\tau} \bar{u}_2 - \frac{2u_4}{(1+\bar{u}_2)^2} \frac{\rho(k^2)}{k^2} \left( \frac{dt}{d\tau} \right)^2 . \tag{98}$$

Hence, using definitions (84), we obtain:

$$\boxed{\frac{d\bar{u}_2}{d\tau} = -2\frac{dt}{d\tau} \bar{u}_2 - \frac{2\bar{u}_4}{(1+\bar{u}_2)^2} .} \tag{99}$$

The flow equation for the coupling $u_4$ can be deduced following the same strategy. Taking the fourth derivative with respect to $M$ of the flow equation (70) and discarding the odd functions which vanish in the symmetric phase, we thus obtain:

$$\frac{du_4}{d\tau} = -\frac{2u_6}{(1+\bar{u}_2)^2} \rho(k^2) \left( \frac{dt}{d\tau} \right)^2 + \frac{12u_4^2}{(1+\bar{u}_2)^3} \frac{\rho(k^2)}{k^2} \left( \frac{dt}{d\tau} \right)^2 . \tag{100}$$

Taking into account definition (86), the equation reads as:

$$\frac{d\bar{u}_4}{d\tau} = -\dim_\tau(u_4)\bar{u}_4 - \frac{2\bar{u}_6}{(1+\bar{u}_2)^2} + \frac{12\bar{u}_4^2}{(1+\bar{u}_2)^3}.$$

(101)

Finally, we get for $u_6$, setting $u_8 \approx 0$, we have:

$$\frac{d\bar{u}_6}{d\tau} = -\dim_\tau(u_6)\bar{u}_6 + 60\frac{\bar{u}_4\bar{u}_6}{(1+\bar{u}_2)^3} - 108\frac{\bar{u}_6^3}{(1+\bar{u}_2)^4}.$$

(102)

### 4.3.2 Non-zero vacuum expansion

Future numerical investigations show the limits of the development in the vicinity of the zero vacuum. Also in this section, we extend the formalism to the case of a non-zero vacuum. This formalism is particularly suitable for investigations in the deep IR, and we will assume that the vacuum only affects the zero component $M(p_\mu) \sim M\delta_{\mu,0}$, neglecting the momentum dependence of the classical field $M(p_\mu)$. This approximation works well at a large scale, where a symmetry breaking scenario is expected, requiring an expansion around a non-vanishing vacuum $M \neq 0$. For this reason, we consider the following parameters:

$$U_k[\chi] = \frac{u_4(k)}{2!}\left(\chi - \kappa(k)\right)^2 + \frac{u_6(k)}{3!}\left(\chi - \kappa(k)\right)^3 + \cdots$$

(103)

We denoted as $\kappa(k)$ the *running vacuum*. The global normalization is chosen such that, for $M_0(p) = M\delta_{p0}$, $\Gamma_k[M = M_0] = NU_k[\chi]$, and $N\chi := M^2/2$. The 2-point vertex $\Gamma_k^{(2)}$ moreover is defined as:

$$\Gamma_{k,\mu\mu'}^{(2)} = \left(z(k)p^2 + \frac{\partial^2 U_k}{\partial M^2}\right)\bigg|_{M^2=2N\chi}\delta_{p_\mu,-p_{\mu'}}.$$

(104)

Note that we introduced the anomalous dimension $z(k)$, which has a non-vanishing flow equation for $\kappa \neq 0$. This equation replaces the formula (93), the second derivative of the potential playing the role of an effective mass. The flow equation for $U_k$ can be deduced from (70), setting $M = M_0$ on both sides. For large $N$, taking the continuum limit, we get:

$$\dot{U}_k[M] = \frac{1}{2}\int dp^2 \, k\partial_k(r_k(p^2))\rho(p^2)\left(\frac{1}{\Gamma_k^{(2)} + r_k}\right)(p,-p).$$

(105)

In the computation of the flow equations, it is suitable to rescale the dimensionless couplings

$$\bar{u}_{2p} \to z^{-p}\bar{u}_{2p},$$

(106)

for instance $\bar{u}_2 := z^{-1}k^{-2}u_2$. This ensures that the coefficient in front of $p^2$ of the kinetic action remains equal to 1. This additional rescaling adds a term $n\eta(k)$ in the flow equation, where $\eta$, the *anomalous dimension* is defined as:

$$\eta(k) := \frac{\dot{z}(k)}{z(k)}.$$

(107)

Despite that the computation is thereby greatly simplified, the factor $z$ in front of the regulator (81) does not have to affect the boundary conditions for $k = 0$ and $k = \infty$, namely:

$$\Gamma_{k=\infty} \to H, \quad \text{and} \quad \Gamma_{k=0} \to \Gamma.$$

(108)

In particular, the first one of this conditions requires that $r_{k\gg1} \sim k^r$, for some positive $r$. This is obviously the case for $z = 1$ because $r_{k\gg1} \sim k^2$. But the dependence of $z$ on $k$ can break this condition. This may happens if the flow reach a fixed point with anomalous dimension $\eta_* \neq 0$. The anomalous dimension behaves as $z(k) = k^{\eta_*}$ and $r_{k\gg1} \sim k^{2+\eta_*}$. Hence, the requirement $r > 0$ imposes in turn:

$$\eta_* > -2, \tag{109}$$

that we call *regulator bound*. Note that this is a limitation of the regulator, not of the method. Moreover, the flow equation being intrinsically non-autonomous, no exact fixed points are expected and the criterion should be more finely defined. Generally, the LPA at the lowest order in the DE makes sense only in regimes where $\eta$ remains small enough, for $|\eta| \gtrsim 1$ [100, 101].

**RG equation for $\eta = 0$.** As a first approximation we focus on standard LPA, setting $z(k) = 1$ or equivalently $\eta = 0$. From (105), we arrive to the following expression for the effective potential flow equation:

$$\dot{U}_k[\chi] = \left( 2 \int_0^k \rho(p^2) p \, dp \right) \frac{k^2}{k^2 + \partial_\chi U_k(\chi) + 2\chi \partial_\chi^2 U_k(\chi)}. \tag{110}$$

As discussed in the previous section, we express it in terms of the flow parameter $\tau$, to obtain:

$$U'_k[\chi] = k^2 \rho(k^2) \left( \frac{dt}{d\tau} \right)^2 \frac{k^2}{k^2 + \partial_\chi U_k(\chi) + 2\chi \partial_\chi^2 U_k(\chi)}. \tag{111}$$

Accordingly with the definitions adopted in the symmetric phase, we define the scaling of the effective potential as:

$$\partial_\chi U_k(\chi) k^{-2} = \partial_{\bar{\chi}} \bar{U}_k(\bar{\chi}), \quad \chi \partial_\chi^2 U_k(\chi) k^{-2} = \bar{\chi} \partial_{\bar{\chi}}^2 \bar{U}_k(\bar{\chi}), \tag{112}$$

leading to:

$$\boxed{U'_k[\chi] = \left( \frac{dt}{d\tau} \right)^2 \frac{k^2 \rho(k^2)}{1 + \partial_{\bar{\chi}} \bar{U}_k(\bar{\chi}) + 2\bar{\chi} \partial_{\bar{\chi}}^2 \bar{U}_k(\bar{\chi})}.} \tag{113}$$

The equation (112) fixes the relative scaling of $U_k$ and $\chi$. The previous relation fixes furthermore the absolute scaling, in sense that flows equations must have to be invariant under a global reparametrization. This leads to:

$$U_k[\chi] := \bar{U}_k[\bar{\chi}] k^2 \rho(k^2) \left( \frac{dt}{d\tau} \right)^2. \tag{114}$$

In order to find the appropriate rescaling for $\chi$, we define $\bar{\chi}$ as $\chi = A\bar{\chi}$ for some scale dependent factor $A$. Global invariance imposes:

$$U_k[\chi] := \bar{U}_k[A^{-1}\chi] k^2 \rho(k^2) \left( \frac{dt}{d\tau} \right)^2. \tag{115}$$

Expanding in power of $\chi$ on both sides, we find for the linear term:

$$\partial_\chi U_k(\chi = 0)\chi = \partial_{\bar{\chi}} \bar{U}_k[\bar{\chi} = 0] \bar{\chi} k^2 \rho(k^2) \left( \frac{dt}{d\tau} \right)^2, \tag{116}$$

or, from (112):

$$\partial_\chi U_k(\chi = 0)\chi = \partial_\chi U_k(\chi = 0)\chi A^{-1} \rho(k^2) \left( \frac{dt}{d\tau} \right)^2. \tag{117}$$

Then, assuming $\partial_\chi U_k(\chi = 0)\chi \neq 0$, we obtain finally:

$$A = \rho(k^2)\left(\frac{dt}{d\tau}\right)^2, \tag{118}$$

and:

$$\chi = \rho(k^2)\left(\frac{dt}{d\tau}\right)^2 \bar\chi. \tag{119}$$

This equation in turn fixes the dimension of $\kappa$, which have to be the same as $\chi$. Finally, the flow equations for the different couplings can derived from definitions:

$$\left.\frac{\partial U_k}{\partial \chi}\right|_{\chi=\kappa} = 0, \tag{120}$$

$$\left.\frac{\partial^2 U_k}{\partial \chi^2}\right|_{\chi=\kappa} = u_4(k), \tag{121}$$

$$\left.\frac{\partial^3 U_k}{\partial \chi^3}\right|_{\chi=\kappa} = u_6(k). \tag{122}$$

The first equation means that we require to make the expansion around a local minimum of the effective potential. The two other equations are a consequence of the parametrization for $U_k$. To derive the flow equations for dimensionless couplings, it is suitable to work with a flow equation with fixed $\bar\chi$. The flow equation (113) is however written at fixed $\chi$. To convert one into the other, let us observe that:

$$U_k'[\chi] = \rho(k^2)\left(\frac{dt}{d\tau}\right)^2\left[\bar U_k'[\bar\chi] + \dim_\tau(U_k)\bar U_k[\bar\chi] - \dim_\tau(\chi)\bar\chi\frac{\partial}{\partial\bar\chi}\bar U_k[\bar\chi]\right], \tag{123}$$

where $\dim_\tau(U_k)$ and $\dim_\tau(\chi)$ denote respectively the canonical dimension of $U_k$ and $\chi$. To compute them we return on the definition of dimensionless quantities. Explicitly:

$$\boxed{\dim_\tau(U_k) = t'\frac{d}{dt}\ln\left(k^2\rho(k^2)\left(\frac{dt}{d\tau}\right)^2\right),} \tag{124}$$

and

$$\boxed{\dim_\tau(\chi) = t'\frac{d}{dt}\ln\left(\rho(k^2)\left(\frac{dt}{d\tau}\right)^2\right).} \tag{125}$$

The final expression for the effective potential RG equation then becomes:

$$\bar U_k'[\bar\chi] = -\dim_\tau(U_k)\bar U_k[\bar\chi] + \dim_\tau(\chi)\bar\chi\frac{\partial}{\partial\bar\chi}\bar U_k[\bar\chi] + \frac{1}{1 + \partial_{\bar\chi}\bar U_k(\bar\chi) + 2\bar\chi\partial^2_{\bar\chi}\bar U_k(\bar\chi)}. \tag{126}$$

From this expression it is straightforward to deduce the explicit expressions for coupling constant. Using the definition (120) we have: $\partial_{\bar\chi}\bar U_k'[\bar\chi = \bar\kappa] = -\bar u_4\bar\kappa'$. Therefore, deriving equation (126), we get for $\bar\kappa'$:

$$\bar\kappa' = -\dim_\tau(\chi)\bar\kappa + 2\frac{3 + 2\bar\kappa\frac{\bar u_6}{\bar u_4}}{(1 + 2\bar\kappa\bar u_4)^2}. \tag{127}$$

In the same way, taking second and third derivatives, and from the conditions (121) and (122), we get:

$$\bar u_4' = -\dim_\tau(u_4)\bar u_4 + \dim_\tau(\chi)\bar\kappa\bar u_6 - \frac{10\bar u_6}{(1 + 2\bar\kappa\bar u_4)^2} + 4\frac{(3\bar u_4 + 2\bar\kappa\bar u_6)^2}{(1 + 2\bar\kappa\bar u_4)^3}, \tag{128}$$

and

$$\bar u_6' = -\dim(u_6)\bar u_6 - 12\frac{(3\bar u_4 + 2\bar\kappa\bar u_6)^3}{(1 + 2\bar\kappa\bar u_4)^4} + 40\bar u_6\frac{3\bar u_4 + 2\bar\kappa\bar u_6}{(1 + 2\bar\kappa\bar u_4)^3}. \tag{129}$$

**The flow equation for $\eta$.** We now assume that $\eta(k) \neq 0$. From definition, assuming that $z$ depends only on the value of the vacuum, we have:

$$z[M = \kappa] \equiv \frac{d}{dp^2} \Gamma_k^{(2)}(p, -p) \Big|_{M=\sqrt{2\kappa}}. \tag{130}$$

Therefore:

$$\eta(k) := \frac{1}{z} k \frac{dz}{dk} = \frac{1}{z} \frac{d}{dp^2} \dot{\Gamma}_k^{(2)}(p, -p). \tag{131}$$

The flow equation for $\Gamma_k^{(2)}$ can be deduced from (70), taking the second derivative with respect to the classical field. Because the effective vertex are momentum independent in the LPA, the contributions involving $\Gamma_k^{(4)}$ have to be discarded from the flow equation for $z$. Therefore:

$$\dot{z} := (\Gamma_k^{(3)}(0, 0, 0))^2 \frac{d}{dp^2} \sum_q \dot{r}_k(q^2) G^2(q^2) G((q+p)^2) \Big|_{M=\sqrt{2\kappa}, p=0}, \tag{132}$$

where, according to LPA, we evaluated the right-hand side over uniform configurations. After a few algebras, we can prove the following statement:

**Proposition 1** *The anomalous dimension $\eta(k)$ for $\kappa \neq 0$ is given by:*

$$\eta(k) = 2(t')^{-2} \frac{(3\sqrt{2\bar{\kappa}}\bar{u}_4 + (2\bar{\kappa})^{3/2}\bar{u}_6)^2}{(1 + 2\bar{\kappa}\bar{u}_4)^4}. \tag{133}$$

**Proof.** We have $G_k(p, p') =: G_k(p)\delta(p+p')$ is the inverse of $\Gamma_k^{(2)}(p, p') + r_k(p^2)\delta(p+p')$, with $\Gamma_k^{(2)}$ given by equation (104). The expression of $\Gamma_k^{(3)}(0, 0, 0)$ can be easily obtained; taking the third derivative of the effective potential for $M$:

$$\Gamma_k^{(3)}(0, 0, 0) = 3u_4\sqrt{2\kappa} + u_6(2\kappa)^{3/2}. \tag{134}$$

Using the modified Litim regulator, we get:

$$\dot{r}_k(p^2) = \eta(k)r_k(p^2) + 2zk^2\theta(k^2 - p^2), \tag{135}$$

and

$$\frac{d}{dp^2} r_k(p^2) = -z\theta(k^2 - p^2). \tag{136}$$

In the LPA$'$, the diagonal components of the effective propagator take the form:

$$G_k(p^2) = \frac{1}{zp^2 + z(k^2 - p^2)\theta(k^2 - p^2) + \mathcal{M}^2(g, h, \kappa)}, \tag{137}$$

where $\mathcal{M}^2$ denotes the effective mass, i.e. the second derivative of the effective potential. Finally, we have to compute integrals like

$$I_n(k, p) = \int_{-k}^{k} \rho(q^2) q(q^2)^n dq \, G_k((p+q)^2). \tag{138}$$

We focus on small and positive $p$. In that way the integral decomposes as $I_n(k, p) = I_n^{(+)}(k, p) + I_n^{(-)}(k, p)$, where:

$$I_n^{(\pm)}(k, p) = \pm \int_0^{\pm k} \rho(q^2) q(q^2)^n dq \, G_k((p+q)^2). \tag{139}$$

Since $p > 0$, in the negative branch, $(q + p)^2 < k^2$, and:

$$I_n^{(-)}(k, p) = \frac{1}{zk^2 + \mathcal{M}^2} \times \int_{-k}^{0} \rho(q^2) q(q^2)^n dq, \tag{140}$$

which is independent of $p$. In the positive branch however:

$$I_n^{(+)}(k, p) = \frac{1}{zk^2 + \mathcal{M}^2} \int_0^{k-p} \rho(q^2) q(q^2)^n dq + \int_{k-p}^{k} \rho(q^2) q(q^2)^n dq \frac{1}{z(q + p)^2 + \mathcal{M}^2}. \tag{141}$$

Hence, taking the first derivative with respect to $p$, we get:

$$\frac{d}{dp} I_n^{(+)}(k, p) = -\frac{1}{zk^2 + \mathcal{M}^2} \rho(q^2) q(q^2)^n |_{q=k-p} + \rho(q^2) q(q^2)^n dq \frac{1}{z(q + p)^2 + \mathcal{M}^2} |_{q=k-p}$$
$$- 2z \int_{k-p}^{k} \rho(q^2) q(q^2)^n dq \frac{(q + p)}{(z(q + p)^2 + \mathcal{M}^2)^2}.$$

The first two terms cancel exactly, and then:

$$\frac{d}{dp} I_n^{(+)}(k, 0) = -2z \int_{k-p}^{k} \rho(q^2) q(q^2)^n dq \frac{(q + p)}{(z(q + p)^2 + \mathcal{M}^2)^2}. \tag{142}$$

To conclude, taking second derivative and setting $p = 0$, we obtain:

$$\frac{1}{2} \frac{d^2}{dp^2} I_n(k, 0) = -\frac{z\rho(k^2)(k^2)^{n+1}}{(zk^2 + \mathcal{M}^2)^2} =: I_n''(k, 0). \tag{143}$$

Therefore:

$$z\eta(k) = \frac{(3u_4\sqrt{2\kappa} + u_6(2\kappa)^{3/2})^2}{(zk^2 + \mathcal{M}^2)^2} \left(2zk^2 I_0''(k, 0) + z\eta(k)(k^2 I_0''(k, 0) - I_1''(k, 0))\right).$$

To introduce $\tau$-dimensionless quantities, we have to remark that both $u_4\kappa$ and $u_6\kappa^2$ have the same scaling dimension. Finally, using renormalized and dimensionless quantities $\bar{u}_{2p}$, and replacing the effective mass by its value:

$$\bar{\mathcal{M}}^2 = \partial_{\bar{\chi}} \bar{U}_k(\bar{\kappa}) + 2\bar{\kappa} \partial_{\bar{\chi}}^2 \bar{U}_k(\bar{\kappa}) = 2\bar{\kappa}\bar{u}_4, \tag{144}$$

we arrive to the expression (133). □

Note that to derive this expression we took into account the additional rescaling coming from $z$ accordingly to the requirement that the coefficient in front of $p^2$ in the kinetic action remains equals to 1. This in particular implies to change $\bar{\kappa} \to z^{-1}\bar{\kappa}$ with respect to the strict LPA definition. Due to the factors $z$ in the definition of barred quantities, $\eta(k)$ appears in the flow equations. The net result is a translation of canonical dimensions

$$\dim_\tau(u_{2n}) \to \dim_\tau(u_{2n}) - n\frac{dt}{d\tau}\eta(k), \tag{145}$$

in the equations obtained previously for $z = 1$. We moreover have to take into account the additional contribution coming from the derivative of the regulator.

# 5 Investigations for standard models of noise

In this section we will apply the formalism presented in the previous section to concrete situations, considering a series of spectra in the neighbourhoods of common universal noise models. We will start by a review of the vicinity of the universality class corresponding to the MP spectrum, studied in references [52–55]. We will then consider the case of Wigner's universality class. Although this distribution is not positive, we will be able to make it positive by performing a translation on the spectrum, arranging to place the signal in the tail of the spectrum, on the positive side. Finally, we will consider the case of a noise corresponding to data materialized by a tensor and not a matrix. This less-common universality class corresponds to the tensor PCA, more widely studied in the last few years [61–64]. The tensor case will also allow us to confront our methods to the case where we do not have an analytical formula for the eigenvalues of the covariance matrices. Moreover, we will see that the definition of the covariance matrix is not unique. The theory of random tensors [102, 103] proposes many of them and we will be able to build an effective criterion to decide which are the best. Note that the simplest definition, based on the simplest of melonic graphs has been studied previously [55]. Notwithstanding the fact that some results have been previously studied, the investigations presented in this following section (including MP) go far beyond our previous studies.

## 5.1 Empirical methodology

If proposing new detection algorithms for quasi-continuous spectra where standard methods fail to give satisfactory results is part of our long term goals, it would be difficult to draw clear conclusions from this kind of analysis without having understood the properties of the RG stream of reference spectra beforehand. Indeed, our paradigm replaces the search for principal components of the standard PCA by that of principal flows, relevant in the IR. We could then speak of *principal flow analysis* (PFA). But to be able to identify the principal flow of a type of spectrum and to say with certainty that the characteristics of this flow are compatible with the presence of a signal requires a prior understanding of the expected properties of this flow. Note that this is not specific to our study but a reflection of the general approach in physics. Physics experiments are performed in such a way as to keep a precise control on the different parameters of the experiment. We propose here such an approach. We will build spectra by corrupting a signal made of a deterministic matrix or tensor and normalized by a certain rank by a random matrix or tensor materializing the noise. The signal will be weighted by a parameter $\beta \in [0, 1]$, interpolating between a regime without signal and a regime of strong signal. We will only work with the Gaussian set, and normalize all our distributions so that the variance is the same for all components. Thus we will keep a precise control on the numerical parameters, the characteristics of the distribution and the strength of the signal that we can vary. In that way we can able to obtain empirical statement about general properties of spectra in vicinity of universal class for matrices or tensors.

## 5.2 Marchenko-Pastur universality class

In this section, we will first discuss numerical analyses concerning spectra in the neighbourhood of the universality class of MP. This case has been extensively studied in the papers cited in reference [52–54], so this section takes up most of their material.

In section 4, we illustrated the dependence of the canonical dimensions on the scale, for an MP distribution, and emphasized two points. The first point is that at a large scale only two couplings are relevant, the quartic and the sextic, the latter tending to be asymptotically marginal. The second point is the appearance of a dimensional crisis around the first third of the spectrum. From this scale and going towards more and more ultraviolet scales, the num-

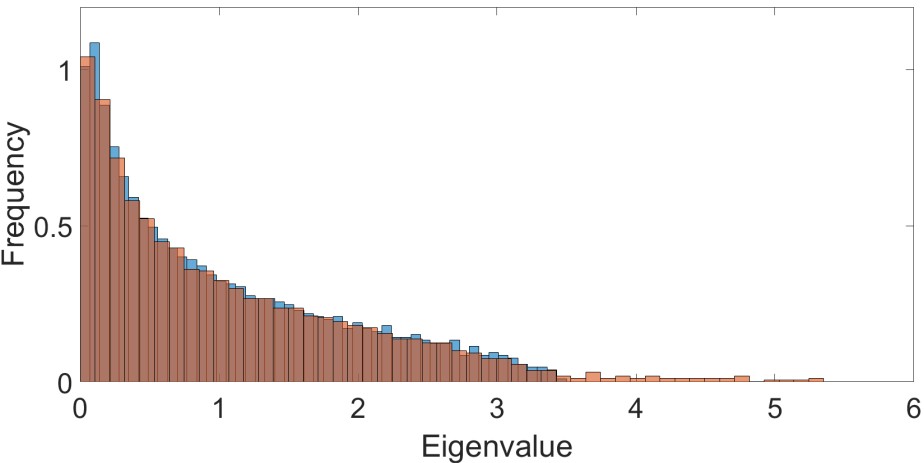

Figure 11: Blue histogram: Typical spectrum for a 2000×1500 white Wishart matrix. Brown histogram: Perturbation with a deterministic matrix of rank 50.

ber of marginal operators as well as their dimensions grow uncontrollably (see Figure 10). This observation constitutes one of the first pieces of the empirical statement 1, and can be verified on real spectra, at large but finite $N$ and $P$, following the conventions given in the section 5.1. Such a spectrum corresponds to the histogram in blue on Figure 11, for $N = 2000$ and $P = 1500$, with a standard deviation fixed at 1. On the same figure, we can observe the blue histogram obtained by taking another matrix of the same statistical set, and by adding a deterministic matrix $m$ of rank 50, whose normalized column vectors materialize a signal. Figure 12 shows the behavior of the canonical dimension in both cases for the quartic coupling (see equation (85)). The RG thus allows establishing a first criterion indicating the presence of a signal in a spectrum in the vicinity of the Gaussian point. For a noisy signal, the quartic and sextic couplings will be relevant, and the theory will be essentially interactive. On the contrary, when a signal is input, the couplings will tend to become irrelevant, and the Gaussian fixed point will become stable. This situation is reminiscent of the standard theory of ferromagnetism, where the Gaussian theory is stable in dimension $> 4$, and becomes unstable in dimension $< 4$ [2]. Here, it is the shape of the distribution that replaces the dimension, and we will say that a dimension becomes critical when it approaches a power law $\rho(p^2) \sim (p^2)^\delta$ with $\delta = 1$ (see definition 3). Such a difference in behavior makes the asymptotic states distinguishable, and establishes a simple detection criterion:

**Empirical statement 2** *The emergence of a signal in a data set around a white Wishart ensemble corresponds to a critical behavior.*

We will see in the following that this statement is not limited to Wishart-type white noise.

**Remark 2** *Note that all these simulations take into account the warning concerning the crisis of the dimension, around the eigenvalue $\lambda \sim \lambda_+/3$ (see the discussion after equation (89)). For the concrete case we study, this means that we consider our field theory approximation valid between the eigenvalue $\sim 2.5$ and the largest eigenvalue $\sim 3.4$.*

Unlike in ordinary field theories, the canonical dimensions are scale-dependent and therefore the flow equations never form an autonomous system. This implies in particular that, in this context, it cannot exist true fixed points of the RG flow, i.e. points where all the $\beta$ functions (see equations (98), (101) and (102)) vanish exactly. However, it is instructive to plot the flow numerically. Figure 13 shows the typical behavior of the RG flow corresponding to blue and

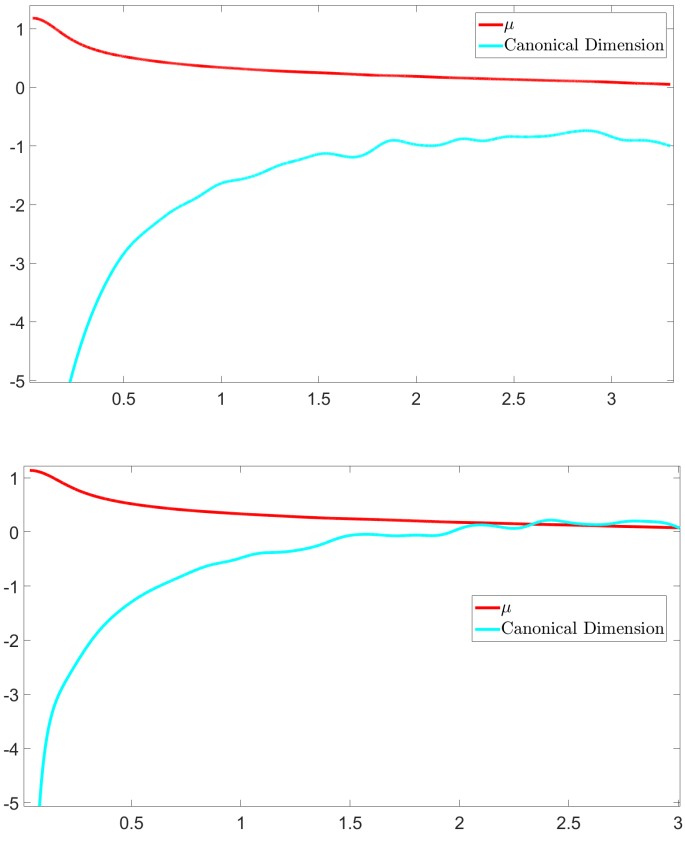

Figure 12: Blue line: Canonical dimension $\dim_\tau(u_4)$ for blue histogram without signal (on the top) and for the brown histogram with signal (on the bottom). Red curve is the empirical eigenvalue distribution in both cases.

brown histograms of Figure 11. On the figure are represented the flows associated respectively with the blue and brown histograms of Figure 11. What is striking at first is that, although there is no real fixed point, there is nevertheless a region that behaves "almost" like a Wilson-Fisher fixed point, separating the stream into two regions. These two figures illustrate the important point of our discussion. The asymptotic behavior of some trajectories is likely to vary in the presence of a sufficiently strong signal. Around the Gaussian fixed point, the trajectories have two outcomes. Either they gain the region $u_2 > 0$, and the $\mathbb{Z}_2$ symmetry is restored in the IR, or they reach the region $u_2 < 0$ and the symmetry is broken in the IR. In a given truncation, we can search numerically the set of trajectories which, for some initial conditions around the Gaussian fixed point, end in the symmetric phase ($u_2 > 0$) in the IR. For a purely noisy dataset, and using the sextic truncations given by equations (98), (101) and (102), we typically obtain the Figure 14. The set of initial conditions leading to the symmetric phase forms a compact domain around the Gaussian point (in purple), which we will call $\mathcal{R}_0$.

**Remark 3** *The reader should keep in mind that, since the flow is not described by an autonomous system, a global fixed point cannot exist. However, there can be "fixed trajectories" along which the beta functions vanish. Asymptotically, as the theory behaves like a 3D quantum field theory, these lines act similarly to an ordinary fixed point, but only in the asymptotic limit. The region discussed in this paragraph, which behaves "like" a Wilson-Fisher fixed point, represents the endpoint of such a trajectory.*

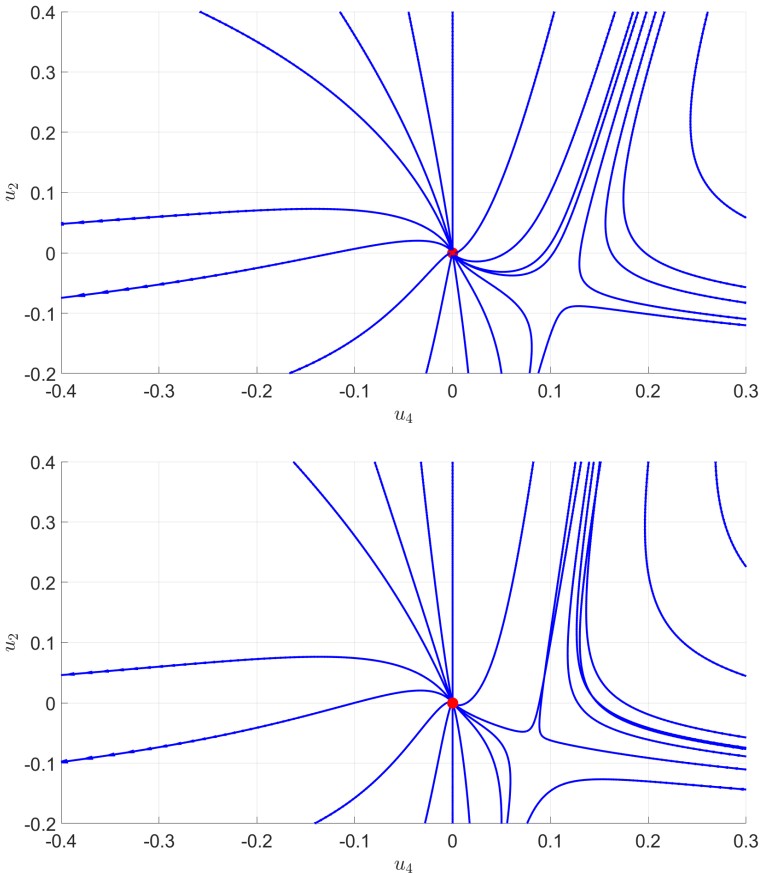

Figure 13: Behavior of the RG flow in the vicinity of the Gaussian fixed point. On the top: For a purely noisy dataset (blue histogram). On the bottom: with a deterministic signal (brown histogram).

We can now investigate what happens if we add a signal, that is if we consider the blue histogram rather than the blue histogram in Figure 11. The result is shown in Figure 15. We observe a reduction in the size of the $\mathcal{R}_0$ region. In other words, some trajectories that used to end up in the symmetric phase now end up in the non-symmetric phase. The exploration of this phase renders obsolete the development that led to the equations (98), (101) and (102); and the local potential formalism lends itself better to this kind of investigation.

By limiting ourselves again to a truncation of order 6, i.e. for a potential of the form (103); we can follow the evolution of the $\mathcal{R}_0$ zone but also of the $\kappa$ vacuum and the effective potential. These results are summarized in Figures 16, 17 and 18. To realize the Figure 16, we multiplied the signal (materialized by the deterministic matrix $m$) by a factor $\beta \in [0, 1]$, continuously interpolating between the blue ($\beta = 0$) and brown ($\beta = 1$) histograms of Figure 11. An illustration of how the size of the $\mathcal{R}_0$ region reduces with signal strength can be seeing in this figure. The more $\beta$ increases, the more the area shrinks. Note that to obtain this figure we used the effective potential formalism and that the axes correspond respectively to $\kappa$, $u_4$ and $u_6$; $\kappa$ being the running expectation value for the classical field.

Figure 17 shows the evolution of the effective IR potential for a typical trajectory taking its initial conditions in the $\mathcal{R}_0$ region. When $\beta = 0$, the $\mathbb{Z}_2$ symmetry persists in the IR. But as $\beta$ increases, the shape of the potential changes, the $\kappa = 0$ vacuum becomes unstable and two stable non-zero voids emerge. This situation is again evocative of the physics of critical phenomena, the value of $\beta$ playing the role played by the inverse of the temperature ($\beta \equiv 1/T$).

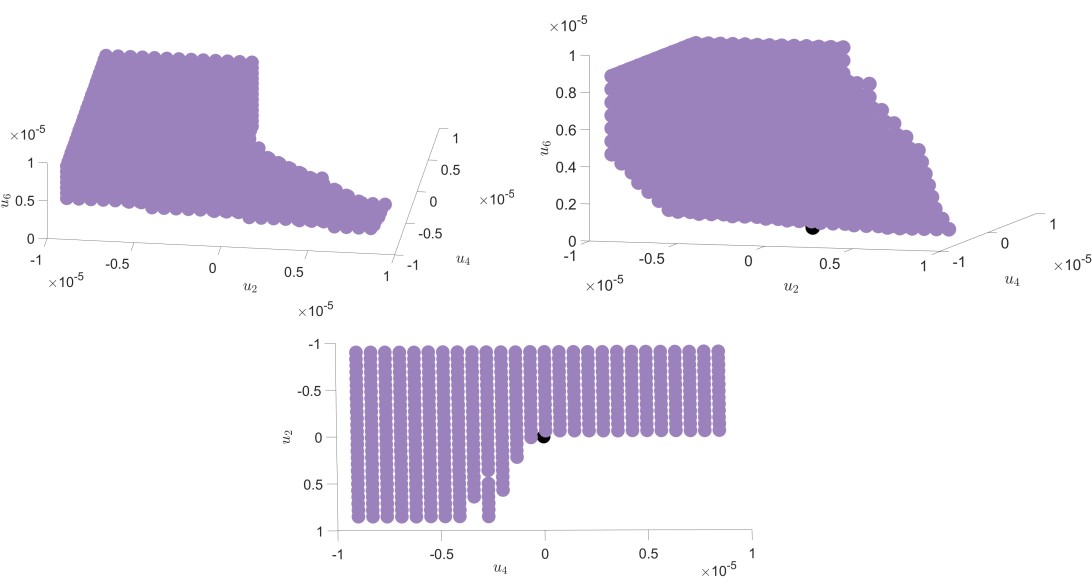

Figure 14: Three different points of view on the region $\mathcal{R}_0$ for the blue histogram around the Gaussian point (materialized by the black point) for a local sextic truncation.

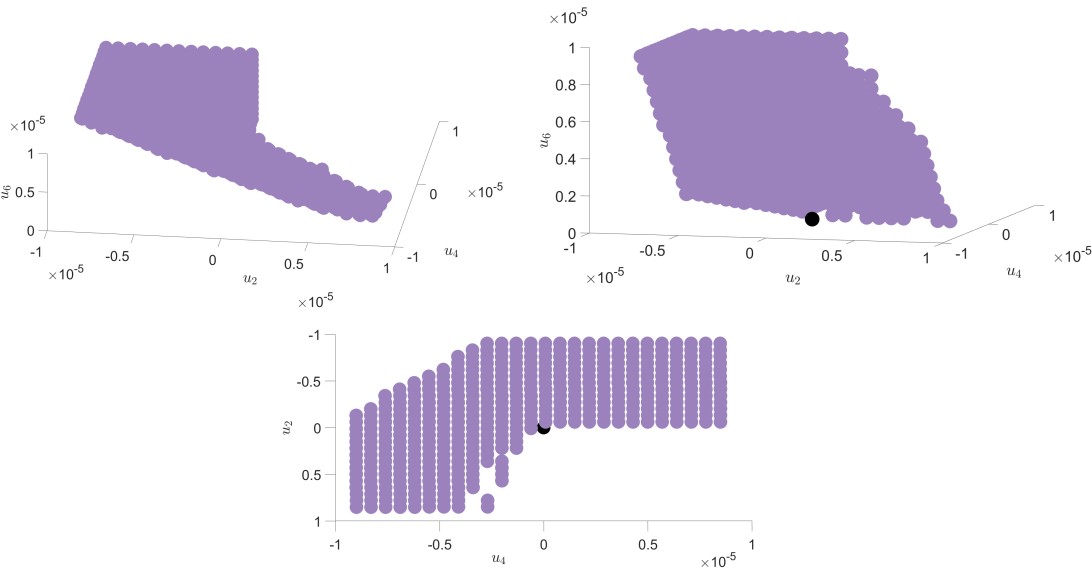

Figure 15: Three different points of view on the region $\mathcal{R}_0$ for the blue histogram around the Gaussian point (materialized by the black point) for a local sextic truncation.

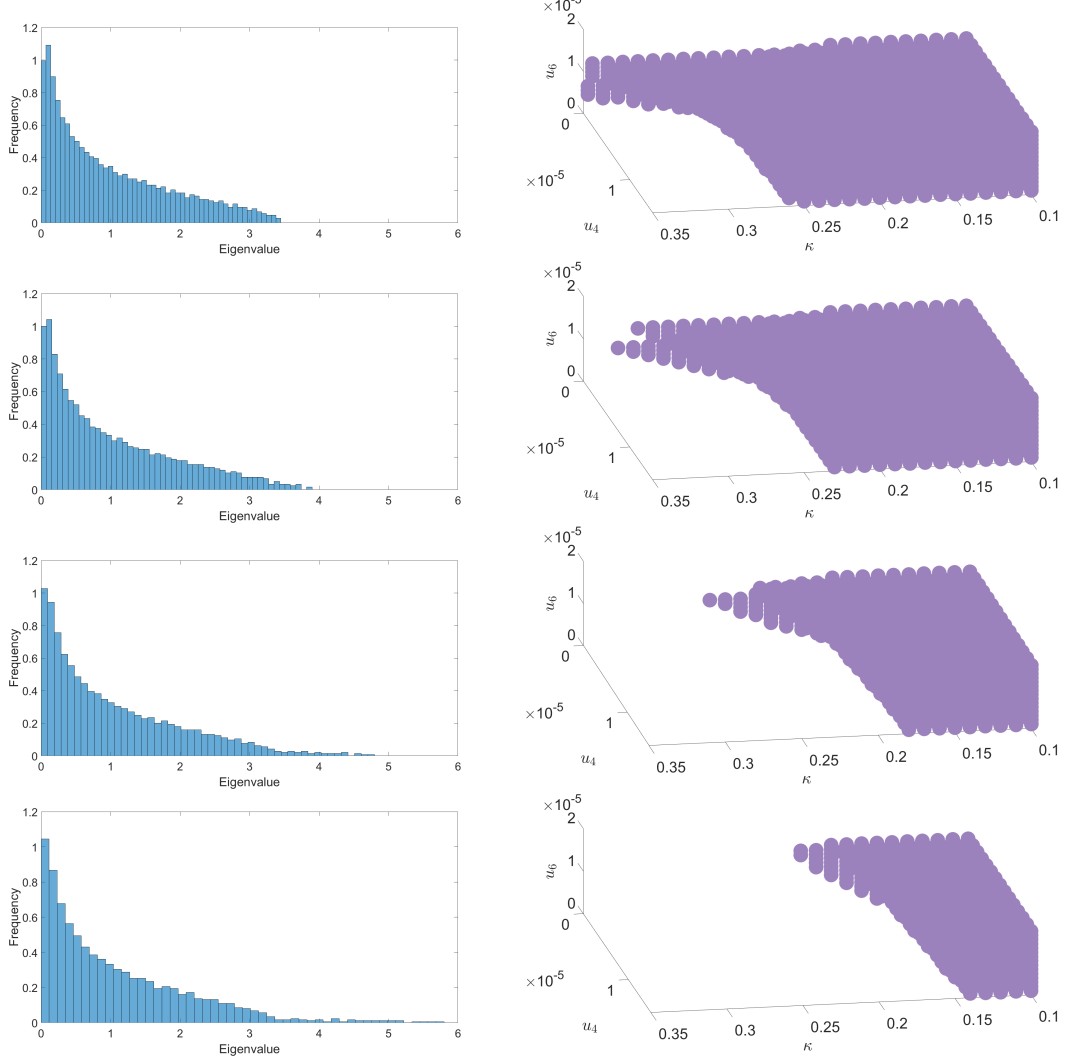

Figure 16: On the left: A series of spectra obtained by varying the signal strength $\beta$. From top to bottom, $\beta = 0, 0.4, 0.7, 1$. On the right: the $\mathcal{R}_0$ area in the $(\kappa, u_4, u_6)$ truncation of the effective potential $U_k[\chi]$.

Finally, Figure 18 illustrates the expected behavior for $\kappa(k)$ along typical trajectories, going towards a broken or asymptotically restored symmetry. In this respect, it is worth observing that the trajectories leading to the restoration do not just converge to zero, but become negative. This effect indicates that the potential does not cancel at zero, which can be seen in Figure 17. Finally, to conclude, these numerical analyses assume the validity of the LPA; which is generally questionable. In particular, this approximation assumes that the anomalous dimension plays a negligible role. We have discussed a formalism that takes into account the anomalous dimension in the section 4, and we can indeed verify that taking into account these effects leads only to tiny deviations from the predictions of the LPA. Figure 19 illustrating that the expected values for $\eta$ are all $\ll 1$.

We have been able to illustrate almost the whole proposition 1. However, we still need to clarify an important point. These investigations seem to suggest that formalism would allow for the detection of the slightest presence of a signal in a spectrum. But as it is known, there is always a threshold effect and we should be able to understand the existence of such a threshold with our theory.

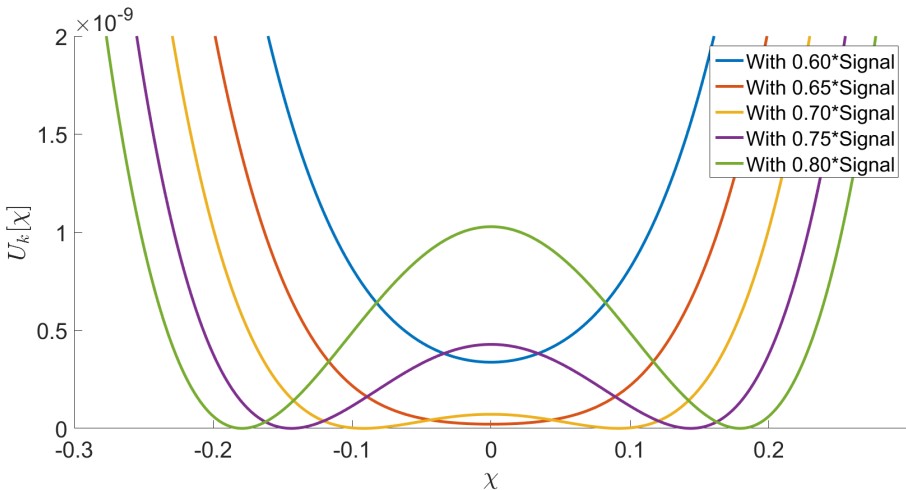

Figure 17: Shape of the IR effective potential for different values of the signal strength $\beta$.

To this end, we need to clarify a little what we mean by "IR potential" in the Figure 17. Generally, the deep IR corresponds to the $k \to 0$ limit. Here, however, we have to take care that the "zero" modes must be excluded from the partial integration procedure (see discussion at the end of the section 4.1).

More precisely, we expect to obtain an effective theory for the "zero" modes, and the partial integration procedure should stop just before. Since the eigenvalue spacing is of the order of $1/N$, the smallest value of $k^2$ must be $\sim 1/N$ and not exactly 0. So when we talk about IR potential, we must understand $\bar{U}_k$, evaluated for $k \sim 1/\sqrt{N}$. This observation allows fixing the typical size of the mass for a phase transition to occur. Indeed, it is not enough that $\bar{u}_2$ reaches a negative value for a transition to occur. $|u_2|$ must have a finite value. Because $u_2 = k^2 \bar{u}_2$, if $\bar{u}_2$ tends to a negative but finite value, $u_2$ could vanish in the limit $k \to 0$. In our case, $k$ is always finite, but $1/N$ is assumed to be a very small number, and $|\bar{u}_2|$ has to be very large, of order $N$, so that $|u_2|$ has a finite value. Similarly in the symmetric phase, the mass being interpreted as the inverse of the largest eigenvalue in the IR, its value must be finite for arbitrarily large values of $N$, and we must select in the purple region $\mathcal{R}_0$ those initial conditions which respect this constraint. We can easily show that there are such trajectories (see Figure 20). We can search, among the trajectories of the region $\mathcal{R}_0$ those which have this characteristic. What we obtain is represented in the Figure 21. We will say that these trajectories are physical, and we will denote by $r_0 \subset \mathcal{R}_0$ this subset of the physical initial conditions.

This physical region allows us to consider the existence of a detection threshold from another angle. We have seen that when $\beta$ increases, the size of the $\mathcal{R}_0$ region decreases. However, as long as this shrinkage does not reach the sub-region $r_0$, the physical states remain insensitive to the presence of the signal. It is only when this region is reached that the physical asymptotic states are altered. Hence, the RG allows us to simply understand the existence of an intrinsic detection threshold. This would deserve to be refined. One should for instance take into account the global precision of the measurement system, and introduce error bars systematically. Nevertheless, the final conclusions would remain qualitatively the same, i.e. that there is an intrinsic limit to the data, a finite domain between blue and purple regions, which the RG "perceives".

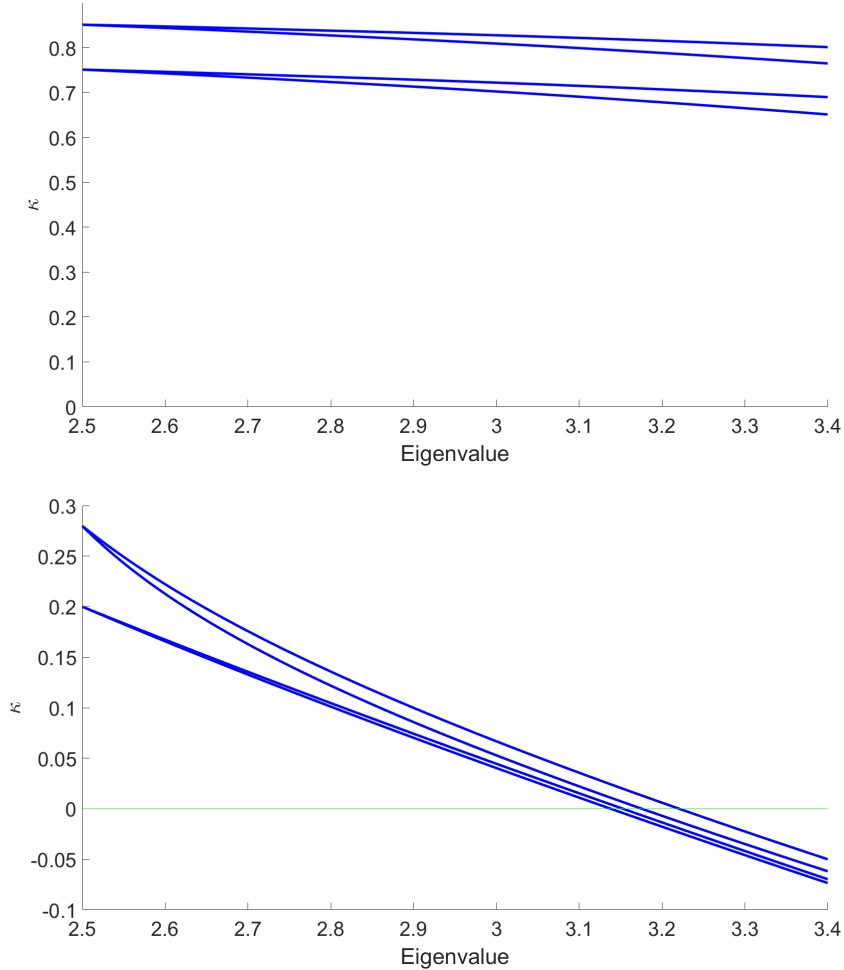

Figure 18: Illustration of the evolution of $\kappa$. On the top: For some RG trajectories (on the left), $\kappa$ decreases toward a negative value, which corresponds to a restoration of the $\mathbb{Z}_2$-symmetry. On the bottom: For other trajectories however, $\kappa$ stays almost constant in the range of eigenvalues that we consider, and does not lead to a restoration of the symmetry.

## 5.3 Wigner's universality class

In this section we want to highlight the universality of the proposed framework. For this, we illustrate that the results presented in the previous section related to the MP law are still true for other noise models. In this section, we focus on the signal detection around the well known Wigner's law. In Figure 22 we show the typical spectrum that we analysed with the proposed framework. In the left, we show an spectrum of a large random symmetric matrix with Gaussian entries that for which the distribution converges to the Wigner's law. We consider this case as a reference spectrum associated to data with only noise. Then we build a matrix data which can be regarded as a disturbance of this reference data in the sense that we added to it a matrix of rank 50 that we consider as a signal. The spectrum of such matrix is illustrated on the right side of this Figure 22.

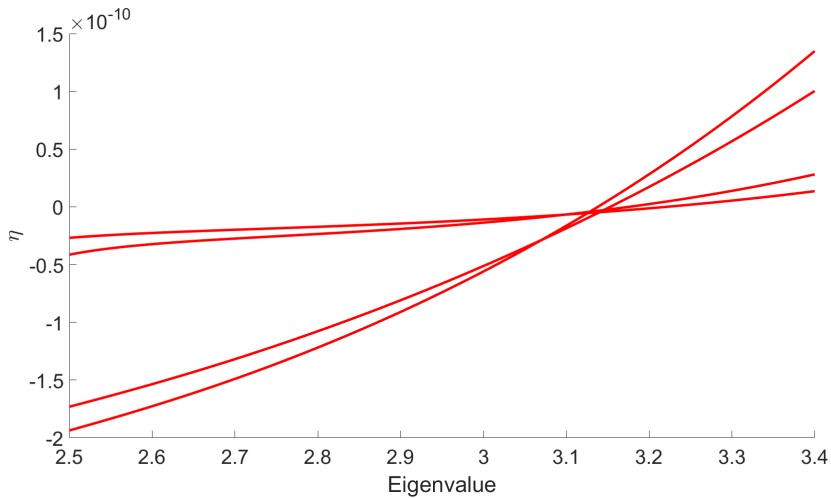

Figure 19: Evolution of the anomalous dimension.

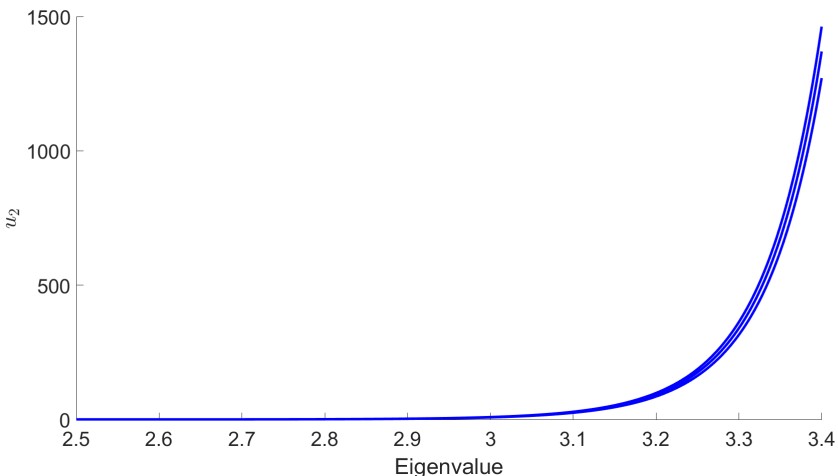

Figure 20: Illustration of the evolution of the $u_2$ for eigenvalues between 2.5 and 3.4 in the case of pure noisy data. We can see that the values of $u2$ for these examples are of the same magnitude as $N = 2000$.

The first main observations are related to the canonical dimensions illustrated in Figure 23 for large but finite matrices. In contrast to the MP case, the relevant sector is more sensitive to the size of the matrix, and for finite $N$, it is more generally spanned by the first three even local couplings[28] ($u_4$, $u_6$ and $u_8$). Beyond those interactions, all the couplings are irrelevant. However, as a universal feature of our framework we can see in these canonical dimensions that all these couplings tends to be shifted to the top which means that they tends to be irrelevant.

The second main observation is related to the behavior of the RG trajectories. Figure 24, shows the numerical behaviour of the quartic truncation, on the top, for the data without signal and on the bottom, for the data with a signal. As for the MP case, we observe the existence of a region analogous to a Wilson-Fisher fixed point (even if it is not a true fixed point). When, we add a signal (on the bottom) we can see clearly, that the behaviour of the global flow is affected: the effective Wilson-Fisher region is shifted toward the Gaussian fixed point (illustrated in red) and the size of the symmetric phase is reduced.

---

[28]The octic coupling becoming irrelevant as we goes toward the analytic form.

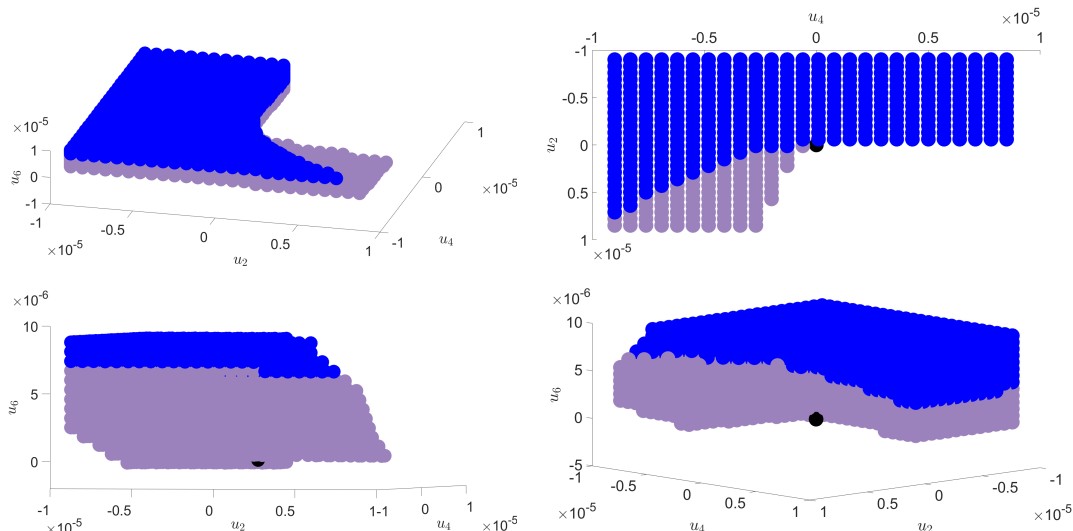

Figure 21: The physical region $r_0 \subset \mathcal{R}_0$. All trajectories starting from this region end with a mass of order 1.

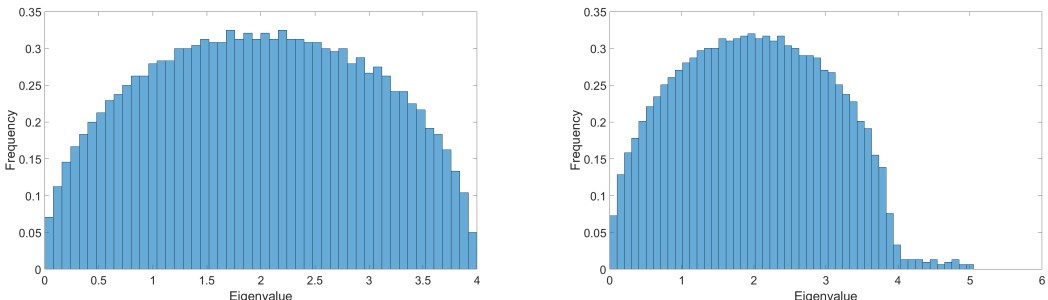

Figure 22: Left histogram: Typical spectrum for a $3000 \times 3000$ symmetric matrix with random entries following the Gaussian distribution. Right histogram: Perturbation with a deterministic matrix of rank 50.

Finally, the Figure 25 shows the compact region associated to the RG trajectories for which we have a symmetry restoration in the IR in the case of a sextic truncation using LPA. Again, the presence of a signal is indicated by a symmetry breaking for some trajectories taking their initial conditions in the symmetric phase of the case of pure noise.

## 5.4 Tensorial universality classes

In this section, we focus on the most difficult issue where the data is tensorial. Mathematically we mean that dataset have to be materialized not as a $P \times N$ matrix $X = \{X_{ai}\}$ as it was the case before, but as a tensor $P_1 \times \cdots P_{d-1} \times N$, of rank $d$: $\mathbf{T} = \{T_{a_1 \cdots a_{d-1} i}\}$. The tensor generalization of PCA has been considered for some years [64] in an attempt to generalize the result of [68] for the spike matrix model through the basic equation:

$$T_{ijk} = \beta v_i v_j v_k + \frac{1}{\sqrt{N}} X_{ijk} \,, \tag{146}$$

for some purely Gaussian noise $X$ being a $N^3$ tensor with entries $X_{ijk} \sim \mathcal{N}(0,1)$ and $v_i \in \mathbb{S}_{N-1}$ is a deterministic unit vector. This equation aims to generalize the one-spike matrix model (1).

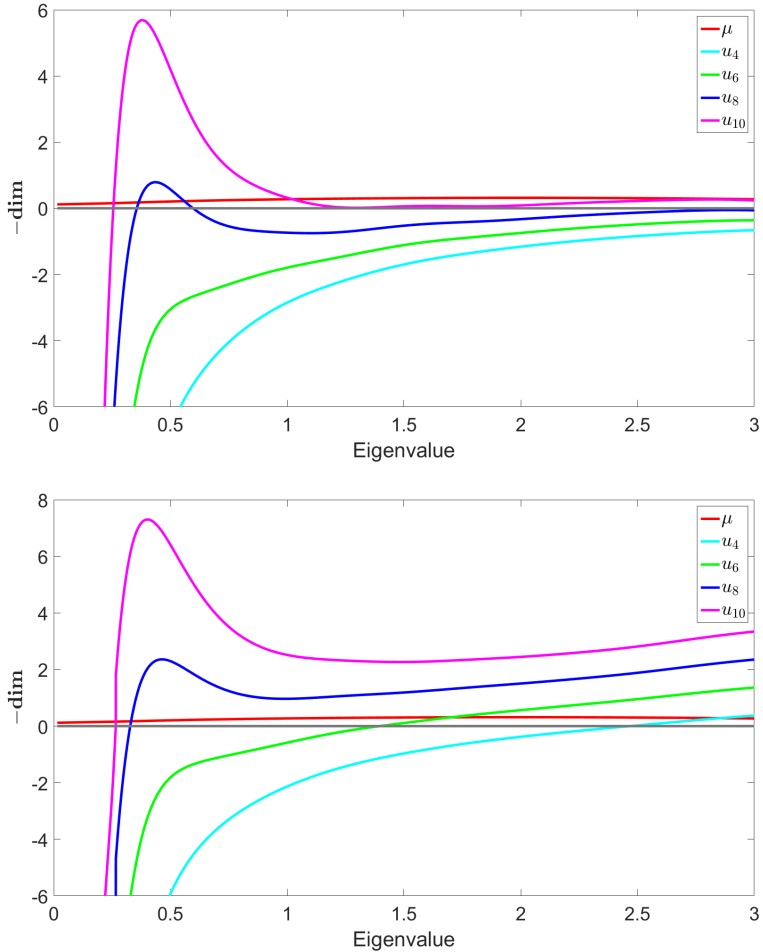

Figure 23: Top: Canonical dimensions related to $u_4$, $u_6$, $u_8$ and $u_{10}$ for the numerical Wigner spectrum of the data without signal. Bottom: Canonical dimensions in the same range of eigenvalues for the spectrum of the data with a signal.

The recent results of [64, 104–106] seem to indicate that a phase transition, similar to the one observed in the matrix case, also occurs in the tensor case. Hence, as for the matrix, there exists a critical value $\beta_c$ of order 1, such that it is impossible to detect or recover a signal below it. For matrices, recovering use *maximum likelihood* (ML) estimator, which is equivalent to computing the largest eigenvector. For tensors, unfortunately, the computation of the ML estimator is NP-Hard. In practice, however, what is relevant is the *size of the signal* from which detection is allowed from detection algorithms in a polynomial time. This value $\beta_0$ for symmetric tensors depends non-trivially on $N$, the size of the tensor, $\beta_0 \sim N^\gamma$. Generally for rank 3 tensors, $\gamma = 1/4$ or $1/2$, depending on the used algorithm to estimate the empirical threshold. Until recently moreover $\gamma \gtrsim 1/4$ looks like an algorithmic lower bound. Note that a recent algorithm named SMPI for *Selective Multiple Power Iteration* [107] (appeared during the redaction of this paper) promises to outperform these algorithms, providing empirically $\beta_0$ of order 1.

In the large $N$ limit, this poses a computational issue, as signal detection and recognition of low-rank signal using algorithmic methods is possible in a polynomial time for $\beta \gtrsim \mathcal{O}(N^\gamma)$. The tensor case is therefore a privileged topic of investigation for alternative approaches like ours. We can mention the recent results [106], which extend the results obtained in the matrix case for the spike model, from a spectral point of view, for random tensors.

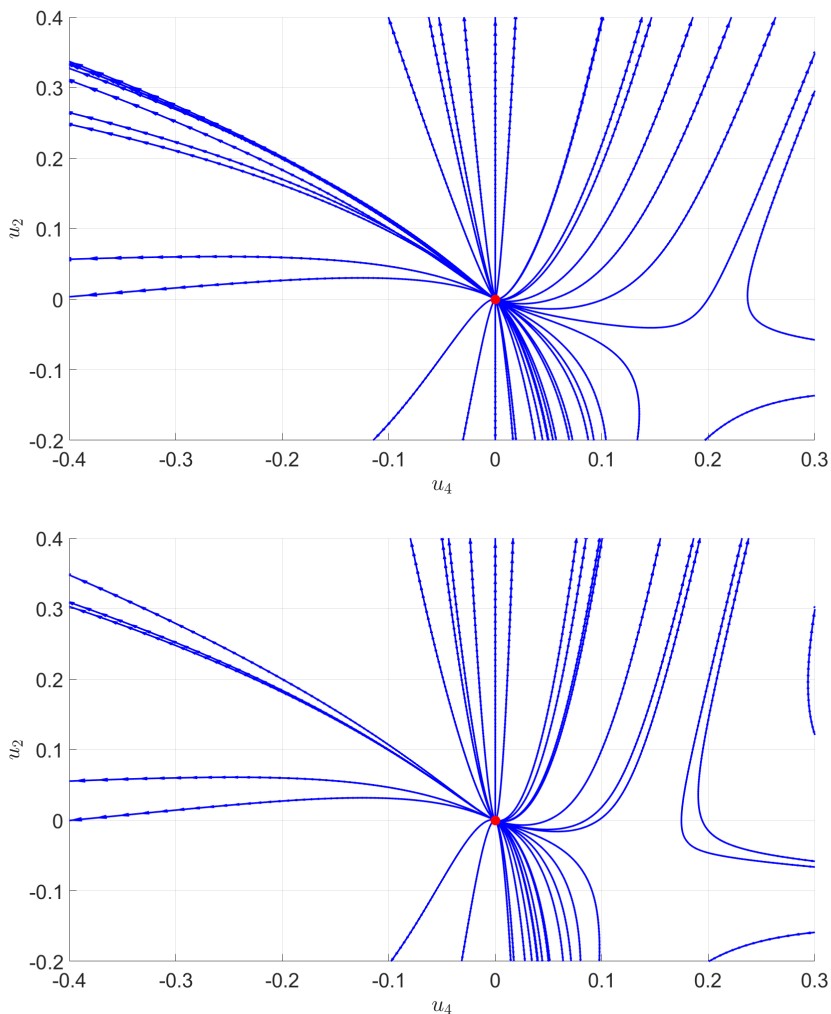

Figure 24: Behavior of the RG flow in the vicinity of the Gaussian fixed point for the case of the Wigner's law for a quartic truncation. On the top: for the spectrum of the data without signal. On the bottom: for the spectrum of the data with a signal.

The theory of random tensors has developed considerably in recent years in the context of quantum gravity [102], and we will largely rely on this formalism, which exploits the notion of tensor invariants. Let us note that the authors of the references [104, 105] have exploited this formalism, which has proven to be very efficient.

### 5.4.1 A digest of random tensor formalism

In this section introduce the notations and definitions usually considered for random tensor models [102]. Let $\mathcal{E}_M(\mathbb{R})$ a $M$ dimensional real vector space: $u \in \mathcal{E}_N(\mathbb{R}) \Rightarrow u = \{u_1, \cdots, u_M\} \in \mathbb{R}^M$. It is equipped with the standard Euclidean scalar product $\langle , \rangle$ defined as:

$$\forall u, v \in \mathcal{E}_M(\mathbb{R}) \rightarrow \langle u, v \rangle := \sum_{n=1}^{M} u_n v_n. \tag{147}$$

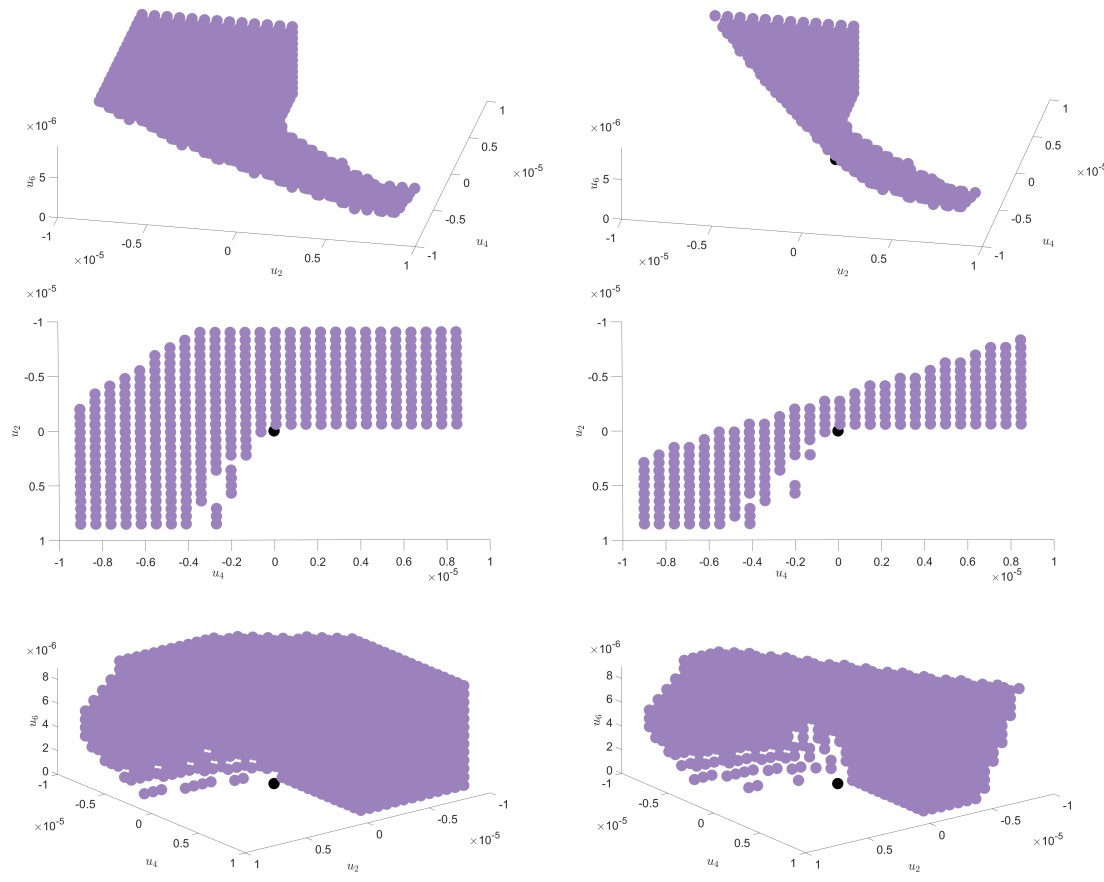

Figure 25: Three different points of view on the region $R_0$ around the Gaussian point (materialized by the black point) for a local sextic truncation. On the left: for the spectrum of the data without signal. On the right: for the spectrum of the data with a signal.

A $P_1 \times \cdots P_{d-1} \times N$ rank $d$ tensor $\mathbf{T} = \{T_{a_1 \cdots a_{d-1} i} \in \mathbb{R}\}$ is a multi-linear form belong $\mathcal{E}_{P_1}(\mathbb{R}) \otimes \cdots \otimes \mathcal{E}_{P_{d-1}}(\mathbb{R}) \otimes \mathcal{E}_N(\mathbb{R})$. For a pair of tensors $\mathbf{T}_1$ and $\mathbf{T}_2$, we can define an inner product, inherited from the Euclidean product over $\mathcal{E}_M(\mathbb{R})$:

$$(\mathbf{T}_1, \mathbf{T}_2) \rightarrow \langle \mathbf{T}_1, \mathbf{T}_2 \rangle := \sum_{k_1=1}^{P_1} \cdots \sum_{k_{d-1}=1}^{P_{d-1}} \sum_{k_d=1}^{N} (T_1)_{k_1 \cdots k_d} (T_2)_{k_1 \cdots k_d}. \tag{148}$$

Tensor $\mathbf{T}$ transforms naturally under rotations:

$$T_{k_1 \cdots d_d} \rightarrow T'_{k1 \cdots k_d} = \sum_{k_1, \cdots, k_d} \prod_{j=1}^{d} \mathcal{O}^{(j)}_{k_j l_j} T_{l_1 \cdots l_d}, \tag{149}$$

where $O^{(j)} \in \mathrm{O}(P_j)$. Such a quantity, which is invariant under transformation (149), is said to be a *tensorial invariant*. The concept of tensorial invariant can be extended for quantities involving a larger number of tensors, provided that the index $k_i$ of a given tensor is contracted with the index $l_i$ of an other tensor. We call *color* the number $i$ indexing $k_i$. Obviously this construction implies that the number on tensors must be even. Tensorial invariants can be pictured as $d$-colored regular graphs as follows. To each tensor we associate a *black node*, with

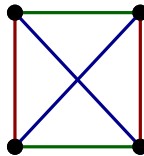
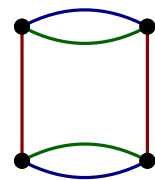
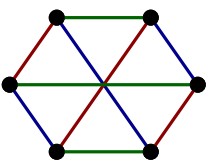
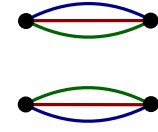

Figure 26: Examples of tensorial invariants for $d = 3$. The last one on the right is non connected.

$d$ colored half edges hooked to it:

$$T_{k_1 k_2 \cdots k_d} \implies \begin{array}{c} k_1 \\ k_2 \\ k_3 \end{array} \!\!\!\!\!\!\!\!\!\!\!\!\! k_d . \tag{150}$$

A tensor invariant made of $2P$ tensors can then be constructed by connecting the lines in pairs according to their colors. Figure 26 provides elementary examples of tensor invariants for $d = 3$. A tensorial invariant can be connected or not. We will call *bubble* the tensor invariants whose graphs are connected. Although interactions are inherently non-local in the ordinary sense, rotation invariance and connectivity allow us to define a notion of locality [108]. We will adopt the following definition:

**Definition 6** *We will say that a real function $f(T)$ is local if it can be expanded (finite or not) in bubble-labelled monomials. For instance, for $d = 3$:*

$$f(T) = a_1 + a_2 \times \!\!\!\!\!\!\!\! \text{⬤} \!\!\!\!\!\!\!\! + a_3 \times \!\!\!\!\!\!\!\! \text{▨} \!\!\!\!\!\!\!\! + a_4 \times \!\!\!\!\!\!\!\! \text{▥} \!\!\!\!\!\!\!\! + \cdots, \tag{151}$$

*for some reals parameters $\{a_1, a_2, a_3 \cdots\}$ that we call components. Moreover a local function whose expansion has only one bubble $b \neq \emptyset$ will be called observable.*

In this paper, we will focus on Gaussian noise. A random tensor will be Gaussian if it is distributed according to the probability measure $d\mu(T) = e^{-S_G(T)}/Z$, for:

$$S_G(T) = \frac{N^\alpha}{2a} \sum_{k_1 \cdots k_d} (T_{k_1 \cdots k_d})^2, \tag{152}$$

for some positive real number $a > 0$. The normalization factor $Z$ being such that:

$$\int d\mu(T) T_{k_1 \cdots k_d} T_{l_1 \cdots l_d} = a N^{-\alpha} \delta_{k_1 l_1} \delta_{k_2 l_2} \cdots \delta_{k_d l_d}. \tag{153}$$

The power $\alpha$ will be adjusted so that the $1/N$ expansion of the expectation values for products of observables exist. The expectation value $\langle f_1(T) \times \cdots \times f_K(T) \rangle$ for the product of $K$ observables $f_1(T), \cdots f_K(T)$ reads as,

$$\langle f_1(T) \times \cdots \times f_K(T) \rangle := \int d\mu(T) f_1(T) \times \cdots \times f_K(T). \tag{154}$$

The right hand side can be computed using Wick theorem [82] as a product over all allowed contractions of tensors pairwise. The result takes the following form:

$$\langle f_1(T) \times \cdots \times f_K(T) \rangle = \sum_{G \in \mathbf{G}_K} \mathcal{A}(G), \tag{155}$$

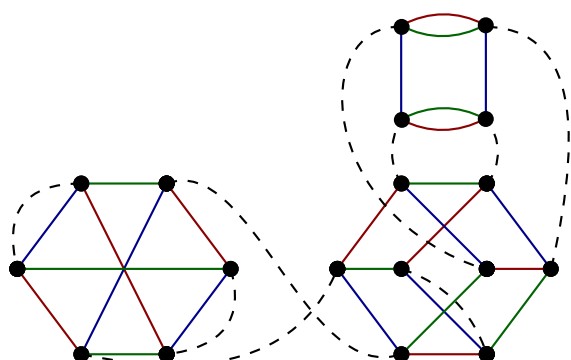

Figure 27: A typical Feynman graph involving 3 vertices for $d = 3$.

where $\mathbf{G}_K$ designates the set of Feynman diagrams having $K$ vertices, and $\mathcal{A}(G)$ the Feynman amplitude corresponding to the graph $G$. Figure 27 shows a typical graph $G \in \mathbf{G}_K$ for $K = 3$. In the Figure, the Wick contractions, involving the propagator (153), are pictured with dotted edges. If we associate to them a color 0, the resulting graph is $(d + 1)$-colored and regular. The knowledge of the free propagator allows to compute Feynman amplitudes exactly, and it is not hard to check that:

$$\mathcal{A}(G) = a^{L(G)} N^{-\alpha L(G) + F(G) + \sum_{j=1}^{K} \rho(b_j)} \prod_{j=1}^{K} \tilde{a}_{b_j}, \tag{156}$$

where $a_b =: \tilde{a}_b N^{\rho(b)}$ designates the component along the bubble $b$ defining the observable $f_b(T)$, the intrinsic scaling $\rho(b)$ depending only on $b$. In that equation moreover, $L(G)$ designates the number of Wick contractions, and $F(G)$ the number of closed faces:

**Definition 7** *A face $f$ is a bicolored cycle, which can be open or closed, corresponding respectively to open and closed faces. Its boundary $\partial f$ is the subset of dotted edges along the cycle.*

The exponents $\rho(b)$, as well as the constant $\alpha$ must be chosen in such a way that the development in $1/N$ exists, and that it favors a certain family of graphs when $N \to \infty$, all having the same scaling behavior with $N$. A solution was found in [109] for tensors of rank 3 to which we will limit ourselves in our investigations,[29] and we have the following theorem:

**Theorem 4** *Let $G$ a 4-colored Feynman graph. Fixing $\alpha = 3/2$, we define the degree $\omega(G)$ as follows*

$$\omega(G) = 3 + \frac{3}{2} L(G) - \sum_{b | N_b > 2} \rho(b) n_b(G) - |F|, \tag{157}$$

*where $n_b$ is the number of bubbles of type $b$, $N_b$ the number of nodes for the bubbles $b$, $L(G)$ the number of dotted edges, $|F|$ denote the total number of bicolored cycles in $G$ and the degree of the bubble, $\rho(b)$ is:*

$$\rho(b) = 3 - \frac{|F_b|}{2}, \tag{158}$$

*where $|F_b|$ is the number of closed bicolored cycles in $b$. In that way, Feynman amplitude $\mathcal{A}(G)$ for $G$ behaves as:*

$$\mathcal{A}(G) \sim N^{3 - \omega(G)}. \tag{159}$$

Leading order diagrams are then defined by the condition $\omega(G) = 0$ form a unique family of graphs which obey a simple recursive definition and are said *melonics*.

---

[29]The numerical resources required for the simulations increase dramatically with the rank of the tensor.

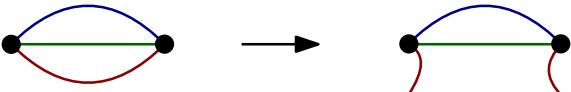

Figure 28: Illustration: Definition of the covariance matrix as the cut of a line of a tensor invariant.

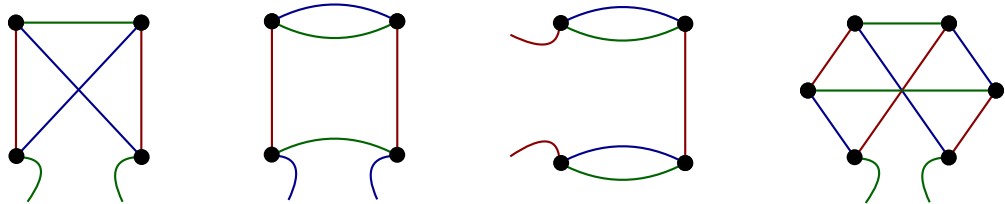

Figure 29: Covariance matrices obtained from $c_i$-cuts. From left to right: Edge deletion for 1, 2, 1 and 1 dipoles.

### 5.4.2 Definition of covariances

It is now necessary to generalize to the case of tensors the construction of the covariance matrix considered for matrices. For a matrix $X = \{X_{ai}\}$, the covariance matrix is defined by the averaging of the euclidean scalar product $C_{ij} = \frac{1}{P} \sum_a X_{ai} X_{aj}$. We are looking for a generalization of this construction to the case of tensors. One can think of:

$$C_{ij} = \left\langle \sum_{a_1, a_2, \cdots a_{d-1}} T_{a_1, a_2 \cdots, a_{d-1} i} T_{a_1, a_2 \cdots a_{d-1} j} \right\rangle. \tag{160}$$

This definition was used in [55], where the authors conducted numerical investigations that we will summarize in the next section. However, we will also seek to extend this natural definition. Let us note at first that, from the point of view of the formalism presented in the previous section, the definition (160) can be understood as the opening of the color edge $d$ (see Figure 28). The question is: Would it make sense to extend this observation to more complicated bubbles, involving a larger number of tensors? One could for example imagine cutting an edge of color $d$ in one of the three bubbles of Figure 26. We will call $k$-dipole any pair of nodes joined directly by $k$-colored edges, and we will speak of $c_i$-cut to designate the opening of the $i$-colored line on a $k$-dipole. In Figure 28, we have thus opened the $d$ color edge of a 3-dipole. Figure 29 provides another example, for the third diagram of Figure 26, in which we cut the edge of color $d$ on a 1-dipole.

To understand why tensor invariants would be useful, we must return to matrices. The interest of "traces" for matrices is well known and reflects the invariance of intrinsic properties of the signal by rotation. This observation explains why the search for eigenvalues is so central in data analysis. In the case of matrices, we can easily switch from eigenvalues to traces, and we can focus on one or the other at will. In the case of tensors, however, the notion of eigenvalue is not as obvious [110], but the notion of invariant is as we recalled in the previous section. For this reason, it is expected that each invariant, and thus each possible definition of the covariance matrix, carries partial information about the problem. A simple question we could try to answer with our RG formalism would be the following: Among all these definitions, which one is the most suitable from the point of view of signal detection? Note that this approach exploiting tensor invariants in the framework of PCA is not the first. A notable attempt was made in the recent work [84]. However, our approach is the first to exploit the RG.

Table 2: List of covariance matrices considered in our numerical investigations.

| Covariance | Family | Valence | Degree | Graph |
|:---:|:---:|:---:|:---:|:---:|
| $\mathbf{C}_1$ | | 2 | $\rho(b_*) = \frac{5}{2}$ | |
| $\mathbf{C}_2$ | | 4 | $\rho(b_*) = 2$ | |
| $\mathbf{C}_3$ | | 4 | $\rho(b_*) = 2$ | |
| $\mathbf{C}_4$ | Melon | 6 | $\rho(b_*) = \frac{3}{2}$ | |
| $\mathbf{C}_5$ | | 6 | $\rho(b_*) = \frac{3}{2}$ | |
| $\mathbf{C}_6$ | | 6 | $\rho(b_*) = \frac{3}{2}$ | |
| $\mathbf{C}_7$ | | 6 | $\rho(b_*) = \frac{3}{2}$ | |
| $\mathbf{C}_8$ | Complete | 4 | $\rho(b_*) = \frac{5}{2}$ | |

### 5.4.3 Numerical investigations

We will consider eight different definitions of covariance matrices in our investigations, noted $\mathbf{C}_I$, $I = 1, \cdots, 8$, all represented with their characteristics in the table 2 for non-symmetric tensors of size $50 \times 50 \times 50$. The corresponding typical spectra are shown in Figure 30, the signal being materialized as a non-symmetric deterministic tensor of rank $R = 50$, having a trivial singular decomposition of the form

$$X_{a_1,a_2,a_3}^{(\text{signal})} = \sum_{k=1}^{R} u_{a_1}^{(k)} v_{a_2}^{(k)} w_{a_3}^{(k)}. \tag{161}$$

Three observations must be made at this level.

- First, the rate of convergence of the distributions seems to depend on the definitions of the covariance matrix, and some numerical instabilities are expected for this reason. This is a consequence of the numerical resources required to simulate a random tensor concerning random matrices. This can also be a direct advantage of our approach, which would only require a numerical smoothing of the spectrum, less demanding in terms of computational resources to calculate the RG flow.

- Second, the shape of the distribution differ from a definition to another. The blue histogram for $\mathbf{C}_1$ for instance is almost symmetric around the eigenvalue 1, whereas the blue histogram for $\mathbf{C}_5$ is not symmetric, which a sharp behavior around the smaller eigenvalue, reminiscent of the MP law.

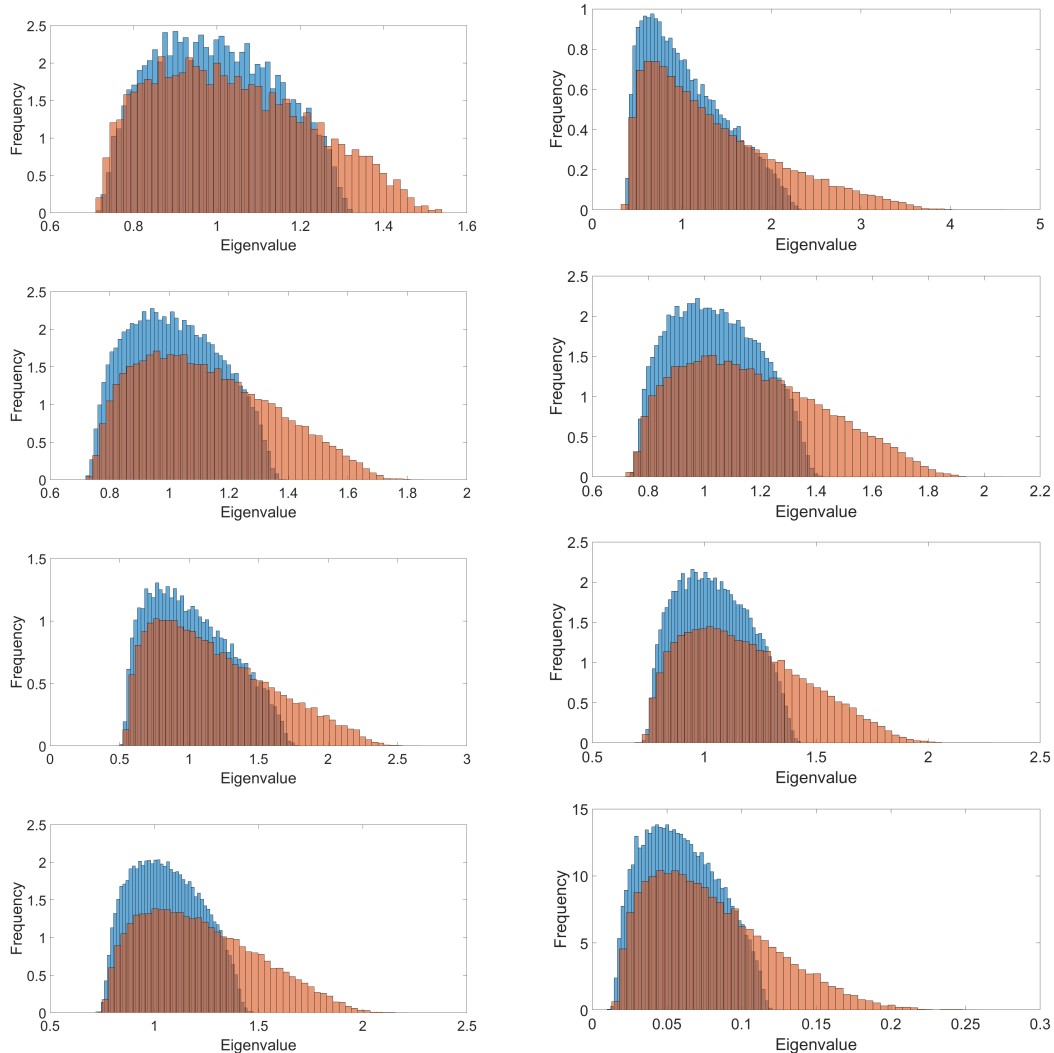

Figure 30: On the left, from top to bottom, we illustrate the histograms corresponding to $\mathbf{C}_1$, $\mathbf{C}_2$, $\mathbf{C}_3$ and $\mathbf{C}_4$. On the right, from top to bottom, we illustrate the histograms corresponding to $\mathbf{C}_5$, $\mathbf{C}_6$, $\mathbf{C}_7$ and $\mathbf{C}_8$. In all the plots, blue histogram corresponds to data without signal and the brown histogram corresponds to data with the maximum intensity of signal considered in our experiments.

- The third remark concern the stability of the smaller eigenvalue of the distribution. The smaller eigenvalue when a signal is added receives a positive shift, as pictured in Figure 31 for $\mathbf{C}_6$. This effect has been systematically corrected on the spectra of the figure. However, it should be noted that all definitions are unequal on this point. For example, $\mathbf{C}_1$ and $\mathbf{C}_8$ are not very sensitive, while the effect is very important for $\mathbf{C}_6$, for a signal of the same strength.

The definition $\mathbf{C}_1$ has already been considered in [55], and we will recall its main conclusions. As in the section 5.2, it is instructive to plot the typical flow behavior for a quartic truncation in the symmetric phase (i.e. assuming the expansion of the effective potential around $\chi = 0$ makes sense). The result is shown in Figure 32, obtained by numerical integration of flow equations. As for MP, one can see that ending regions of some RG trajectories are different if a signal is added to the spectrum. Moreover, there exist again a region that behaves as an effective fixed point like Wilson-Fisher. For MP, we argued that it could never be a true fixed

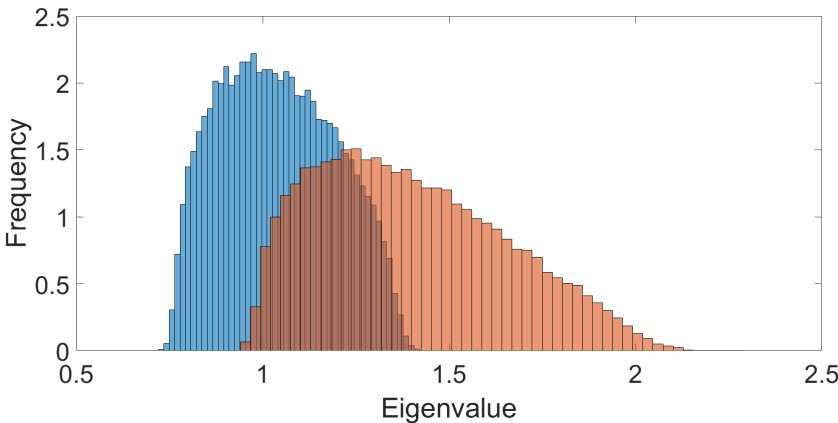

Figure 31: The original histograms corresponding to $\mathbf{C}_6$ before shifting the brown histogram corresponding to data with a signal in order to align it with the blue histogram corresponding to data without a signal.

point because scaling dimensions depends non trivially on the scale. For tensor, the situation is slightly different. Figure 33 illustrates the numerical behavior of canonical dimension with or without signal, and with or without averaging over a few numbers of draws. Two main differences have to be noticed with respect to MP:

- The scaling dimensions diverge again in the deep UV, but are all negative, meaning that local couplings are all irrelevant in this regime.

- In the deep IR moreover, the scaling dimensions for couplings become almost constant up to a given scale.

The first point illustrates that the flow behaves much better in the UV than for tensors and that the learnable region extends in fact over almost the whole spectrum. The second point shows that fixed point solutions exist for tensors, at least approximately (i.e. within numerical instabilities). For the rest, the previous conclusions (see 1) remain true for the definition $\mathbf{C}_1$: The presence of a signal affects the relevance of the couplings (the mainstream), and the canonical dimensions all decrease simultaneously. This concerns especially the quartic and sextic couplings, which tend to become irrelevant when a strong enough signal is added to the noise. Thus once again we see that the presence of a signal affects the sector relevant to the theory, tending to make the non-Gaussian perturbations irrelevant. As for MP, we find that the anomalous dimension remains a negligible correction in the IR, ensuring the validity of the LPA in this region. Figure 34 shows the symmetrical phase obtained for a sextic truncation, in the case of purely noisy data (on the left) and when a signal materialized by a deterministic tensor is added (on the right). We have also considered two approximation schemes, in the first one the potential is developed around $\chi = 0$ (top figure), and in the second round $\chi = \kappa$ (bottom figure). In both cases, we see that the conclusions remain the same as for the matrices and that the signal has the effect of reducing the size of the symmetric phase. Figure 35 shows the symmetry broken in the point of view of the effective potential.

Remarkably, all these conclusions remain true for all definitions proposed in the table 2. In particular, definitions 1 to 7 show strong similarities. Definition 8 however stands out slightly, especially for the canonical dimensions. The empirical dimension (with and without signal) is represented in Figure 36. Contrary to what we observed for $\mathbf{C}_1$ and which remains true for all definitions up to $\mathbf{C}_7$, all dimensions are positive in the IR. Only the quartic and sextic couplings have positive dimensions along the stream, but for a finite period of its history only.

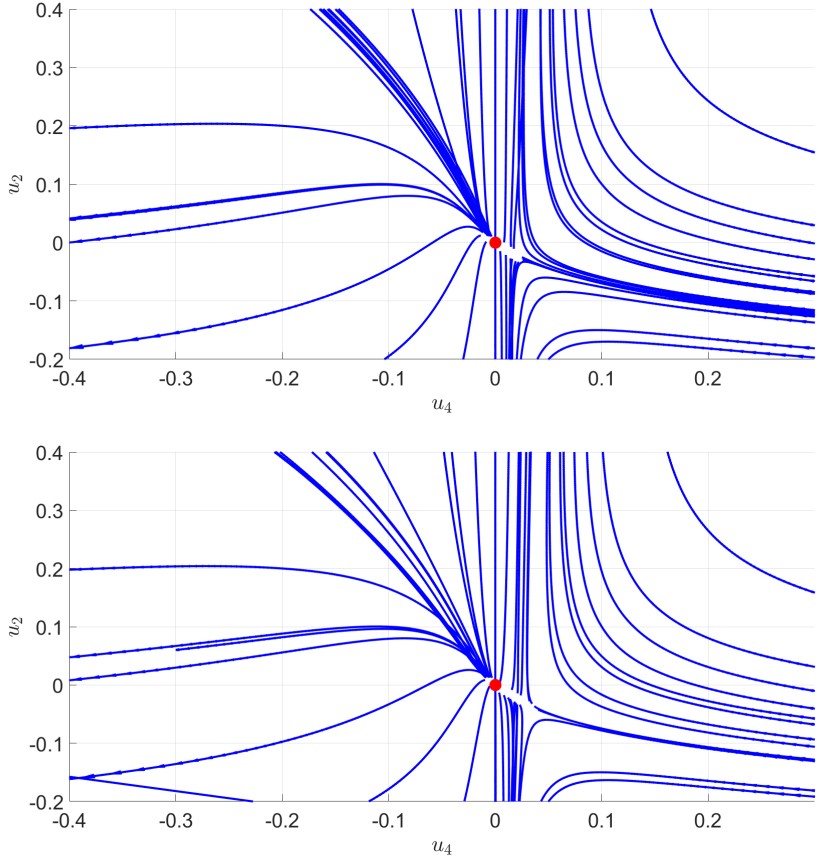

Figure 32: Behavior of the RG flow for a quartic truncation in the symmetric phase, using covariance $\mathbf{C}_1$. On the top, the flow for the blue histogram (without signal). On the bottom, the flow for the brown histogram (with signal).

Moreover, all dimensions become irrelevant in the UV, as seen before. This illustrates an important point. The sensitivity of the canonical dimensions to the presence of a signal differs from one definition to another, and also depends on the scale at which the signal is sought. Thus although the conclusions given by the proposition 1 seem universal, the RG seems to give a new criterion on the practical relevance of the different definitions of covariance. In Figure 37, we track the evolution of the average canonical dimensions as the signal strength is increased. We see that for weak signals (signal intensity < 1) the canonical dimensions associated with the theories $\mathbf{C}_1$ and $\mathbf{C}_5$ change much faster than the others, and we, therefore, expect their relevant sector to be more affected than that of the other theories. In this sector, on the other hand, the canonical dimensions of the $\mathbf{C}_8$ theory change very little. Conversely, in the strong signal regime (signal intensity > 1), the $\mathbf{C}_6$, $\mathbf{C}_7$ and $\mathbf{C}_8$ theories seem to be more sensitive, the $\mathbf{C}_8$ theory being the most sensitive of all. These conclusions seem to agree with the conclusions of [84] concerning the relevance of the "tetraedron" graph defining $\mathbf{C}_8$, the authors showing that combinatorial properties of such a graph simplify proofs for detection theorem. Hence our RG formalism could provide answers on open topics such as tensor PCA.

## 6 Concluding remarks and open issues

This pedagogical article has reviewed analogous field-theory models discussed in a series of recent papers. We have proposed an alternative point of view on signal detection in the case of

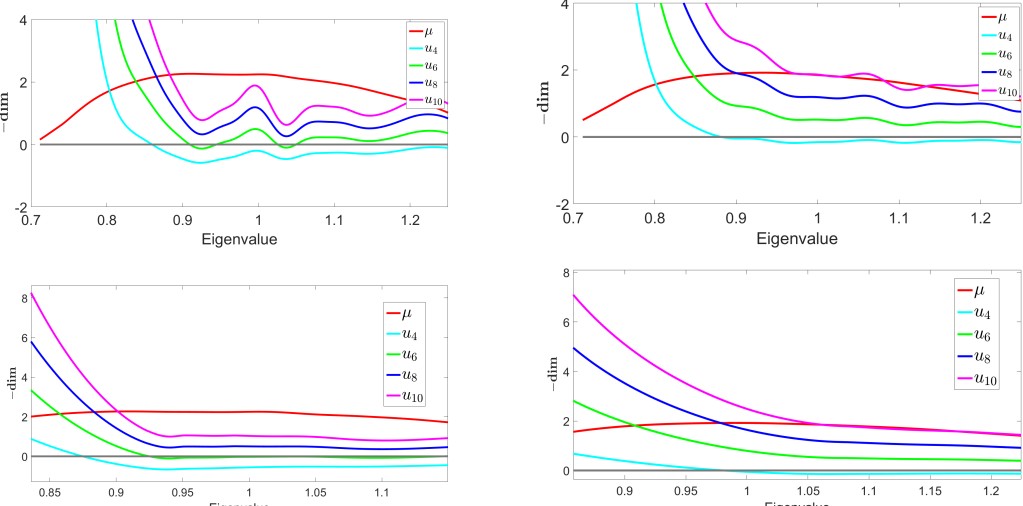

Figure 33: Canonical dimensions for $\mathbf{C}_1$. On left, we illustrate the results for data without signal and on right the results for data with the maximum intensity of signal considered in our experiments. On the top Figures are obtained without numerical smoothing. On the bottom we show the same figure after averaging over a few draws. In both cases, the red curve denotes the shape of the fitted eigenvalue spectrum.

an almost- continuous spectrum. In that way and through RG-arguments, we have been able to discuss the universality of the empirical proposition (1), but also to characterize a type of tensor invariant as optimal from the point of view of signal detection. Our main message then is, that it is possible to understand signal detection by the significant changes on the universal properties of noise models, in particular for the number of relevant couplings by which asymptotic states in the IR are distinguished. That is reminiscent of the physics of critical phenomena, and makes it possible to consider signal detection as a phase transition, breaking the native $\mathbb{Z}_2$ symmetry of models based on a principle of maximum entropy. Moreover, the RG allows a natural understanding of the existence of a detection threshold due to the existence of a compact subset of physically acceptable initial conditions, included in the symmetric phase. Open questions will be considered in later publications and include the following. We have worked in the continuum limit. Therefore, to compute derivatives and scaling dimensions, a smooth fitting of the empirical distributions is required. That may induce some spurious effects. A discrete version of the RG (via discrete sums and finite differences) could avoid such difficulties. More generally, the methods we have used to construct the RG are approximate. The LPA easily connects the UV and IR two-point functions, with all the effects of quantum fluctuations included in the mass, and this approximation is certainly justified in the IR, where the anomalous dimension remains a tiny correction. Yet, if we approach UV scales we can expect more serious difficulties. The LPA might no longer be valid anymore, and corrections or more sophisticated methods should be considered [111–113]. Furthermore, after the threshold of the "dimensional crisis", the number of relevant parameters and their canonical dimensions could become very large for some noise models like MP. In this case, the very notion of truncation could become problematic. Another issue concerns the formalism itself, based on an equilibrium field theory where configurations are weighted by a Boltzmann weight $p[\phi] \propto e^{-H[\phi]}$. The assumption of equilibrium comes from the maximum entropy principle, but for a system exhibiting a symmetry breaking of the microscopic Hamiltonian, the ergodic assumption that the probability of finding the system in a certain state is given by Boltzmann's law may be questionable. A possible alternative could use a dynamical description, based on a fictitious

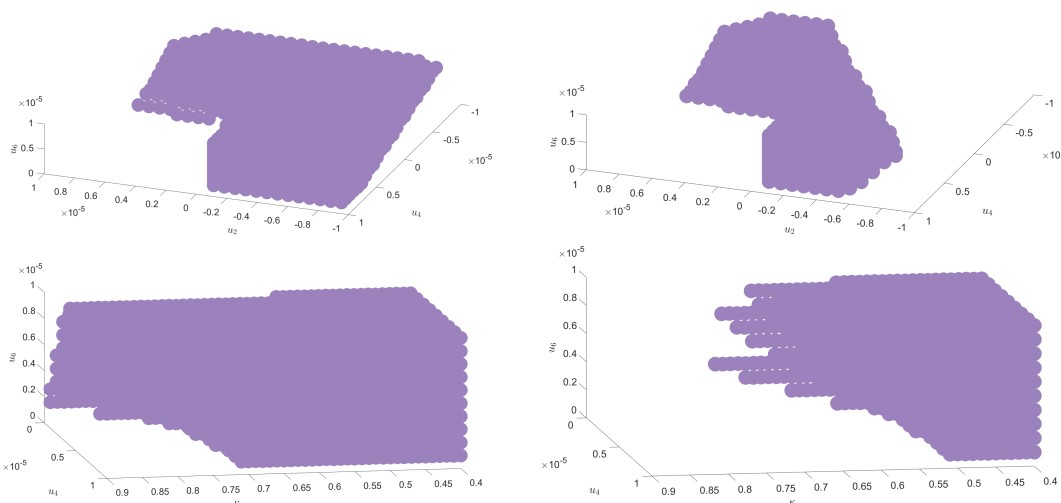

Figure 34: Illustration of the compact region $\mathcal{R}_0$ (illustrated with purple dots) in the vicinity of the Gaussian fixed point providing initial conditions ending in the symmetric phase. On the left: the region for purely i.i.d random tensors in the expansion around $\chi = 0$ (on the top) and around a running vacuum $\chi = \kappa$ (on the bottom). On the right: the same regions when a signal build as a deterministic tensor is added.

time $t$, by constructing not spatial but temporal averages. The ergodicity breaking would then correspond to the fact that the correlations do not cancel in the limit where the width of the interval on which the averages are constructed tends to infinity.

Introducing such a dynamical model would interpret the field as a time-dependent variable, whose dynamics would be governed by a disordered Langevin-type equation, the disorder being given by the covariance matrix. Such a model would have to admit the field theory discussed in this paper as a special case. The presence or absence of a signal would then correspond to a breaking of ergodicity. Such models are reminiscent of spin glass physics, and have been studied extensively in the literature [114–117], and recently by the renormalization group [34].

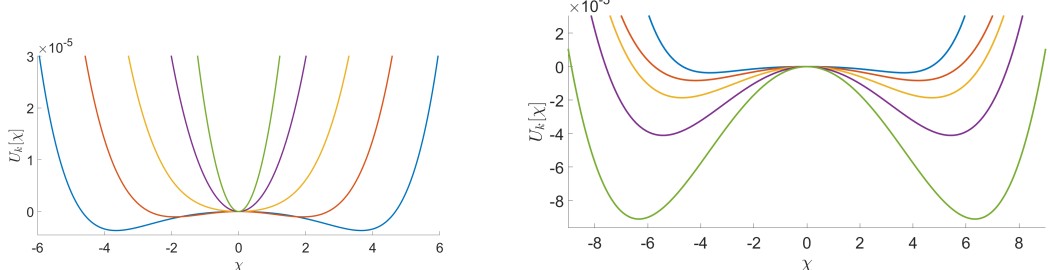

Figure 35: Illustration of the evolution of the potential associated to the coupling $u_2$, $u_4$ and $u_6$ for a truncation around $\chi = 0$, on the right without signal, on the left with a signal. This example corresponds to specific initial conditions (in blue) taking in the interior of the purple region. We illustrate different points of the trajectory, from UV to IR respectively by the red, yellow, purple curves. The green curves correspond to the ending point of the considered trajectory.

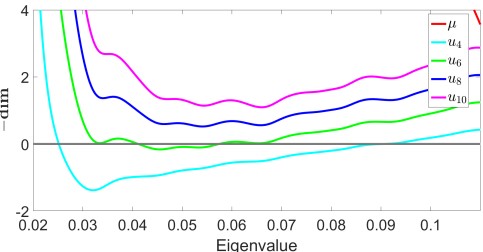 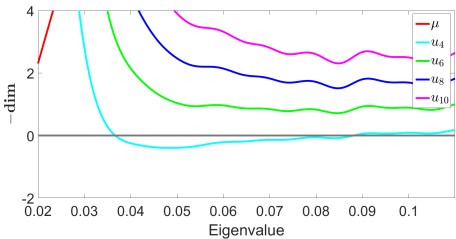

Figure 36: Canonical dimensions for $\mathbf{C}_8$. On left, we illustrate the results for data without signal and on right the results for data with the maximum intensity of signal considered in our experiments.

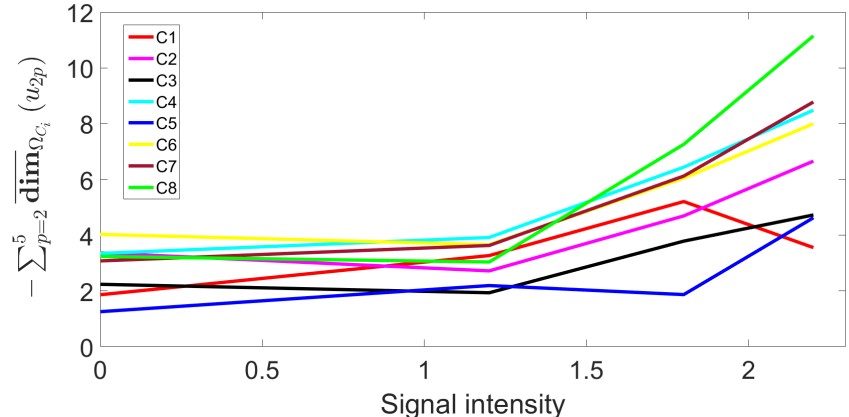

Figure 37: Results obtained for the different covariance matrices (from $\mathbf{C}_1$ to $\mathbf{C}_8$) considered in our experiments. These results indicate the amount of "shift" in the canonical dimensions with respect to the intensity of the signal present in the data. In details, we compute the sum of the mean over the learnable region ($\Omega_{C_i}$ for each covariance matrix) of the canonical dimensions corresponding to the first four even local interactions ($u_4$, $u_6$, $u_8$ and $u_{10}$).

Finally, an important point of practical interest could be to understand how these theoretical results could lead to new detection algorithms or the improvements of the existing ones. For the moment, the leap seems too big, and a better understanding of the theory seems necessary before such practical applications can be considered.

## Acknowledgments

The authors specifically thank Julie Michel for carefully proofreading the manuscript. Her contribution has greatly improved the presentation.

## A  Symmetries and resummation for noisy nearly Gaussian models*

This section explores different aspects of the theory for a purely noisy signal near to the Gaussian point. Here we formalize the case where the interactions are non local, anticipating further investigations beyond the scope of this paper.

## A.1 Ward-Takahashi identities

In quantum field theory, the symmetries look like identities between correlations functions given by Ward-Takahashi identities. We must now investigate such identities for the model (38), focusing on the quartic model. Let us consider the generating functional:

$$Z[\tilde{D}, g, \chi] := \int [d\Phi] e^{-\frac{1}{2} \sum_{i,j=1}^{N} \phi_i \tilde{D}_{ij} \phi_j - \frac{g}{4!} \sum_{i=1}^{N} \phi_i^4 + \sum_{i=1}^{N} \chi_i \phi_i}. \tag{A.1}$$

It is formally invariant under the global transformation:

$$\phi_i \to \phi_i' = \sum_{j=1}^{N} O_{ij} \phi_j, \tag{A.2}$$

for $O \in \mathrm{O}(N)$, because the functional integral covers all the configurations for the field $\Phi$. This formal invariance, for infinitesimal transformations, can be expressed as a functional relation between correlations functions - see [82] for more details. For an infinitesimal transformation $O_{ij} = \delta_{ij} + \varpi_{ij}$, with $\varpi_{ij} = -\varpi_{ji}$ along the Lie algebra of the rotation group $\mathrm{O}(N)$, the variation of the partition function reads:

$$\delta Z = \left\langle -\frac{1}{2} \sum_{i,j=1}^{N} \phi_i [\varpi, \tilde{D}]_{ij} \phi_j - \frac{g}{6} \sum_{i,j=1}^{N} \varpi_{ij} \phi_j \phi_i^3 + \sum_{i,j=1}^{N} \chi_i \varpi_{ij} \phi_j \right\rangle \equiv 0. \tag{A.3}$$

The first term involves the commutator of the kinetic kernel with $\varpi$.

**Effective Ward-Takahashi identities.** If we consider a nearly Gaussian model for purely noisy data, the matrix $\tilde{D}$ must have to be close to $C^{-1}$ which, following our assumptions, is a positive random matrix. In that way, we assume that eigenvalues and eigenvectors are closed for these two matrices, the correction being given by equations (50) and (49). The matrix $C$ is in principle deterministic, but for a purely noisy signal it has no particular structure and, following the original assumption of Wigner,[30] $C$ can by replaced by an element of a suitable random matrix ensemble - like for Wigner or Wishart ensembles. The point is, that for large $N$, these ensembles are essentially invariant by rotation in law meaning that $C$ is as probable as $OCO^T$, or at least asymptotically for $N \to \infty$ for some rotation matrix $O$.

The statistical properties of the large matrix do not change if we apply a global rotation on its elements. This is explicitly the case for Wigner and white Wishart ensembles,[31] such for all the noise models that we will consider in this article (see Section 5). Hence, for $N$ large enough, we expect the relevant features of the field theory to be essentially not sensitive to the specific draw in the considered matrices universality class chosen to define the matrix $\tilde{D}$. In particular, one expect the two realizations $\tilde{D}^{-1} \approx C$ and $O\tilde{D}^{-1}O^T$ are indistinguishable by the field theory. This observation can be translated by a formal equivalence relation:

$$Z[O\tilde{D}O^T, g, \chi] \sim Z[\tilde{D}, g, \chi]. \tag{A.4}$$

The symbol $\sim$ means that the two theories cannot be distinguished by their ability to describe the correlations of the field $\Phi$. From this point of view, the physically relevant object is the equivalence class itself, defined by relation (A.4). We therefore will see that, in the neighbourhood of the Gaussian point, the theory projects itself as a class functional. The relation (A.2) defines on the other hand a formal identification between points of the phase space, related by

---

[30]In its seminal papers, Wigner focused on the Hamiltonian of a nucleus with a large number of nucleons.

[31]Even for more general definitions of this ensemble, where rotational invariance holds only asymptotically.

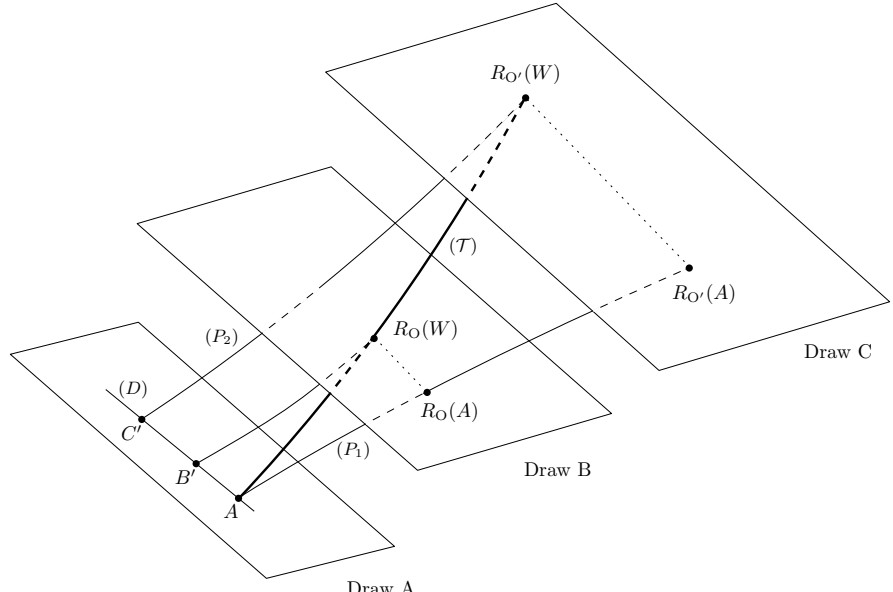

Figure 38: Qualitative illustration of the projection. We consider three draws for the random matrix $K$, say $A$, $B \equiv R_O(A)$ and $C \equiv R_{O'}(A)$, such that kinetic kernels are $OKO^T$ on $B$ and $O'KO'^T$ on $C$ for some orthogonal matrices $O$ and $O'$, the above relation defining the maps $R_O$ and $R_{O'}$. The three planes materialize the theory space for each draw, the solid edges $(P_1)$ and $(P_2)$ are equivalence class for fixed interactions and the heavy solid edge $(\mathcal{T})$ is the global translation of fields $\phi_i \to \phi_i' = \sum_{j=1}^N O_{ij}\phi_j$, that we denote as $R_O(W)$ (a global rotation acting on the interaction space $W$). The solid edge $(D)$ on $A$ materializes the equivalent class of models, defined by the equivalence relation $A \sim R_O(A)$.

a global rotation in the space of parameters defining the theory. Indeed, we can view the quartic coupling $(g/4!)\sum \phi_i^4$ as a specific value for the general coupling $\sum_{i,j,k,l} w_{ijkl}\phi_i\phi_j\phi_k\phi_l$, assuming that tensor $w_{ijkl}$ transforms under rotations as:

$$w_{ijkl} \to \sum_{p,q,r,s} w_{pqrs}O_{pi}O_{qj}O_{rk}O_{sl} =: (O_\triangleright w)_{ijkl}. \tag{A.5}$$

We denote as $W = (\chi, \tilde{D}, w)$ and we introduce the map $R_O$ which converts some draw $C$ as $R_O(C) := OCO^T$ and acts on $R_O(W) := (O_\triangleright\chi, O_\triangleright\tilde{D}, O_\triangleright w)$, where $O_\triangleright\tilde{D} := O\tilde{D}O^T$ and $O_\triangleright\chi := O\chi$. We also indicate as $Z[W]$ the corresponding partition function,

$$Z[W] := \int [d\Phi] e^{-\frac{1}{2}\sum_{i,j=1}^N \phi_i\tilde{D}_{ij}\phi_j - \sum_{i,j,k,l}^N w_{ijkl}\phi_i\phi_j\phi_k\phi_l + \sum_{i=1}^N \chi_i\phi_i}, \tag{A.6}$$

and the global translation invariance of the functional integral under transformation (A.2) reads as:

$$Z[W] \equiv Z[R_O[W]]. \tag{A.7}$$

With equivalence (A.3), identity (A.7) establish another equivalence, relying partition functions having the same kinetic kernel but interactions transformed by $O$: $Z[(O_\triangleright\chi, \tilde{D}, O_\triangleright w)] \sim Z[(\chi, \tilde{D}, w)]$. Figure 38 illustrates the construction of the projection into this equivalence class on models.

Let us show that $Z[W]$ is a class functional regarding to the equivalence relation (A.7). If we consider an infinitesimal transformation $O = I + \varpi$, equivalence (A.7) implies:

$$\left\langle \frac{1}{2} \sum_{i,j=1}^{N} \phi_i [\varpi, \tilde{D}]_{ij} \phi_j \right\rangle \sim 0, \tag{A.8}$$

in other words:

$$\sum_{i,j=1}^{N} [\varpi, \tilde{D}]_{ij} C_{ji} \sim 0, \tag{A.9}$$

where $\langle \phi_i \phi_j \rangle \equiv C_{ij}$, the full propagator. This relation is trivially satisfied at zero order in $\epsilon$. Indeed, writing $C_{ij} = \sum_\mu \lambda_\mu u_i^{(\mu)} u_j^{(\mu)}$ and neglecting the difference between $u^{(\mu)}$ and $\tilde{u}^{(\mu)}$ (see (50)), the previous sum reads:[32]

$$\sum_{i,j,l,\mu=1}^{N} [\varpi_{il} \tilde{D}_{lj} - \tilde{D}_{il} \varpi_{lj}] u_i^{(\mu)} u_j^{(\mu)} \lambda_\mu = \sum_{i,j,l,\mu=1}^{N} \frac{\varpi_{il} u_i^{(\mu)} u_l^{(\mu)} - \varpi_{lj} u_l^{(\mu)} u_j^{(\mu)}}{N} + \mathcal{O}(\epsilon) = \mathcal{O}(\epsilon). \tag{A.10}$$

This holds again at the first order in $\epsilon$. Defining $f(\mu, \nu) := \epsilon \Xi_{\mu\nu} (\tilde{\lambda}_\mu - \tilde{\lambda}_\nu)^{-1}$ following equation (50), relation (A.8) at first order in $\epsilon$ reads as:

$$\delta_\epsilon := \sum_{i,j,l,\mu \neq \nu} f(\mu, \nu) \frac{\varpi_{il} \tilde{u}_i^{(\nu)} \tilde{u}_l^{(\mu)} \lambda_\mu - \varpi_{lj} \tilde{u}_l^{(\nu)} \tilde{u}_j^{(\mu)} \lambda_\nu}{N}, \tag{A.11}$$

which can be simplified as:

$$\delta_\epsilon = \sum_{i,j,\mu \neq \nu} f(\mu, \nu)(\lambda_\mu - \lambda_\nu) \varpi_{ij} \tilde{u}_i^{(\nu)} \tilde{u}_j^{(\mu)} = \epsilon \sum_{i,j,\mu \neq \nu} \Xi_{\mu\nu} \varpi_{ij} \tilde{u}_i^{(\nu)} \tilde{u}_j^{(\mu)}. \tag{A.12}$$

Eigenvectors being delocalized, the missing term $\mu = \nu$ must be of order $1/N$, moreover $\sum_{i,j} \varpi_{ij} \tilde{u}_i^{(\mu)} \tilde{u}_j^{(\mu)} = 0$ as $\varpi_{ij}$ is a skew symmetric matrix. Hence:

$$\delta_\epsilon = \sum_{i,j,\mu \neq \nu} \Xi_{\mu\nu} \varpi_{ij} \tilde{u}_i^{(\nu)} \tilde{u}_j^{(\mu)} = \sum_{i,j,\mu,\nu} \Xi_{\mu\nu} \varpi_{ij} \tilde{u}_i^{(\nu)} \tilde{u}_j^{(\mu)} = \sum_{i,j} \varpi_{ij} \Xi_{ij} = 0, \tag{A.13}$$

the last equality coming from the fact that $\Xi$ is symmetric. Then $Z[\tilde{D}, \chi, g]$ is almost a class function, at least is the vicinity of the Gaussian point.

This constraint has an impact in which the variation (A.3) becomes:

$$\delta Z = \sum_{i,j=1}^{N} \varpi_{ij} \left\langle -\frac{g}{6} \phi_j \phi_i^3 + \chi_i \phi_j \right\rangle = 0. \tag{A.14}$$

This equation must be true for all $\varpi$ along with the Lie algebra of the group $O(N)$. Taking into account that $\varpi_{ij} = -\varpi_{ji}$, the previous equation implies:

$$-\frac{g}{6} \left( G_{iiij}^{(4)} - G_{jjji}^{(4)} \right) + \left( \chi_i M_j - \chi_j M_i \right) = 0, \tag{A.15}$$

---

[32]This can be derived more simply from the the observation that:

$$\mathrm{Tr}[\varpi, \tilde{D}] \tilde{D}^{-1} = \mathrm{Tr}\, \varpi[\tilde{D}, \tilde{D}^{-1}] = 0.$$

where:

$$G^{(2n)}_{i_1 i_2 \cdots i_{2n}} := \frac{1}{Z[W]_{\chi=0}} \frac{\partial^{2n} Z[W]}{\partial \chi_{i_1} \partial \chi_{i_2} \cdots \partial \chi_{i_{2n}}}, \quad \text{and} \quad M_i := \frac{1}{Z[W]_{\chi=0}} \frac{\partial Z[W]}{\partial \chi_i}. \tag{A.16}$$

We are now investigating some consequences of this equation. Taking the first derivative with respect to $\chi_l$ and set $\chi = 0$, and assuming that $M$ is small and discarding contributions of order $gM$ where $M$ is small, we obtain:

$$-\sum_{l=1}^{N}(\tilde{D}_{jl} G^{(3)}_{ill} - \tilde{D}_{il} G^{(3)}_{jll}) + \delta_{il} M_j - \delta_{jl} M_i = 0. \tag{A.17}$$

At first order in $M$, we have:

$$G^{(2n+1)}_{i_1,\cdots,i_{2n+1}}(M) = G^{(2n+1)}_{i_1,\cdots,i_{2n+1}}(0) + \sum_{k=1}^{N} G^{(2n+2)}_{i_1,\cdots,i_{2n+1},k}(0) M_k + \mathcal{O}(M^2), \tag{A.18}$$

and in particular, discarding contributions of order $gM$:

$$G^{(3)}_{ijk} = \sum_{l=1}^{N}(C_{ij} C_{kl} + C_{ik} C_{jl} + C_{il} C_{jk}) M_l + \mathcal{O}(gM), \tag{A.19}$$

leading to:

$$-M_j C_{ik} + M_i C_{jk} - \delta_{kj} C_{im} M_m + \delta_{ki} C_{jm} M_m + \delta_{il} M_j - \delta_{jl} M_i = 0. \tag{A.20}$$

Then, summing over $l$ and $j$, we get:

$$-M_j \sum_{i,k=1}^{N} C_{ik} + N\bar{M} \sum_{k=1}^{N} C_{jk} - \sum_{i,m=1}^{N} C_{im} M_m + N \sum_{m=1}^{N} C_{im} M_m + N(M_j - \bar{M}) = 0, \tag{A.21}$$

where $\bar{M} := \sum_{i=1}^{N} M_i$. Exploiting rotational invariance for large $N$, we expect $NC_{ij} \approx \sum_i C_{ij}$, and the previous equation reduces to:

$$(-M_j + \bar{M}) \left( \sum_{k,l=1}^{N} C_{kl} - N \right) \approx 0, \tag{A.22}$$

implying:

$$\boxed{M_j \approx \frac{1}{N} \sum_{i=1}^{N} M_i.} \tag{A.23}$$

This equation holds for all $j$. In other words, each components of the *classical field* $M_i$ self averages, and equals to the means field $\bar{M}$. Hence, the classical field $M$ inherits of the delocalization of the eigenvectors of the kinetic kernel $\tilde{D}$, which almost equals $C^{-1}$. Note that the derivation of these identities assumes $N$ is large. Now, let us derive twice with respect to the source, i.e. applying $\partial^2/\partial\chi_k \partial\chi_l$ on the left hand side of (A.15). Setting $\chi = 0$, we get:

$$-\frac{g}{6}\left(G^{(6)}_{iiijkl} - G^{(6)}_{jjjikl}\right) - \sum_m (\tilde{D}_{im} G^{(4)}_{mjkl} - \tilde{D}_{jm} G^{(4)}_{mikl})$$
$$+ \left(\delta_{ki} G^{(2)}_{jl} + \delta_{li} G^{(2)}_{jk} - \delta_{jl} G^{(2)}_{ik} - \delta_{kj} G^{(2)}_{il}\right) = 0, \tag{A.24}$$

where $G_{ij}^{(2)} \equiv C_{ij}$, and the last term comes from the second derivative of (A.8) (which is not zero) and reads for $\chi = 0$:

$$
\begin{aligned}
\frac{1}{2} \sum_{i,j=1}^{N} [\varpi, \tilde{D}]_{ij} G_{ijkm}^{(4)} &= \frac{1}{2} \sum_{i,j,l=1}^{N} (\varpi_{il} \tilde{D}_{lj} - \tilde{D}_{il} \varpi_{lj}) G_{ijkm}^{(4)} \\
&= \frac{1}{2} \sum_{i,j,l=1}^{N} \varpi_{il} (\tilde{D}_{lj} + \tilde{D}_{jl}) G_{ijkm}^{(4)} \\
&= \frac{1}{2} \sum_{i,j,l=1}^{N} \varpi_{ij} (\tilde{D}_{jl} G_{ilkm}^{(4)} - \tilde{D}_{il} G_{jlkm}^{(4)}).
\end{aligned}
\tag{A.25}
$$

**Schwinger-Dyson equations.** Finally, let us show how these equations can be derived from the assumption that fields vanish at the boundaries of the formal Lebesgue integral defining the generating functional, better know as the Schwinger-Dyson equation (SDE). Recall that SDE arising from the functional relation [82]:

$$
Z[\tilde{D}, g, \chi] := \int [d\Phi] \frac{\partial}{\partial \phi_i} \left( \phi_j \, e^{-\frac{1}{2} \sum_{i,j=1}^{N} \phi_i \tilde{D}_{ij} \phi_j - \frac{g}{4!} \sum_{i=1}^{N} \phi_i^4 + \sum_{i=1}^{N} \chi_i \phi_i} \right) \equiv 0.
\tag{A.26}
$$

Computing derivative of each terms, we get:

$$
\delta_{ij} Z[\tilde{D}, g, \chi] - \sum_m \langle \phi_j \tilde{D}_{im} \phi_m \rangle - \frac{g}{6} \langle \phi_i^3 \phi_j \rangle + \chi_i \langle \phi_j \rangle = 0,
\tag{A.27}
$$

or:

$$
\delta_{ij} - \sum_m \tilde{D}_{im} G_{mj}^{(2)} - \frac{g}{6} G_{iiij}^{(4)} + \chi_i M_j = 0.
\tag{A.28}
$$

In the vicinity of the Gaussian point, $\sum_m \tilde{D}_{im} G_{mj}^{(2)}$ is almost $\delta_{ij}$ and the two first terms cancel. Taking the skew symmetric part of the remaining terms, we recover the Ward identity (A.15). Applying $\partial^2 / \partial \chi_k \partial \chi_l$ on both sides of equation (A.27), and setting $\chi = 0$ at the end, we get:

$$
\frac{g}{6} G_{iiijkl}^{(6)} + \sum_m \tilde{D}_{im} G_{mjkl}^{(4)} - (\delta_{ij} G_{kl}^{(2)} + \delta_{il} G_{jk}^{(2)} + \delta_{ik} G_{jl}^{(2)}) = 0.
\tag{A.29}
$$

If we construct the skew symmetric part of that equation with respect to $i$ and $j$, we get:

$$
\frac{g}{6} (G_{iiijkl}^{(6)} - G_{jjjikl}^{(6)}) + \sum_m (\tilde{D}_{im} G_{mjkl}^{(4)} - \tilde{D}_{jm} G_{mikl}^{(4)}) - (\delta_{il} G_{jk}^{(2)} + \delta_{ik} G_{jl}^{(2)} - \delta_{jl} G_{ik}^{(2)} + \delta_{jk} G_{il}^{(2)}) = 0,
\tag{A.30}
$$

which is nothing but the Ward identity (A.24). The same thing remains true for higher correlation functions.

## A.2 Resummation for large $N$

In that section, we show how the quartic model can be formally solved in the large $N$ limit. The methods presented in this section and the following one have been quite extensively studied in similar cases, where the dominant large $N$ sector exhibits a branched structure [108, 118].

We focus on the quartic model, with configuration field probability $p(\Psi)$ given by:

$$
p(\Psi) = \frac{1}{Z} \exp \left( -\frac{1}{2} \sum_{\mu=1}^{N} \psi_\mu \tilde{\lambda}_\mu^{-1} \psi_\mu - \frac{g}{8N} \left( \sum_{\mu=1}^{N} \psi_\mu^2 \right)^2 \right).
\tag{A.31}
$$

We denote as $\mathfrak{G} := \lim_{N \to \infty} G^{(2)}$ the relevant contributions to the effective propagator $G^{(2)}$. From the definition we have $G^{(2)} := C$, hence $C \to \mathfrak{G}$ in law in the large $N$ limit.

First, let us show how we can obtain a closed equation for the 2-point function. In this section we denote as $\Sigma$ the *leading order* 1PI 2-point function. The diagrams forming $\Sigma$ can be obtained from the vacuum diagrams by cutting them by one line.[33] Let us now consider the vacuum diagrams in the $N \to \infty$ limit and try to define their structure by a recurrence on the number of vertices $p$. For $p = 1$, there are two allowed configurations:

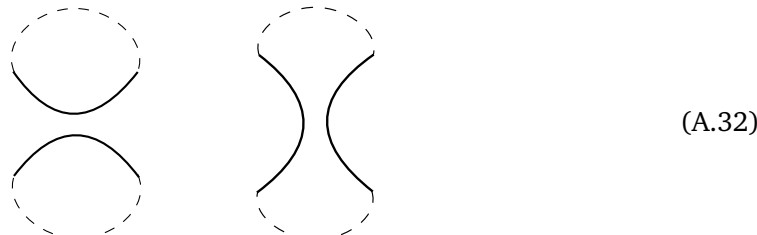

$$(A.32)$$

where the solid edges corresponds to Kronecker delta defining the quartic interaction and the dotted edge materialize the Wick contractions with the free propagator $\tilde{D}^{-1}$. We have the following definition:

**Definition 8** *A face is a cycle made of a succession of dotted and solid lines. It can be closed or open. The boundary $\partial f = \{e\}$ of a face $f$ is defined as the set $e$ of dotted edges building the face.*

Each face involves a sum over eigenvalues, which is of order $N$, and each vertex carry a factor $1/N$. The Feynman amplitudes $\mathcal{A}_{\mathcal{G}}$ in the perturbative expansion of partition function:

$$Z = \sum_{\mathcal{G}} \mathcal{A}_{\mathcal{G}}, \qquad (A.33)$$

are then power countable, the amplitude $\mathcal{A}_{\mathcal{G}}$ labeled with the diagram $\mathcal{G}$ involving $F$ faces ans $V$ vertices scaling as $\mathcal{A}_{\mathcal{G}} \sim N^{F-V}$. We will define the degree of divergence of the graph as $\omega(\mathcal{G}) := F - V$. The diagram on the left then involves two faces and scale as $\frac{1}{N} \sum_{\mu,\nu} \lambda_\mu \lambda_\nu \to N (\int \mu(\lambda) \lambda d\lambda)^2 = \mathcal{O}(N)$ ($\omega = 1$) where, in contrast, the diagram on the right involves only one and then scale as $\frac{1}{N} \sum_\mu \lambda_\mu^2 \to \int \mu(\lambda) \lambda^2 d\lambda = \mathcal{O}(1)$ ($\omega = 0$). Its contribution is therefore overwhelmed by the one of the first diagram. For higher order diagrams we introduce a new representation known as the *intermediate field representation* [108, 119, 120]. The rule is as follows: to each vertex we match a thick line, and to each face a "loop" vertex. In that way, the two diagrams (A.32) read:



$$(A.34)$$

We will therefore proof the following statement:

**Proposition 2** *Relevant diagrams in the large $N$ limit are trees in the intermediate field representation.*

The first step of the proof has be done, the relevant diagram to the first order being a tree (on the left of the equation (A.34)). We consider a tree with $n$ edges (i.e. a Feynman diagram involving $n$ vertices) whose 39 provides an example. Now let's try to find out how to go from

---

[33]Recall that the lines in question are Wick's contractions of the perturbation theory.

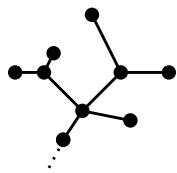

Figure 39: A typical tree in the intermediate field representation. The dotted line means that the tree continues.

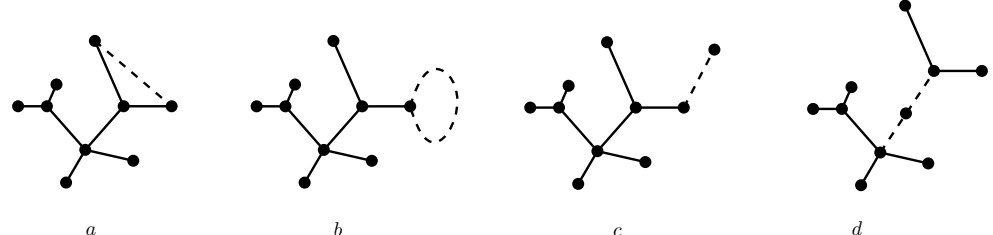

Figure 40: The four moves allowing to construct a $n + 1$ tree from a $n$ tree.

a tree of order $n$ to a tree of order $n + 1$. It is easy to convince oneself that only 4 moves allow to do this, all listed in Figure 40. We will study each of them separately.

In intermediate field representation, each line costs $1/N$ and each vertex creates a factor $N$. The configurations of types (c) and (d) preserve the tree structure, which add a line and create a face. The variation of power count is thus null $\delta\omega = 0$. On the contrary, the configurations (a) and (b) add a line without creating new vertices, so $\delta\omega = -1$. The proposition is then proved, and the leading order amplitudes behave as $\mathcal{A}_\mathcal{G} \sim N$. The 1PI 2-point diagrams, involved in the Feynman expansion of $\Sigma$, can then be obtained by "cutting" a dotted line in a vacuum diagram. Because the operation destroys a face, we deduce that the corresponding amplitudes must behave as $N^0$ ($\omega = 0$). The tree structure also implies that the cut line must be on one of the leaves of the tree, otherwise the resulting diagram will not be 1PI. Thus, the resulting diagram should have the following structure:

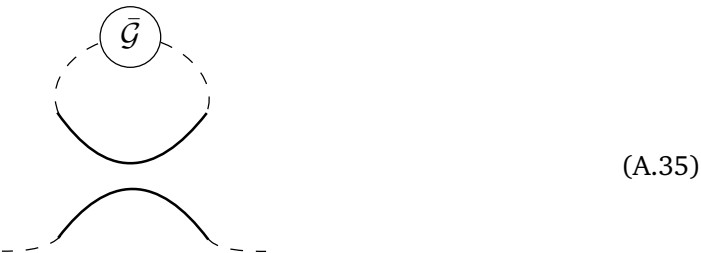

$$(A.35)$$

where the remaining graph $\bar{\mathcal{G}}$ in the white disk corresponds to the part of the diagram minus the vertex where the leaf has been opened.

It is not difficult to convince oneself that these graphs contribute to the 2-point function $\mathfrak{G}$. In formula:

$$\Sigma_{\mu\nu} = -4\delta_{\mu\nu}\left(\frac{g}{8N}\right)\sum_{\mu=1}^{N}\mathfrak{G}_{\mu\mu} \to -\frac{1}{2}g\left(\int \mu(\lambda)\lambda d\lambda\right)\delta_{\mu\nu}, \qquad (A.36)$$

where on the right-hand side, we take into account the factor 4 counting the number of independent contractions accordingly to the one-loop diagram. Then, $\Sigma$ is diagonal and this result generalizes our previous conclusions (equation (53)): only the mass is shifted in the large $N$ limit.

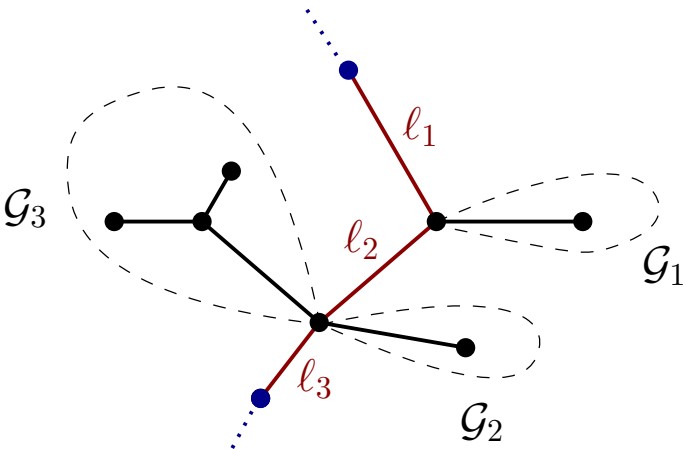

Figure 41: Typical leading order 4-point diagram.

In the same vein, 4-point diagrams can be resumed as a compact expression. Leading order 1PI 4-point diagrams can be obtained from 1PI 2-point diagrams by opening a second dotted edge, which for the same reason as above must be located on a leaf. For example, the diagram 41 illustrates the general structure: the two "open" blue sheets labelled with a dotted line are linked together by a single path $\mathcal{L}$ of minimal length formed by the red segments $\mathcal{L} := (\ell_1, \ell_2, \ell_3)$. We will call this path the skeleton of the graph, the length $L$ of the skeleton being given by the number of segments constituting it, here $L = 3$. Each segment of the skeleton is attached to the next by a vertex, to which one are attached related components $(\mathcal{G}_1, \mathcal{G}_2$ and $\mathcal{G}_3)$.

A moment of reflection shows that these components are actually diagrams involved in the development of the two-point function $\mathfrak{G}$. All these contributions can be resumed in effective loops where the propagator $\tilde{D}^{-1}$ can be replaced by $\mathfrak{G}$. We define as $\pi_p^{(4)}(\mu_1, \mu_2, \mu_3, \mu_4)$ the vertex function[34] corresponding to the resummation of trees which skeleton has length $L = p$ up to the replacement $\mathfrak{G} \to C$. The resulting diagram takes the form:

$$\pi_3^{(4)}(\mu_1, \mu_2, \mu_3, \mu_4) = \overset{\mu_1}{\underset{\mu_2}{\Big)}} \Big( \overset{\textcircled{C}}{\underset{\textcircled{C}}{\Big)} \Big( \overset{\textcircled{C}}{\underset{\textcircled{C}}{\Big)} \Big( \overset{\mu_3}{\underset{\mu_4}{} + \text{perm}, \tag{A.37}$$

where perm denotes the permutation of external edges.

This can be formally rewritten as:

$$\pi_3^{(4)}(\mu_1, \mu_2, \mu_3, \mu_4) = \left( \overset{\textcircled{C}}{\underset{\textcircled{C}}{\Big)} \right)^2 \left\{ \overset{\mu_1}{\underset{\mu_2}{\Big)}} \Big( \overset{\mu_3}{\underset{\mu_4}{} + \overset{\mu_1}{\underset{\mu_3}{\Big)}} \Big( \overset{\mu_2}{\underset{\mu_4}{} + \overset{\mu_1}{\underset{\mu_4}{\Big)}} \Big( \overset{\mu_2}{\underset{\mu_3}{} \right\}. \tag{A.38}$$

The loop integral factorizes outside and the external contributions are only products of Kronecker deltas. In formula:

$$\pi_3^{(4)}(\mu_1, \mu_2, \mu_3, \mu_4) = \gamma_3 \frac{g}{N} \Upsilon^2 (\delta_{\mu_1\mu_2}\delta_{\mu_3\mu_4} + \delta_{\mu_1\mu_3}\delta_{\mu_2\mu_4} + \delta_{\mu_1\mu_4}\delta_{\mu_3\mu_2}), \tag{A.39}$$

where $\gamma_3$ is a symmetry factor and $\Upsilon$ the *strength* of the loop,

$$\Upsilon := \frac{g}{N} \sum_{\mu=1}^{N} \mathfrak{G}_{\mu\mu}^2 \to g \int \mu(\lambda)\lambda^2 d\lambda. \tag{A.40}$$

---

[34]I.e. with external propagators amputated.

The symmetry factor $\gamma_3$ can be computed using perturbation theory, for the leading order two-loops diagrams involving three vertices. It corresponds to diagrams as (A.38), but with $C \to \tilde{D}^{-1}$. Each vertex generates a factor $-1/8$. The permutation of positions for all the vertex generates a factor $3!$, which is exactly compensated by the factor $1/3!$ arising from the expansion of the exponential. There is moreover an additional factor 2 per vertex, corresponding to the two different orientations which do not affect the topology of the graph and an additional factor 2 per loop, counting the number of contractions. We have therefore a factor $2^3$ to count all these permutations. Finally, each configurations for external edges, for instance $(\mu_1, \mu_2)$ on one side and $(\mu_3, \mu_4)$ on the other side must be multiplied by $4 = 2^2$, the configuration $(\mu_1, \mu_2)$ being equivalent to $(\mu_2, \mu_1)$. Hence:

$$\gamma_3 = -\frac{1}{4}. \tag{A.41}$$

Generalizing the argument, it is not hard to check that:

$$\gamma_p := \frac{(-1)^p}{2^{p-1}}, \tag{A.42}$$

and $\pi_p^{(4)}(\mu_1, \mu_2, \mu_3, \mu_4)$ must be read:

$$\pi_p^{(4)}(\mu_1, \mu_2, \mu_3, \mu_4) = -\frac{g}{N}\left(-\frac{g}{2}\int \mu(\lambda)\lambda^2 d\lambda\right)^{p-1}(\delta_{\mu_1\mu_2}\delta_{\mu_3\mu_4} + \delta_{\mu_1\mu_3}\delta_{\mu_2\mu_4} + \delta_{\mu_1\mu_4}\delta_{\mu_3\mu_2}). \tag{A.43}$$

The full 4-points vertex function $\Gamma^{(4)}(\mu_1, \mu_2, \mu_3, \mu_4)$ is then obtained by summing the contributions from $p = 1$ to $p = \infty$:[35]

$$\Gamma^{(4)}(\mu_1, \mu_2, \mu_3, \mu_4) = -\sum_{p=1}^{\infty} \pi_p^{(4)}(\mu_1, \mu_2, \mu_3, \mu_4), \tag{A.44}$$

which can be formally computed using (A.43), and we get:

$$\Gamma^{(4)}(\mu_1, \mu_2, \mu_3, \mu_4) = \frac{g/N}{1 + \frac{g}{2}\int \mu(\lambda)\lambda^2 d\lambda}(\delta_{\mu_1\mu_2}\delta_{\mu_3\mu_4} + \delta_{\mu_1\mu_3}\delta_{\mu_2\mu_4} + \delta_{\mu_1\mu_4}\delta_{\mu_3\mu_2}). \tag{A.45}$$

The effective coupling $g_{\text{eff}}$ is then defined for vanishing external momenta, namely:

$$g_{\text{eff}} = \frac{g}{1 + \frac{g}{2}\int \mu(\lambda)\lambda^2 d\lambda}. \tag{A.46}$$

To conclude, in the limit where $N$ is very large, it is easy to verify that the Ward identities are identically verified. Let us consider the model (A.31) and the generating functional:

$$Z[\chi] := \int p(\Psi) \exp\left(\sum_{\mu=1}^{N} \chi_\mu \psi_\mu\right) d\Psi, \tag{A.47}$$

and apply an infinitesimal rotation on $\psi_\mu$,

$$\psi_\mu \to \psi'_\mu := \psi_\mu + \sum_\nu \epsilon_{\mu\nu}\psi_\nu, \tag{A.48}$$

where $\epsilon \in \mathfrak{so}(N)$. The path integral being invariant under such a global translation of fields, we get:

$$\sum_{\mu,\nu} \int d\Psi p(\Psi) e^{\sum_{\mu=1}^{N} \chi_\mu \psi_\mu}\left(\tilde{\lambda}_\mu^{-1}\psi_\mu\psi_\nu - \chi_\mu\psi_\nu\right)\epsilon_{\mu\nu} = 0, \tag{A.49}$$

---

[35]The minus sign arise because we are aiming to define $\Gamma^{(4)}(0,0,0,0)$ as the effective coupling constant.

leading to:

$$\int d\Psi p(\Psi) e^{\sum_{\mu=1}^{N} \chi_\mu \psi_\mu} \left( (\tilde{\lambda}_\mu^{-1} - \tilde{\lambda}_\nu^{-1}) \psi_\mu \psi_\nu - (\chi_\mu \psi_\nu - \chi_\nu \psi_\mu) \right) = 0. \qquad (A.50)$$

Taking the second derivative for $\chi$, and setting $\chi = 0$ at the end of the derivation, we obtain:

$$(\tilde{\lambda}_\mu^{-1} - \tilde{\lambda}_\nu^{-1}) G^{(4)}_{\mu_1 \mu_2 \mu \nu} - (\delta_{\mu \mu_1} C_{\nu \mu_2} + \delta_{\mu \mu_2} C_{\nu \mu_1} - \delta_{\nu \mu_1} C_{\mu \mu_2} - \delta_{\nu \mu_2} C_{\mu \mu_1}) = 0. \qquad (A.51)$$

Because $C_{\mu \nu} = \lambda_\mu \delta_{\mu \nu}$, this is further simplified by:

$$(\tilde{\lambda}_\mu^{-1} - \tilde{\lambda}_\nu^{-1}) G^{(4)}_{\mu_1 \mu_2 \mu \nu} - (\lambda_\nu - \lambda_\mu)(\delta_{\mu \mu_1} \delta_{\nu \mu_2} + \delta_{\mu \mu_2} \delta_{\nu \mu_1}) = 0. \qquad (A.52)$$

We assume $\mu \neq \nu$. The 4-point function $G^{(4)}_{\mu_1 \mu_2 \mu \nu}$ admits the following decomposition:

$$G^{(4)}_{\mu_1 \mu_2 \mu \nu} = C_{\mu_1 \mu} C_{\mu_2 \nu} + C_{\mu_1 \nu} C_{\mu_2 \mu} - \Gamma^{(4)}_{\mu_1 \mu_2 \mu \nu} \lambda_{\mu_1} \lambda_{\mu_2} \lambda_\mu \lambda_\nu. \qquad (A.53)$$

We get:

$$(\tilde{\lambda}_\mu^{-1} - \tilde{\lambda}_\nu^{-1}) \lambda_\mu \lambda_\nu - (\lambda_\nu - \lambda_\mu) = \frac{g_{\text{eff}}}{N} (\tilde{\lambda}_\mu^{-1} - \tilde{\lambda}_\nu^{-1}) \lambda_\mu^2 \lambda_\nu^2, \qquad (A.54)$$

or:

$$(\tilde{\lambda}_\mu^{-1} - \tilde{\lambda}_\nu^{-1}) - (\lambda_\mu^{-1} - \lambda_\nu^{-1}) = \frac{g_{\text{eff}}}{N} (\tilde{\lambda}_\mu^{-1} - \tilde{\lambda}_\nu^{-1}) \lambda_\mu \lambda_\nu. \qquad (A.55)$$

This relation shows that the dominant corrections affect only the mass and that the corrections to the wave function corrections cancels identically as $N \to \infty$. An explicit expression can be obtained if we assume that $p_\mu^2 = Z \tilde{p}_\mu^2$ for small $p_\mu$. In that way, setting $p_\mu = 0$, we get:

$$1 - Z = \frac{1}{N} \frac{g_{\text{eff}}}{(m^2)^2}. \qquad (A.56)$$

It would have been different if the Ward identity had included an effective loop, compensating the $1/N$ factor coming from the vertex. This loop does not appear for vector models, but it does in tensor models for which Ward identities give non-trivial results [108, 121–123].[36] Although this model is excessively simple and its resolution obvious, it is however inappropriate for signal detection.

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
