# Peer review of "Functional Renormalization Group Approach for Signal Detection"

_SciPost Physics, doi:SciPost Phys. Core 7, 077 (2024)_

## Round 1 · Referee Report · Anonymous (Referee 1) · 2023-10-30

Report

The authors present a review on a renormalization group approach to signal
detection.I believe that this work is valuable on a conceptual level and the
subject is interesting, however I see some issues with the presented material
that I feel should be covered better or if there are no answers to them it should
be clearly said that this is the case.

Since the paper is complex and long I shall discuss it by logical parts

A) In the first part of the paper authors make a nice introduction to the topic
covering first 2 sections.

B) In the 3rd section the authors attempt to make a link with the field theory

C) In the 4th and 5th sections authors present explicit calculations using renormalization group
procedures.

I have minor questions related with the introductiory parts:

A1) It is not clear from the text what is shown in Fig.4. upper panel As I
understand the MP distribution is obtained from multiplying 2 fully random
matrices. How does the figure illustrate deviation from universality
when the difference could also be understood as a MP distribution with a
different cutoff.

A2) A picture is introduced by which dimensionality is illustrated to be a deciding factor in the
relevance of eigenvectors because it induces different spectra in momentum space. I think this is
a dangerous analogy because the dimensionality is not all that comes into play. Which eigendirections
survive in the IR limit depends on the field theory as well, so I do not unerstand what such an
analogy gains us.

B1) My biggest problem with the paper is the 3rd section. I find it confusing and vague and I recommend
rewriting it in a simpler and more transparent way, sacrificing some of the material for clarity.
There seem to be a lot of ideas there, however as far as I see the only important point is left unanswered
and this is: how does looking at field theoretical models with RG help us detect signals
in continuous spectra? My point is this, when we are doing renormalization group on a field
theoretical model such as the phi^4 theory, we are at the end interested in low energy excitations
of the problem. Why are these excitations relevnat for the signal detection? If we look in comparison
the nicely introduced Wigner semicircle spectrum and some peaks imbedded in it, nothing guarantees
that what happens in the low energy limit of the theory is relevant to detecting whatever peaks
we want to detect. So what I recommend to the authors is offering as simple as possible answer
to this question as the main point point of Sec 3.

B2) If the description of such signals as said in my previous point is not what authors have in mind
they should be clear about what they are hoping to achieve with their RG approach. What kind of a signal
is a signal that they are intersted in and that is analogous to the IR degrees of freedom that
have survived after the integration over small wavevector modes

B3) As far as I see in later the authors introduce a phi^4 like field theory with in general
a nonlocal kernel of interaction as we see in Eq. 3.4. which they then discuss near the Gaussian
limit and as the field theory teaches us find that in some relevant cases the Gaussian theory is unstable
and that higher couplings above quadratic need to be taken into the account. What I find interesting
is that their theory is massive. Why is this the case or in other words why is this case relevant for
their consideration?

B4) Their interaction kernel was introduced as nonlocal. If this is the case why then do they
only consider the standard derivative expansion of the gradient term in p^2n? Why is their field theory
not e.g. with long range interactions? In this case they would have e.g. a fractional gradient
term.

B5) Also why do they only consider the simplest situation of the scalar phi^4 theory, let me elaborate? In
3.3 they discuss the symmetries of the model. Why would they not have some more complicated order parameter
field given the symmetries?

In section 4 they introduce the Wetterich-Morris formalism and they do two kinds of calculations: a)
the calculation of flows at local potential level and b) in field expansion either around 0 or the minimum.
To me this looks completely like a standard calculation that was done a lot of times before in virtually
all the reviews on NPRG. What I think they do different although I do not think it is said in a clear way
is impose an arbitrary distribution of q modes, which I presume, gives them a difference from the completely standard
calculation.

They motivate this calculation by stating (*) that:

"A motivation justifying the non-perturbative formalism use was the surprising
observation that, for most common noise, models power counting shows that the
first perturbations to the Gaussian model - the quartic and sixtic terms - are always
relevant at the tail of the spectrum, (see empirical statement (1))."

The gist of the standard calculation on the other hand is that if you tweak the initial condition
above T_c then you flow into the disordered phase and if you tweak them below T_c you flow into
the ordered phase.

C1) The difference between the standard calculation and their calculation should be stressed better
because it took me some time to understand that they are indeed not doing a standard phi^4
calculation.

C2) If they claim that given their distribution of momenta the picture of the flow is
pretty much as in the standard case (which I think they do since evidence to this effect is
given later in sec 5), I do not understand what they mean by their statement (*) above introduced
as motivation. Do they mean that if they tweak the couplings right that the dimensionless couplings are
going to grow? If this is so this statement has nothing to do with nonperturbative physics it just has
to do with the flow into the ordered phase. Even if you flow into the disordered phase you are going
to obtain some finite dimensionfull values for the couplings when the flow stops. Authors please
clarify this!

C3) The authors seem to claim that there is no fixed point of their flow, however Fig 21 testifies that
there might be one. This is very peculiar. The terminal part of the flow should be dominated by the shape
momentum distribution near q=0. Why do the authors not discuss the asymptotic equations separately?
Considering these equations might give an analytic answer whether there is a nontrivial fixed point or not
in their case.

At the beginning of Sec 5.2 they state (**):

"In section 4, we illustrated the dependence of the canonical dimensions on the
scale, for an MP distribution, and emphasized two points. The first point is that
at a large scale only two couplings are relevant, the quartic and the sixtic, the
latter tending to be asymptotically marginal."

This statement is misleading and incomplete. As said before, in all the cases either the flow to
disordered or the ordered phase if we look at the full function U(phi), it is going to be a
nontrivial function. In the cases when the flow is into the ordered phase it is going to
develop a flat region near phi=0. If the initial condition is tweaked to critical then
you flow to the fixed point. All this is well known within the standard picture.

C4) My question is what kind of initial conditions do they have in mind when they tell their
statement?

C5) What is the relevance of the initial conditions of the flow to their program of signal detection?
In Sec 5 I seem to see that they are interested in the critical situation. Why is that?

As a general statement, I beg the authors to improve the language
throughout the text. In many places the language constructions are atypical of
English language and words are wrong.

All in all I appreciate the author's effort and I agree with publishing the article provided
they take the criticisms I laid out into the account in good faith. In this case it
would certainly meet the requirements of the journal. This will also make the article
much easier to read and accessible to a wider audience.

---

## Round 2 · Referee Report · Anonymous (Referee 1) · 2024-10-6

Report

First of all let me apologize to the authors for being late to review the
article. I have not received the request by email, I only was assigned to review
it on SciPost webpage and only by chance saw that the review was assigned to me
while logging into the webpage on some other business (!!).

Secondly overall I am happy with the author's responses. I understand that it is
a difficult work in progress and that open questions remain. However the version 2
that is found on the SciPost page seems to be largely unchanged from the first
version that the authors have submitted modulo pushing a big part of the Sec 3 into
the appendix which I feel is a very good idea. I can not find the corrections that
authors claim to have done. Maybe they resubmitted some old version of the manuscript?

In particular let me go into the specific points I raised.

A1) Maybe you can plot mu(lambda) on a log scale from epsilon to 1? That would make
the signal that is in the tail more visible.

A2) I do not see the footnote that they claim to have put anywhere. The v2 version
in this part seems completely identical as the v1.

B1) I believe that this point has confused me by my own fault so this point that
I raised I take back as well as B2. One can indeed capture in RG high energy
excitations by summing all the excitations up to the IR limit.

B4) The response is sufficient. They might want to try looking into the Blaizot-
Mendes-Wchchebor (BMW) scheme for some of their future work because it captures
without approximations the full momentum dependence. Might be what they need to
see with more precision the effects they are looking for.

C1) I do not see this comment in v2 of the paper!

C2) I do not see any correction being made! However if they wish to motivate the
nonperturbative formalism it is enough that they wish to follow the flow from the
UV initial condition until all the modes are integrated out.

C3) This answer is completely unclear to me. If the flow "cancels out" how is it not
0 or close to 0? So the figure I was referring to was figure 21 in the previous
text but now figure 17 in v2. I see a completely identical discussion of the
figure 17 as that of figure 21! No change that they claim they did was implemented!

Overall assuming that the authors submitted the wrong version and have
taken into the account the points they discussed in the reply I am OK with this
article being published providing that they first upload the correct version.

Recommendation

Publish (meets expectations and criteria for this Journal)

---

## Round 2 · Author Response

Dear referees, dear editors,
We want to apologize for this delay in our response, which was largely be-
yond our control, and linked to the health problems of one of us. We hope you
will find our answers worthy of your expectations and patience. Furthermore,
we also want to thank the referee for his work, he took the time to read our
manuscript. Even though we did not agree with all of his remarks (which we
justify in what follows), this report also allowed us to improve the first version
and the presentation of our work.

About the part A

A1) It is not clear from the text what is shown in Fig.4. upper panel As I
understand the MP distribution is obtained from multiplying 2 fully random
matrices. How does the figure illustrate deviation from universality when the
difference could also be understood as a MP distribution with a different cutoff.
Answer A1: It’s difficult to illustrate this point on a figure, and one might
get this impression at first glance, but one of the major points highlighted
in our work is precisely that the renormalization group makes the difference!
At the most elementary level, since canonical dimensions are universally fixed
asymptotically (Figure 14). In contrast, the presence of a signal affects the
asymptotic value of these dimensions.

A2) A picture is introduced by which dimensionality is illustrated to be a decid-
ing factor in the relevance of eigenvectors because it induces different spectra
in momentum space. I think this is a dangerous analogy because dimension-
ality is not all that comes into play. Which eigendirections survive in the IR
limit depends on the field theory as well, so I do not understand what such an
analogy gains us.

Answer A2: This argument assumes that we know which field theory we are
talking about, or the number of fields involved in the construction of a given
interaction, for example. However, once this choice is given, the distribution
of moments is the only parameter deciding the relevance of the interactions.
Note that when we talk about "power counting", we generally assume that the
interactions are fixed, however, this showed us the importance of clarifying our
explanation, which we did (by a footnote).

About the part B

Beforehand, we want to remind you that this is not an article, but a review,
based on a series of published articles. Also, we do not claim to be exhaustive,
and certain points noted by the referee are still open questions. We asked
ourselves a lot of questions, we have so far dealt with a few, and this review
was an opportunity for a temporary assessment (requested by some of our
colleagues). In our writing, we have endeavored to follow the (non-definitive)
point of view adopted in the publications on which we base ourselves. Finally,
we also want to note that another article followed this review [2310.07499],
which has just been accepted into a journal, and which addresses yet other
aspects of the problem.

B1) My biggest problem with the paper is the 3rd section. I find it confusing and
vague, and I recommend rewriting it more simply and transparently, sacrificing
some of the material for clarity. There seem to be a lot of ideas there, however
as far as I see the only important point is left unanswered and this is: how does
looking at field theoretical models with RG help us detect signals in continuous
spectra? My point is this, when we are doing a renormalization group on a field
theoretical model such as the phi4 theory, we are at the end interested in low-
energy excitations of the problem. Why are these excitations relevant for signal
detection? Let’s look in comparison at the nicely introduced Wigner semicircle
spectrum and some peaks embedded in it. Nothing guarantees that what happens
in the low energy limit of the theory is relevant to detecting whatever peaks we
want to detect. So what I recommend to the authors is offering as simple as
possible answer to this question as the main point of Sec 3.

Answer B1 We do not understand the point of view of the referee. The signal,
following our investigation, is indeed found in the region of large eigenvalues
(this is also confirmed by the numerical and notably dimensional analyses of
sections 4 and 5), and is also found at the basis of ordinary signal detection
protocols like PCA. We could also say that this is a central hypothesis here,
since we are looking at small deformations around universality (and notable
deviations from the canonical dimension always occur on the side of large
eigenvalues).

B2) If the description of such signals as said in my previous point is not what
authors have in mind, they should be clear about what they are hoping to achieve
with their RG approach. What kind of signal is a signal that they are interested
in and that is analogous to the IR degrees of freedom that have survived after
the integration over small wave vector modes

Answer B2: Let us remember that this is a review, and the style relates
to it. This position has already been defended in our article, and the sec-
tion concretely shows some results which clearly illustrate our expectations.
Concretely, we want to know how the presence of a signal affects the univer-
sal properties of the flow, which is the content (as illustrated for example in
Figures 16-20-21).

B3) As far as I see later the authors introduce a phi4 like field theory with in
general a nonlocal kernel of interaction as we see in Eq. 3.4. which they then
discuss near the Gaussian limit and as the field theory teaches us to find that in
some relevant cases, the Gaussian theory is unstable and that higher couplings
above quadratic need to be taken into the account. What I find interesting is
that their theory is massive. Why is this the case or in other words why is this
case relevant for their consideration?

Answer B3: Considering a phi4 theory is essentially suggested by power
counting. Indeed, the theory is massive, and this is essentially due to the
interpretation we have of the value of mass for IR theory: it is the inverse of
the largest eigenvalue!

B4) Their interaction kernel was introduced as nonlocal. If this is the case,
why then do they only consider the standard derivative expansion of the gradi-
ent term in p2n? Why is their field theory not e.g. with long-range interactions?
In this case, they would have e.g. a fractional gradient term.

Answer B4: This is one of the project we are currently studying and be
released soon. This kind of theory would make it possible to explore the
interior of the "bulk", and not just the region of large eigenvalues. To build
our interactions, we added another hypothesis by working with the relative
variable p, and not p2. On this subject, the second part of section 3 is
an addition to the work mentioned in the reference articles, and we
placed as an appendix in this new version.

B5) Also why do they only consider the simplest situation of the scalar phi4
theory, let me elaborate. In 3.3 they discuss the symmetries of the model.
Why would they not have some more complicated order parameter field given
the symmetries?

Answer B5: We appreciate the referee’s comment of the referee, and this
question is also one of our concerns. We went as simple as possible for these
first investigations, which seems to be sufficient in the deep IR, but we are
currently studying the possibility of other symmetries.

About the part C
General remark: We come from the general remark made upstream by the
referee: The form of the calculations differs little from the standard case.
However, for the sake of pedagogy and to avoid any confusion about the small
specificity of the field theory that we are considering, we preferred to give
sufficient details to the reader.

C1) The difference between the standard calculation and their calculation should
be stressed better because it took me some time to understand that they are in-
deed not doing a standard phi4calculation.

Answer C1: We have added a note (in blue) at the beginning of the
section on this subject. However, we assume that the reader is not necessarily
aware of the formalism of the NPRG.

C2) If they claim that given their distribution of momenta the picture of the
flow is pretty much as in the standard case (which I think they do since evidence
to this effect is given later in sec 5), I do not understand what they mean by
their statement (*) above introduced as motivation. Do they mean that if they
tweak the couplings right that the dimensionless couplings are going to grow?
If this is so this statement has nothing to do with nonperturbative physics it
just has to do with the flow into the ordered phase. Even if you flow into the
disordered phase you are going to obtain some finite dimensionfull values for
the couplings when the flow stops. Authors please clarify this!

Answer C2: We initially had difficulty understanding the referee’s com-
ment before realizing that the error came from us! we wrote "nonperturba-
tive formalism" instead of "vertex expansion formalism". We rely on
the small number of relevant parameters in the IR to justify it, which would
not be possible in the UV, where the canonical dimensions explode! We have
modified the sentence.

C3) The authors seem to claim that there is no fixed point of their flow, how-
ever, Fig 21 testifies that there might be one. This is very peculiar. The
terminal part of the flow should be dominated by the shape momentum dis-
tribution near q=0. Why do the authors not discuss the asymptotic equations
separately? Considering these equations might give an analytic answer whether
there is a nontrivial fixed point or not in their case.

Answer C3: This is an infrared ambiguity! There are no fixed points (because
the canonical dimensions depend on the scale), but there can exist "fixed trajec-
tories", along which the flow cancels out. What we see in the figure is precisely
the terminal form of one of these lines. We have also discussed these “asymp-
totic fixed points” in a recent series [2403.07577, 2403.12217, 2404.11915].
We added a short remark, in blue, on the bottom of the Figure.

C4) My question is what kind of initial conditions do they have in mind when
they tell their statement?

Answer C4: Our assertion is only based on the power counting, therefore
the critical exponents in the vicinity of the Gaussian point, and does not claim
to describe the flow only in this vicinity. We agree with the remarks of the
referee, but it was not in question at this level. In general, all our dimensional
arguments are only valid in the vicinity of the Gaussian point.

C5) What is the relevance of the initial conditions of the flow to their
program of signal detection? In Sec 5 I seem to see that they are interested in
the critical situation. Why is that?

Answer C5: This question goes beyond the study carried out in this manuscript.
In our investigation, we have mainly focused on the Gaussian region for the
definition of our couplings, and we have been able to see a relation between
the intensity of the signal and the appearance of a symmetry breaking. In our
recent investigation, we explore the dependence of the detection threshold (the
moment when the purple region touches the blue region in Figure 25) on the
value of the initial scale (which plays the role of the UV cut-off in ordinary
field theory). We can then define an optimal threshold for detection, which
allows us to set a cut-off value between "signal" and "noise".

Finally, we checked the English throughout the manuscript and improved
the sentences.

On the Behalf of all the authors
Dr Samary

---

## Round 3 · Author Response

We would like to thank the referee for making this comment on our manuscript. We have taken all comments into account. In fact, the version submitted was not the last one due to clumsiness. We have improved it and hope that the referee will be satisfied.
On the Behalf of all the authors
Dr Samary

---

## Round 3 · List of Changes

A1) We preferred to keep the original figure because, in our opinion, it makes
it easier to understand the other figures in the paper, particularly key figures such as Figures 10 and 16. Additionally, it visually illustrates the challenge posed by quasi-continuous spectra. We had also included the caricature figure
to further emphasize and illustrate this point. We hope the referee will not object to this choice. Please note that we have improved the readability of Figure 10 (previously Figure 14).
A2) We apologize for this oversight; indeed, there was an error in the file. The footnote has now been added on page 16 (footnote 9) and referenced again on page 41 (footnote 26).
B4) For some time, we have aimed to adapt the Blaizot-Mendes-Wschebor formalism to this problem, leveraging the fact that the exact 2-point function is known in the IR regime. However, certain subtleties in the construction mean that this work is still ongoing.
C1) We have added the requested comment in the revised version (page 35, in blue, at the end of the paragraph).
C2) We appreciate the referee’s remark and have incorporated it into the comment
which can now be found at the end of the introduction of the section 4. This time, it is included in blue, preceding the earlier remark.
C3) Once again, we apologize for the error in our previous submission. We have made slight improvements to the content of Remark 3, which is now placed on page 54, just above Figure 13. We believe this figure sufficiently addresses the referee’s question regarding the presence of a fixed point.
Finally, we checked the English throughout the manuscript and improved the sentences.

---

## Round 4 · Author Response

Dear editor
Enclose here the new version of our manuscript. We had a few compilation problems due to the action of all the authors. We solved this and removed the blue colouring in the manuscript. We would like to thank the referee for making this comment on our manuscript. We have taken all comments into account. In fact, the version submitted was not the last one due to clumsiness. We have improved it and hope that the referee will be satisfied.
Best Regards
Dr Samary

---

## Round 4 · List of Changes

A1) We preferred to keep the original figure because, in our opinion, it makes
it easier to understand the other figures in the paper, particularly key figures such as Figures 10 and 16. Additionally, it visually illustrates the challenge posed by quasi-continuous spectra. We had also included the caricature figure
to further emphasize and illustrate this point. We hope the referee will not object to this choice. Please note that we have improved the readability of Figure 10 (previously Figure 14).
A2) We apologize for this oversight; indeed, there was an error in the file. The footnote has now been added on page 16 (footnote 9) and referenced again on page 41 (footnote 26).
B4) For some time, we have aimed to adapt the Blaizot-Mendes-Wschebor for- malism to this problem, leveraging the fact that the exact 2-point function is known in the IR regime. However, certain subtleties in the construction mean that this work is still ongoing.
C1) We have added the requested comment in the revised version (page 35, at the end of the paragraph).
C2) We appreciate the referee’s remark and have incorporated it into the footnote,
which can now be found at the end of the introduction of the section 4. This time, it is included in the sentences preceding the earlier remark.
C3) Once again, we apologize for the error in our previous submission. We have made slight improvements to the content of Remark 3, which is now placed on page 54, just above Figure 13. We believe this figure sufficiently addresses the referee’s question regarding the presence of a fixed point.
Finally, we checked the English throughout the manuscript and improved the sentences.
On the Behalf of all the authors Dr Samary

---

## Editorial Decision

published